# Targeting GRPR for sex hormone-dependent cancer after loss of E-cadherin

Jérémy H. Raymond[1,2], Zackie Aktary[1,2], Marie Pouteaux[1,2], Valérie Petit[1,2], Flavie Luciani[1,2], Maria Wehbe[1,2], Patrick Gizzi[3], Claire Bourban[3], Didier Decaudin[4], Fariba Nemati[4], Igor Martianov[5], Irwin Davidson[5], Catherine-Laure Tomasetto[5], Richard M. White[6], Florence Mahuteau-Betzer[7,8], Béatrice Vergier[9], Lionel Larue[1,2,10 ✉] & Véronique Delmas[1,2,10]

Sex inequalities in cancer are well documented, but the current limited understanding is hindering advances in precision medicine and therapies[1]. Consideration of ethnicity, age and sex is essential for the management of cancer patients because they underlie important differences in both incidence and response to treatment[2,3]. Age-related hormone production, which is a consistent divergence between the sexes, is underestimated in cancers that are not recognized as being hormone dependent[4–6]. Here, we show that premenopausal women have increased vulnerability to cancers, and we identify the cell–cell adhesion molecule E-cadherin as a crucial component in the oestrogen response in various cancers, including melanoma. In a mouse model of melanoma, we discovered an oestrogen-sensitizing pathway connecting E-cadherin, β-catenin, oestrogen receptor-α and GRPR that promotes melanoma aggressiveness in women. Inhibiting this pathway by targeting GRPR or oestrogen receptor-α reduces metastasis in mice, indicating its therapeutic potential. Our study introduces a concept linking hormone sensitivity and tumour phenotype in which hormones affect cell phenotype and aggressiveness. We have identified an integrated pro-tumour pathway in women and propose that targeting a G-protein-coupled receptor with drugs not commonly used for cancer treatment could be more effective in treating E-cadherin-dependent cancers in women. This study emphasizes the importance of sex-specific factors in cancer management and offers hope of improving outcomes in various cancers.

Cancer, which is one of the leading causes of premature mortality in humans, is known to have multiple risk factors. Some of these factors are intrinsic to the tumour or cell of origin, but many are extrinsic, originating from the tumour environment or individual behaviour. Sex and age have various effects on tissue and tumour exposure in cancer biology, but their role and, most importantly, their interactions remain poorly understood. Epidemiological data have revealed numerous sex differences in the incidence of various cancers in non-reproductive organs, although the role of sex hormones is sometimes suspected but little studied[1,6,7]. Understanding these sex- and age-related dynamics is essential for improving prevention and treatment strategies.

## E-cadherin as an oestrogen regulator

Despite men having an overall higher cancer risk than women, our sex- and age-stratified analysis of global epidemiological data reveals a higher risk of cancer incidence among premenopausal women (Fig. 1a

and Extended Data Fig. 1a). Of 24 cancer types analysed, 13 showed a significant female bias during this period (P = 0.0248; Extended Data Fig. 1a–f and Supplementary Table 1), with malignant melanoma being the most prominent (Fig. 1b and Extended Data Fig. 1c). Melanoma occurs most frequently in pregnant women and those exposed to high oestrogen levels during puberty[8–10]. This coincides with peak oestradiol levels in young women, whereas testosterone remains stable across ages, underscoring the potential role of oestrogen in initiating cancer (Fig. 1c and Extended Data Fig. 1g).

Genes central to the main female-biased cancers (melanoma, gastric and thyroid), breast cancer and ESR1, which encodes oestrogen receptor-α (ERα), include CDH1 (which encodes E-cadherin), CCND1 (cyclin D1), BRAF and KRAS (Fig. 1d and Supplementary Table 2). Of these, only CDH1 mRNA levels were lower in young women than in men or older women (Fig. 1e), indicating that reduced CDH1 expression could increase oestrogen-driven cancer susceptibility, particularly in melanoma.

[1]Normal and Pathological Development of Melanocytes, Institut Curie, PSL Research University, INSERM U1021, Orsay, France. [2]Université Paris-Saclay, CNRS UMR 3347, Orsay, France. [3]CNRS UMS3286, Plateforme de Chimie Biologique Intégrative de Strasbourg, Strasbourg University/Strasbourg Drug Discovery and Development Institute (IMS), Strasbourg, France. [4]Laboratory of Preclinical Investigation, Department of Translational Research, Institut Curie, PSL University Paris, Paris, France. [5]IGBMC, CNRS UMR7104, INSERM U1258, Université de Strasbourg, Illkirch, France. [6]Ludwig Institute for Cancer Research, University of Oxford, Oxford, UK. [7]CNRS UMR9187, INSERM U1196, Chemistry and Modeling for the Biology of Cancer, Institut Curie, Université PSL, Orsay, France. [8]Université Paris-Saclay, CNRS UMR 9187, INSERM U1196, Orsay, France. [9]Service de pathologie CHU de Bordeaux et Eq. Translational Research on Oncodermatology and Orphean Skin Diseases (TRIO2) BoRdeaux Institute of onCology (BRIC) UMR 1312, INSERM/Université de Bordeaux, Bordeaux, France. [10]These authors contributed equally: Lionel Larue, Véronique Delmas. ✉e-mail: lionel.larue@curie.fr

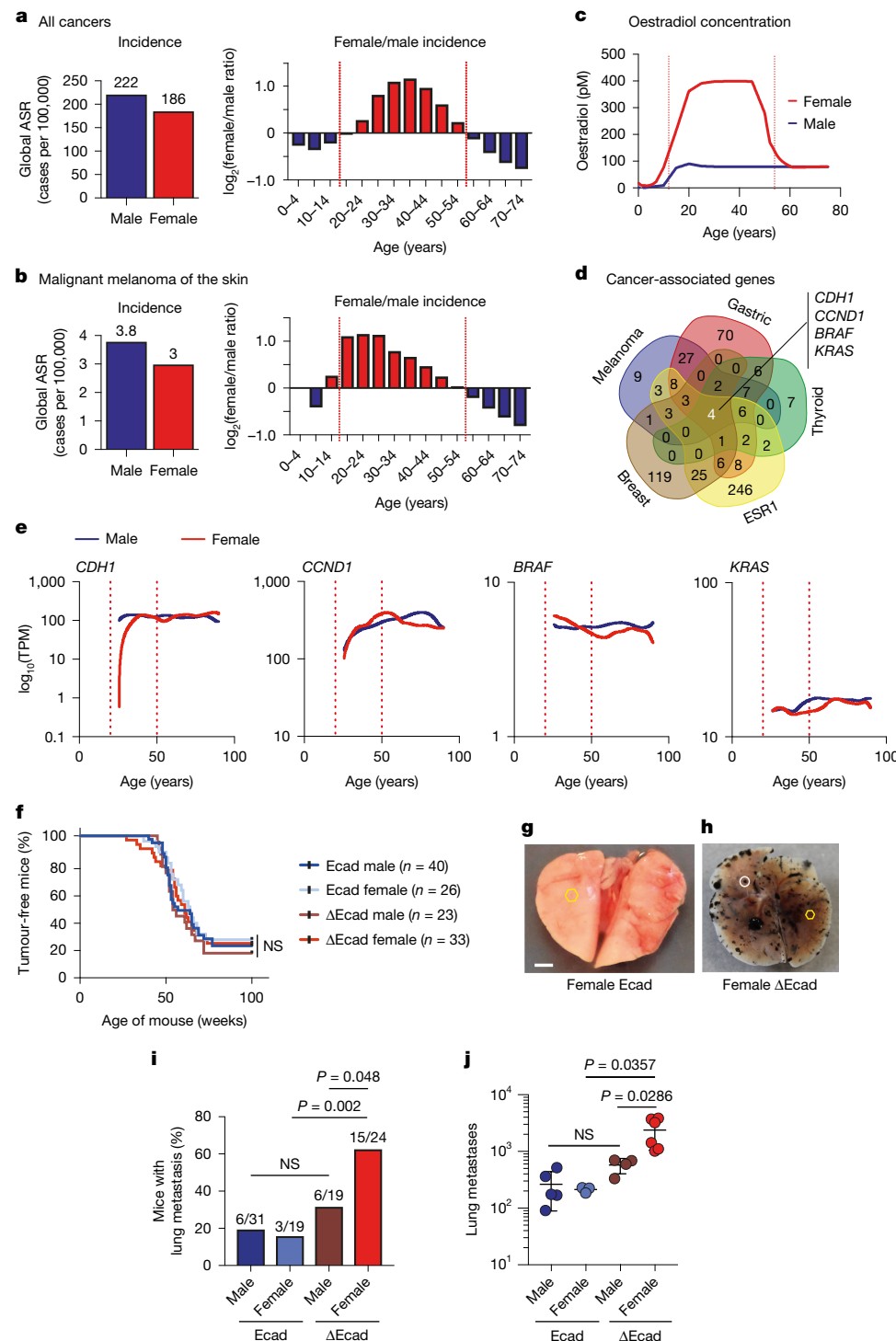

**Fig. 1 | E-cadherin is a crucial hub in oestrogen-mediated cancer responses with sex-dependent effects on melanoma metastasis. a**,**b**, Incidence ratios (women/men) for all cancers (**a**) and malignant melanoma of the skin (**b**) across different age groups, depicted as age-standardized rates (ASR). Data sourced from GLOBOCAN 2020 (ref. 6). **c**, Plasmatic oestradiol concentration stratified by age and sex. **d**, Venn diagram highlighting the overlap between genes associated with cancer, demonstrating a peak in incidence in women aged 15–55 (with melanoma, breast cancer, thyroid cancer and gastric cancer) and genes associated with *ESR1*. The four indicated genes are at the intersection of all five categories. **e**, Expression of *CDH1*, *CCND1*, *BRAF* and *KRAS* in human melanoma (TCGA database) stratified by sex and age. TPM, transcripts per million reads. **f**, Kaplan–Meier survival curves for melanoma-free mice categorized by E-cadherin status and sex; *n* is the number of mice per condition. No significant differences were observed by log-rank analysis. **g**,**h**, Representative lung images from female mice with primary melanoma for Ecad (**g**) or mutated Ecad (**h**). Yellow hexagons, micro-metastases; white circles, macro-metastases; scale bar, 2 mm; **g** and **h** are at the same scale. **i**, Frequency of lung metastasis in mice categorized by Ecad and sex status. **j**, Metastasis quantification based on Ecad and sex status. Significance was assessed by chi-square test for metastasis proportions and two-sided Mann–Whitney test adjusted for multiple testing by the Benjamini–Hochberg method for metastasis counts. NS, not significant.

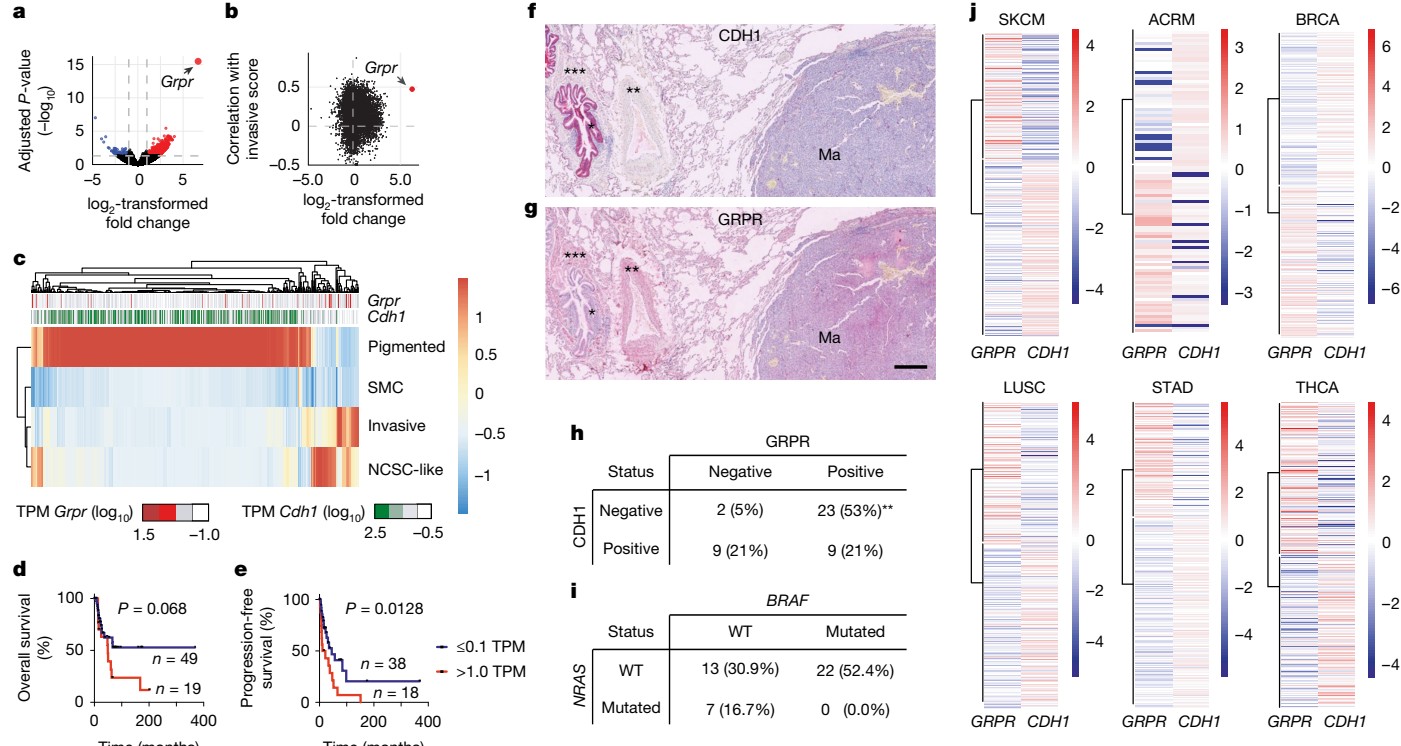

**Fig. 2 | Loss of *Cdh1* Induces *Grpr* expression. a**, Volcano plot illustrating differential gene expression between female Ecad and ΔEcad tumours, with *Grpr* indicated. **b**, Scatter plot showing the correlation of each gene's expression with the invasive score in human tumours, plotted against its differential expression between female ΔEcad and Ecad tumours. Pearson correlation coefficients were used to assess gene-invasive score associations. **c**, Heatmap clustering TCGA SKCM samples based on predominant cell-state signatures. SMC, starved-like melanoma cells; NCSC, neural crest cell-like. **d,e**, Kaplan–Meier survival curves for overall survival (**d**) and progression-free survival (**e**) of TCGA SKCM women categorized by *GRPR* expression (low or absent, ≤0.1 TPM; expressed, >1 TPM).

A log-rank test was used to evaluate significance. **f,g**, Immunohistochemistry staining for CDH1 (**f**) and GRPR (**g**) in human lung melanoma (Ma) metastases. *, Bronchi serve as an internal positive control for E-cadherin staining; ** and ***, smooth muscle acts as internal positive controls for GRPR. Scale bar (300 μm) applies to both **f** and **g**. **h,i**, Metastasis classification based on CDH1 and GRPR expression (**h**) and *NRAS* and BRAF status (**i**) in lung metastasis samples. **P < 0.01, Fisher's exact test. **j**, Heatmap displays showing *GRPR* and *CDH1* mRNA expression in melanomas and various carcinomas from the TCGA database and GSE162682. BRCA, breast-invasive carcinoma; LUSC, lung squamous-cell carcinoma; STAD, stomach adenocarcinoma; THCA, thyroid carcinoma.

## E-cadherin loss drives sex-biased metastasis

To investigate the role of E-cadherin in female-biased melanoma, we developed a melanoma mouse model with conditional *Cdh1* deletion in melanocytes (*Tyr::CreA*/°; *Cdh1*$^{F/F}$). No melanoma developed within two years, indicating that *Cdh1* loss alone is insufficient for tumour initiation. To induce melanoma, we combined *Cdh1* loss with the NRAS(Q61K) mutation using the *Tyr::NRAS*$^{Q61K/°}$; *Cdkn2a*$^{+/-}$ model[11]. We chose *NRAS* over *BRAF* because there is more frequent *CDH1* downregulation in *NRAS*-mutant melanomas and a lack of targeted therapies for *NRAS*-mutated cases (Extended Data Fig. 1h).

We monitored melanoma development in *Tyr::NRAS*$^{Q61K/°}$; *Cdkn2a*$^{+/-}$;°/°;*Cdh1*$^{F/F}$ (Ecad) and *Tyr::NRAS*$^{Q61K/°}$;*Cdkn2a*$^{+/-}$;*Tyr::CreA*/°; *Cdh1*$^{F/F}$ (ΔEcad) mice. *Cdh1* loss did not affect melanoma onset, primary tumour count or penetrance (Fig. 1f). However, lung metastasis was significantly higher in female ΔEcad mice: 63% (15 of 24) had metastases compared with 16% (3 of 19) of Ecad female mice (Fig. 1g–i). Both micro- and macro-metastases were more frequent in female ΔEcad mice compared with other groups (Fig. 1j and Extended Data Fig. 1i,j). These findings indicate that *Cdh1* loss enhances melanoma metastasis in a sex-dependent manner, particularly affecting female mice.

## Ecad loss upregulates *GRPR* expression

RNA-seq analysis of eight primary tumours per genotype and sex revealed a marked upregulation of gastrin-releasing peptide receptor

(*Grpr*) in ΔEcad female melanomas compared with all other groups, including ΔEcad male mice (Fig. 2a, Extended Data Fig. 2a and Supplementary Tables 3–6). H3K27ac chromatin immunoprecipitation followed by sequencing (ChIP–seq) confirmed there were active *Grpr* promoter regions exclusively in ΔEcad female melanoma cell lines (Extended Data Fig. 2b), with expression maintained in female lung metastases (Extended Data Fig. 2c). This upregulation was independent of the *Tyr::Cre* line or the *Cdh1* melanoma model (Extended Data Fig. 2d).

The role of GRPR in promoting metastasis was supported by: first, its association with elevated invasive and neural crest cell-like gene signatures in human melanoma (Fig. 2b,c and Extended Data Fig. 2e–h); second, its reduced overall and progression-free survival in women with high *GRPR* levels, a trend not seen in non-sex-stratified data (Fig. 2d,e and Extended Data Fig. 2i,j); third, ECAD⁻/GRPR⁺ being the dominant phenotype in human lung melanoma metastases, regardless of genotype (Fig. 2f–i); and finally, the expression of GRP, which is GRPR's natural agonist, in human and mouse lungs (Extended Data Fig. 2k–m).

*GRPR* mRNA is widely expressed across tumours, including melanomas[12]. Notably, its expression is inversely correlated with *CDH1* in both skin cutaneous melanoma (SKCM) and acral melanoma (ACRM), as well as in several carcinomas (breast, lung, stomach and thyroid), according to TCGA data and ref. 13 (Fig. 2j). This indicates that E-cadherin may commonly regulate *GRPR* across cancer types.

## GRPR drives lung metastases and is targetable

The expression of GRP (the endogenous ligand of GRPR) in rodent and human lungs indicates that GRPR activation may drive metastasis-supporting mechanisms. We dissociated primary tumours and injected the cells into C57BL6/J mouse tail veins. Only ΔEcad melanoma cells expressing *Grpr* colonized lungs within 30 days, unlike Ecad-expressing cells (Extended Data Fig. 3a). When we engrafted melanoma tumours, ΔEcad female tumours showed significantly faster growth than Ecad male, Ecad female and ΔEcad male tumours (Extended Data Fig. 3b), indicating the cell-autonomous aggressiveness of ΔEcad female melanomas. Cell lines from these tumours confirmed *Grpr* production uniquely in ΔEcad female lines (Supplementary Table 7).

To test the role of GRPR in metastasis, we attempted *Grpr* knockout in ΔEcad melanoma cells and ectopic expression in Ecad cells. We could not knock down or knock out *Grpr* in female ΔEcad cells (Extended Data Fig. 3c–i), implying that GRPR is essential in the absence of E-cadherin. However, we successfully introduced *Grpr* into male mouse 1181 and human 501mel cell lines. *Grpr* expression and activation in these engineered cell lines closely mirrored that of endogenous *Grpr*[+] lines, such as 1057 and Dauv-1 (Extended Data Fig. 3j–m). Tail-vein injection of parental, control and *Grpr*-expressing cells demonstrated that GRPR expression alone is sufficient to promote lung metastasis (Fig. 3a–c). In both male and female NSG mice, 501mel cells ectopically expressing *GRPR* formed lung metastases, consistent with sex-independent expression driven by the CMV promoter.

To evaluate GRPR inhibition in vivo, we used luciferase-expressing 1057 ΔEcad female melanoma cells (1057Luc). After tail-vein injection, numerous Dct- and Grpr-positive lung metastases formed (Fig. 3d,e). Two GRPR antagonists, RC-3095 and PD-176252, had similar in vitro effects (Extended Data Fig. 3n,o), but RC-3095 was used owing to its higher metabolic stability (Extended Data Fig. 3p,q).

Mice injected with 1057Luc cells were randomized into vehicle or RC-3095 treatment groups (Extended Data Fig. 3r,s). Without treatment, thoracic luminescence appeared by day 24 and grew exponentially (Extended Data Fig. 3t). Targeting GRPR with RC-3095 significantly reduced lung colonization, as measured by luminescence and metastasis counts (Fig. 3f–i and Extended Data Fig. 3u). These results underscore the essential role of GRPR in lung metastasis formation, validated through both gain-of-function and pharmacological approaches.

## GRPR fuels key metastasis pathway

To uncover the cellular mechanisms by which GRPR activation by GRP promotes metastasis, we first assessed its effect on growth. In line with the in vivo data, ΔEcad female mouse melanoma cell lines gained colony-forming ability (Fig. 4a). Ectopic GRPR expression in mouse and human melanoma lines strongly induced (in 1181) or enhanced (in 1014 and 501mel) colony formation (Fig. 4b,c and Extended Data Fig. 4a). GRP stimulation promoted growth in GRPR-positive mouse (1057 and 1064) and human (MDA-MB-435S) cell lines, whereas co-treatment with GRP and antagonist RC-3095 (RC) blocked this effect (Fig. 4d,e and Extended Data Fig. 4b). As expected, GRP had no effect on GRPR-negative mouse (1181 and 1014) and human (501mel) cells, unless GRPR was ectopically expressed (Extended Data Fig. 4c–h).

RNA-seq analysis of GRP-stimulated human and murine melanoma lines indicated the activation of key cancer-related pathways, notably anoikis resistance and invasion (Extended Data Fig. 4i–n). GRPR activation reduced apoptosis in unattached mouse and human melanoma cells, indicating resistance to anoikis. Inhibition of GRPR with RC restored sensitivity to anoikis (Fig. 4f,g and Extended Data Fig. 4o–t). GRP did not promote anoikis resistance in male ΔEcad (1062) or female Ecad (1014) cells (Extended Data Fig. 4s,t).

GRP stimulation also promoted invasion exclusively in female GRPR-positive 1057 cells, but not in Ecad-positive (1181 male and 1039 female) or Ecad-negative male (1456) cells (Extended Data Fig. 4u–y). GRP-induced invasion in GRPR-positive lines (including 1057, 1064, MDA-MB-435S, Dauv-1 and GRPR-transduced 1014, 1181 and 501mel) was blocked by RC-3095 (Fig. 4h,i and Extended Data Fig. 4l–n,z–Af). As expected, this inhibition was absent in GRPR-negative cells treated with GRP (Extended Data Fig. 4aa–ac).

## GRPR induces YAP1 through Gα$_{q/11}$

G-protein-coupled receptors signal through various Gα and downstream pathways[12]. We used PamGene kinase assays to identify the main pathway activated by GRP/GRPR in melanoma cells. GRPR activation primarily triggered protein kinase C (PKC) and, to a lesser extent, PKAα and PRKX (Extended Data Fig. 5a). PKCs are typically activated by IP3/DAG signalling downstream of Gα$_{q/11}$, whereas PKA is linked to the cAMP pathway through Gα$_s$ (ref. 12). RNA-seq data supported Gα$_{q/11}$ and PKC activation after GRPR stimulation (Extended Data Fig. 5b). To assess receptor coupling with Gα$_{q/11}$, we measured IP3–IP1 production after GRP treatment. GRP triggered IP1 production in GRPR-positive, but not GRPR-negative cells, confirming Gα$_{q/11}$ activation. RC-3095 blocked this response (Extended Data Fig. 5c–h).

Gα$_q$ has been linked to YAP1 activation in non-cutaneous melanoma models[14]. All GRPR-induced cellular changes could be attributed to YAP1 activation[15–17]. In human cutaneous melanoma, GRPR levels correlated with YAP1 activation scores, and mouse ΔEcad female tumours showed a YAP1 signature that was absent in ΔEcad male or Ecad tumours (Extended Data Fig. 5i,j). We did not observe increased YAP1 levels in ΔEcad females or after GRPR expression in Ecad lines (Extended Data Fig. 5k,l), but YAP1 nuclear localization increased in GRP-treated cells, and less so with RC-3095 (Fig. 4j).

Transcriptomic analysis of GRPR-expressing cells (1057, 1064, 1181-Grpr, 1014-Grpr, Dauv-1 and 501mel-GRPR) showed increased YAP1 signatures and scores after GRPR activation, whereas GRPR-negative controls (1181-Ct and 1014-Ct) showed no change (Fig. 4k and Extended Data Fig. 5m–ac). YAP1 activation was also blocked by RC in GRPR-expressing cells (Extended Data Fig. 5r–w,aa–c). Finally, we confirmed that GRP/GRPR-driven invasion depends on Yap1 and Taz (Fig. 4l). In summary, GRPR activation by GRP stimulates Gα$_q$ signalling, activating a YAP1-regulated metastasis program.

## CDH1–CTNNB1–ESR1 drives female GRPR

H3K27ac ChIP–seq in mouse melanoma cell lines revealed female ΔEcad-specific signatures linked to β-catenin (LEF1), oestrogen receptor-α (ESR1) and YAP1/TEAD (Fig. 5a). *Cdh1* loss is often linked to enhanced β-catenin signalling, reflected here by increased promoter activation of β-catenin targets such as *Apcdd1*, *Axin2*, *Nkd1*, *Notum* and *Sp5*—primarily in ΔEcad female melanomas—with corresponding gene expression (Extended Data Fig. 6a–j). Inhibiting *Cdh1* and ectopic β-catenin expression, or reducing *Apc* in Ecad-expressing cells, elevated *Nkd1/Axin2*, *Esr1* and *Grpr* expression (Fig. 5b,c and Extended Data Fig. 6k–n). Conversely, β-catenin inhibition through siRNA or iCRT3 decreased the expression of *Nkd1*, *Esr1* and *Grpr* (Extended Data Fig. 6o), indicating that β-catenin acts upstream of *Esr1* and *Grpr*.

We identified a β-catenin binding site in intron 1 of *Esr1*, marked by H3K27ac, in female ΔEcad cells (Extended Data Fig. 6p), indicating direct regulation. *Esr1* regulation by β-catenin was validated in human melanoma using iCRT3 (Extended Data Fig. 6q,r). Inhibiting β-catenin (iCRT3 and siCtnnb1) reduced GRP-induced invasion in mouse melanoma (Extended Data Fig. 6s). Gene-set enrichment analysis (GSEA) comparison of Ecad and ΔEcad female tumours showed that ΔEcad tumours overexpressed genes tied to ERα activity (Extended Data Fig. 7a,b). Other sex hormone receptors (*Esr2*, *Gper1*, *Ar* and *Pgr*) were not expressed in murine melanoma, and only ΔEcad female lines

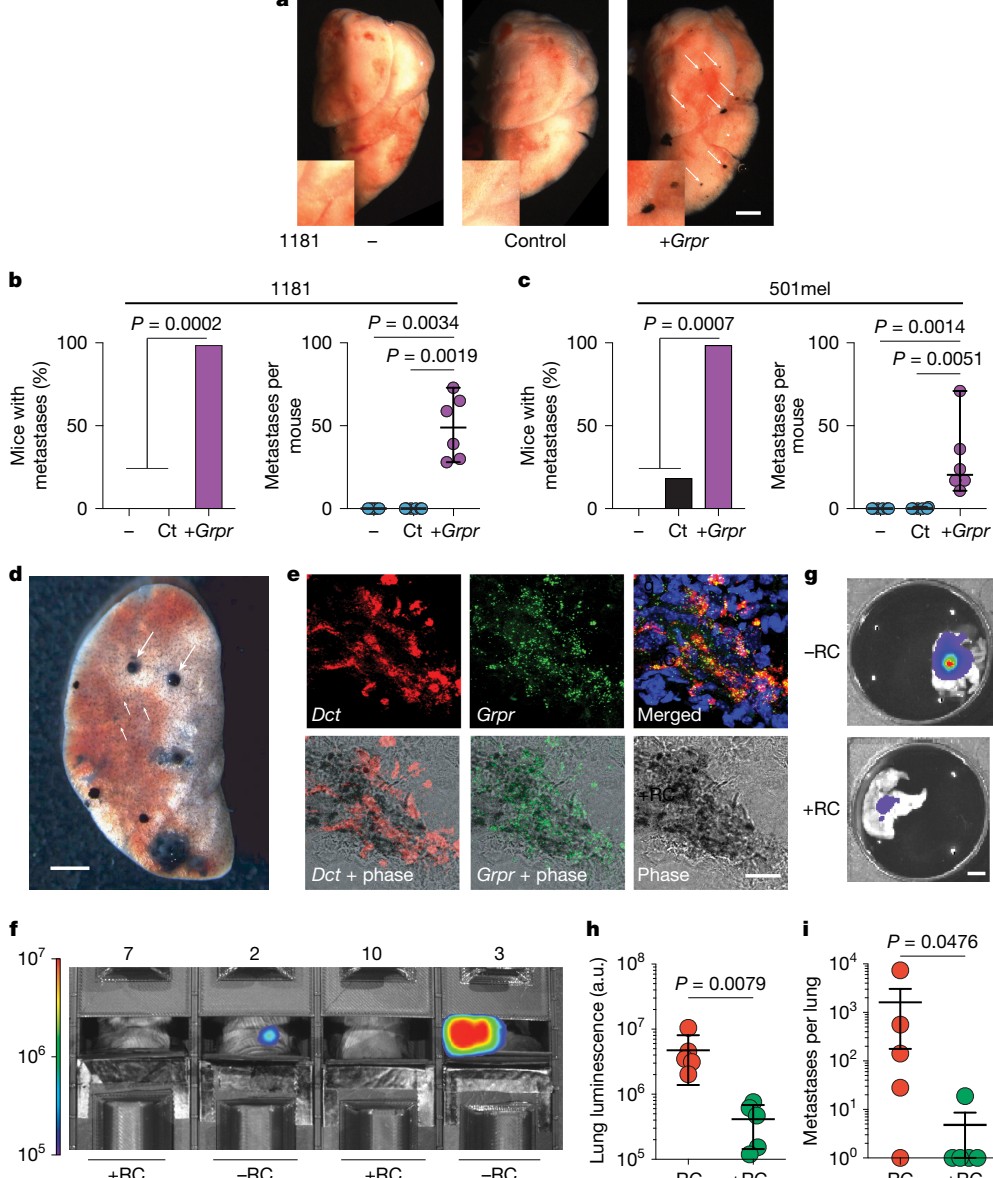

**Fig. 3 | *Grpr* expression and activation drive lung melanoma metastasis.**
**a**, Representative lung images taken 30 days after the injection of $5 \times 10^5$ male melanoma cells (1181) lacking *Grpr* (−, parental; Ct, control) or expressing exogenous *Grpr* (+*Grpr*) into male C57BL/6J mice tail veins. Scale bar, 2 mm. **b**,**c**, Percentage of mice generating metastases (left) and number of metastases per mouse (right) after injection of $5 \times 10^5$ male melanoma cells into tail veins for 1181, 1181 control and 1181 Grpr in C57BL/6J male mice (**b**) and 501mel, 501mel control and 501mel Grpr in NSG male mice (**c**). **d**, Lung metastases observed 28 days after intravenous injection of $5 \times 10^5$ 1057Luc Grpr$^+$ melanoma cells, demonstrating extensive lung colonization (small arrows) and proliferative foci (large arrows). Scale bar, 2 mm. **e**, RNAscope image showing colocalized *Grpr* (green) and *Dct* (red) mRNA in a lung metastasis in mice after tail-vein injection of 1057Luc cells. Scale bar, 50 μM. **f**–**i**, Lung metastases observed after tail-vein injection of $5 \times 10^5$ 1057-Luc Grpr$^+$ cells into C57BL/6J mice treated with RC-3095 (RC, mice 7 and 10) or not treated (mice 2 and 3). Luminescence (recorded by IVIS) from mouse thorax after RC treatment or without treatment (**f**); scale bar, 1 cm. IVIS assessment of ex vivo lung luminescence after 28 days of RC or vehicle treatment images (**g**) and quantification (**h**); scale bar, 4 mm. Estimation of number of metastases from five isolated independent lungs in RC-treated and untreated mice (**i**). Metastasis frequencies were compared by chi-squared test. Metastasis counts were compared using two-sided Mann–Whitney (two groups) or Kruskal–Wallis adjusted by a Dunn's test (multiple groups). a.u., arbitrary units. Data are shown as mean ± s.d. (**b**,**c** and **h**) or s.e.m. (**i**).

expressed both *Grpr* and *Esr1* (Fig. 5d and Extended Data Fig. 7c). ERα protein was also detected in these lines (Fig. 5e).

ERα bound active chromatin associated with the *Grpr* promoter (Extended Data Fig. 7d,e), supporting direct regulation. Silencing *Esr1* in *Esr1*$^+$/*Grpr*$^+$ melanoma reduced *Grpr* mRNA, whereas *Esr1* overexpression increased it (Fig. 5f,g and Extended Data Fig. 7f,g), confirming that *Esr1* regulates *Grpr*. In Ecad-expressing melanoma, *CDH1* silencing in mouse and human cells raised *ESR1* and *GRPR* mRNA (Fig. 5h and Extended Data Fig. 7h), supporting the inverse correlation between

*CDH1* and *GRPR* seen in TCGA data (Fig. 2j). *ESR1* inhibition decreased *GRPR* mRNA, and dual *CDH1*/*ESR1* inhibition blocked *GRPR* upregulation, indicating that *GRPR* induction on *ECAD* loss is at least partly *ESR1* dependent (Fig. 5h and Extended Data Fig. 7h). At the protein level, CDH1 knockdown or knockout upregulated ERα (Fig. 5i and Extended Data Figs. 6t,u and 7i–k), and *ESR1* silencing increased ECAD, showing reciprocal regulation (Fig. 5i and Extended Data Fig. 7l).

The ERα agonist 17β-oestradiol (E2) increased *Grpr* expression, whereas degrader ICI 182,780 (ICI) reduced it (Extended Data Fig. 7m).

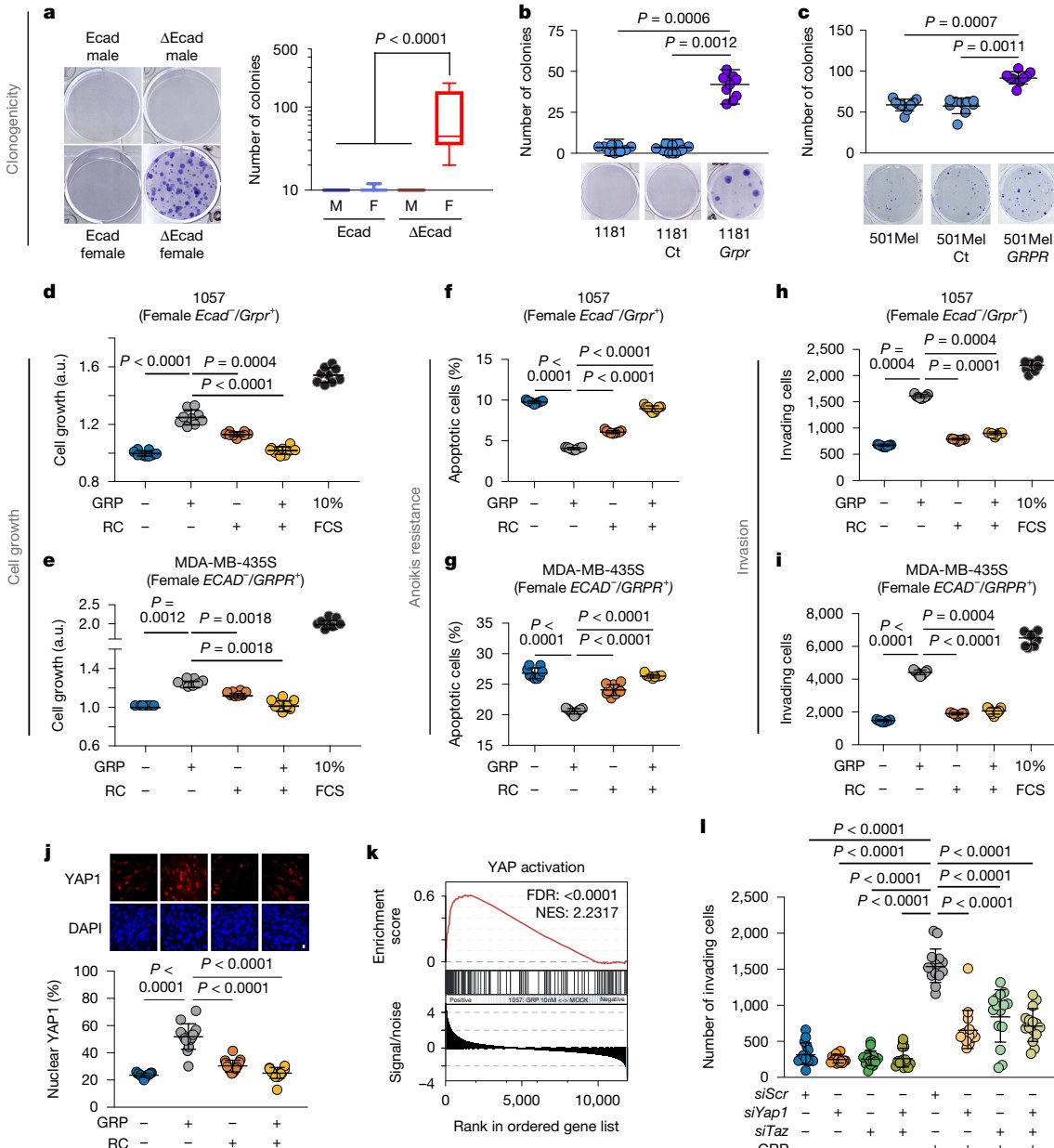

**Fig. 4 | GRPR activates YAP1 to activate the metastatic program. a**, Colony-formation assay done over 10 days, showing colonies from male *Ecad*[+]/*Grpr*[−] and *Ecad*[−]/*Grpr*[+] cells, and female *Ecad*[+]/*Grpr*[−] and *Ecad*[−]/*Grpr*[+] cells. **b,c**, Clonogenic assays for 1181 mouse *Grpr*[−] (**b**) and 501mel human *GRPR*[−] (**c**) melanoma cell lines: parental (left), control (middle) and exogenous *GRPR* expression (right). **d–i**, In vitro assays on mouse 1057 (**d,f,h**) and human MDA-MB-435S (**e,g,i**) *GRPR*[+] cells evaluated for the impact of GRP (10 nM), RC (1 μM) or both under low-serum conditions. GRP promoted cell growth (**d,e**), anoikis resistance (**f,g**) and invasion (**h,i**), effects reversed by RC. GRP + RC effects were compared with vehicle and full-serum controls. Growth and anoikis resistance were assessed after 48 h; invasion at 24 h. NS, not significant. **j**, Representative images and percentage of cells displaying nuclear Yap1 localization in *Grpr*[+] mouse melanoma cells (1057) after a 1-h stimulation with vehicle (lane 1), 10 nM

GRP (lane 2), 1 μM RC (lane 3) or GRP and RC (lane 4). Scale bar, 10 μm. **k**, GSEA of the YAP1 activation signature in *Grpr*[+] 1057 cells. Gene expression was assessed by RNA-seq 4 h after stimulation with 10 nM GRP and normalized using DEseq2 before doing the GSEA. FDR, false discovery rate; NES, normalized enrichment score. **l**, Inhibition of GRP-induced cell invasion after Yap1, Taz or Yap1 + Taz silencing. Each assay was independently repeated at least three times. Multi-group comparisons were done by Kruskal–Wallis tests adjusted by Dunn's correction. Comparisons with the GRP-induced group were done by two-sided Mann–Whitney tests adjusted for multiple comparisons by Benjamini–Hochberg correction. Data are represented as mean ± s.d. and box plots represent the median and the 25–75 percentiles; whiskers represent the minimum and the maximum. At least three independent biological replicates were performed per experiment.

This decrease was linked to lower *Grpr* activity, shown by reduced IP1 production after ICI treatment (Extended Data Fig. 5c–f). Finally, the CDH1–ESR1–GRPR axis operates in a positive-feedback loop: GRPR activation represses *Cdh1*, and E2 reinforces this loop by activating ERα and further inhibiting *Cdh1* (Fig. 5i,j and Extended Data Fig. 7m–o). Thus, ECAD loss initiates a transcriptional program amplified by feedback, explaining the high GRPR expression (Extended Data Fig. 8).

## Fulvestrant inhibits invasion and metastasis

To assess the clinical relevance of oestrogen signalling in *Grpr*-positive melanomas, we examined the impact of oestrogen inhibition using ICI 182,780 (Fulvestrant) both in vitro and in vivo. In vitro, ICI treatment significantly reduced GRP-induced invasion of 1057 ΔEcad melanoma cells, with no effect on *Grpr*-negative lines derived from transgenic mice

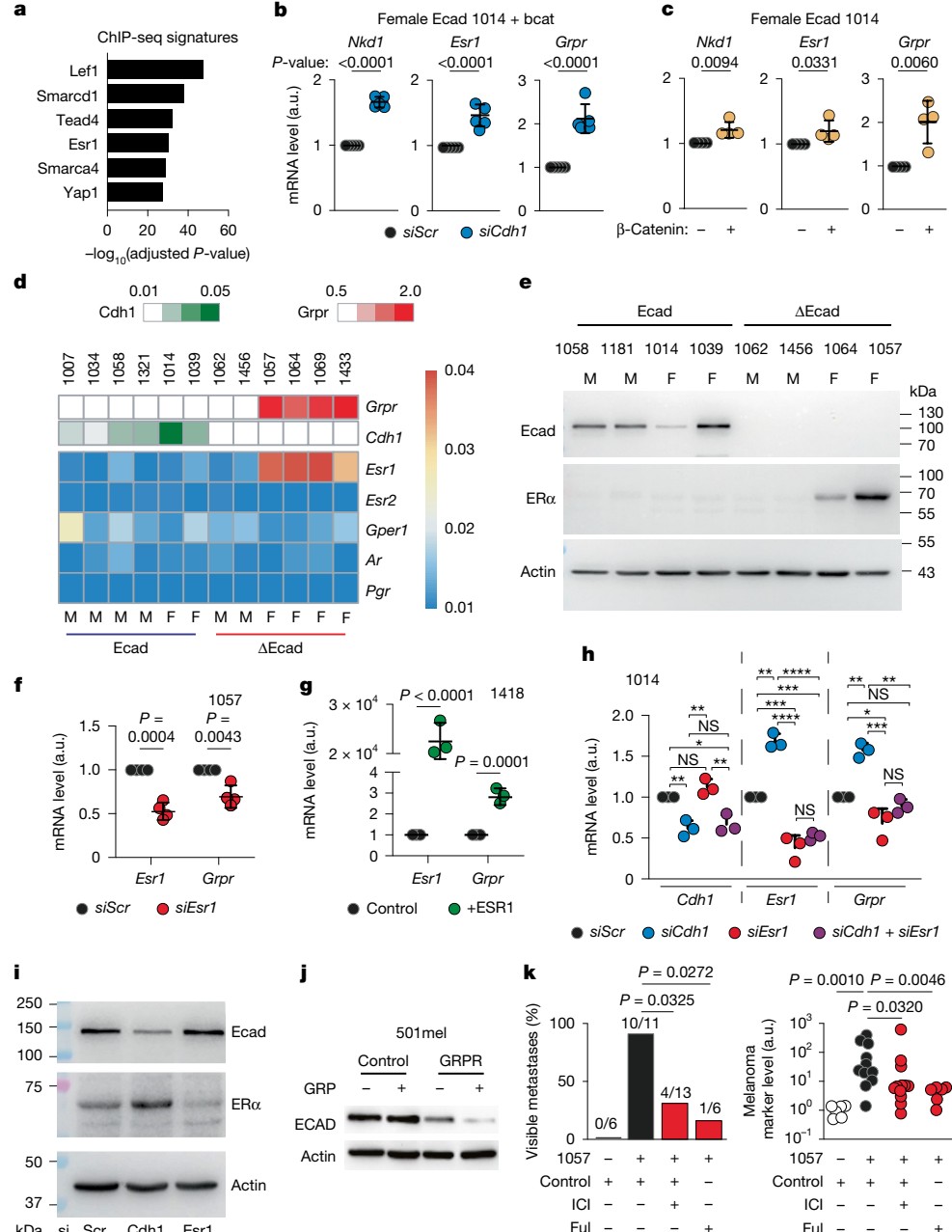

**Fig. 5 | *Grpr* expression in female mice by a *Cdh1*/*Ctnnb1*/*Esr1*/*Grpr* amplification loop. a**, Enrichment of mouse ChIP–seq signatures in ΔEcad female-specific H3K27ac peaks located at gene bodies or promoters. **b,c**, Quantitative PCR with reverse transcription (RT–qPCR) of *Nkd1*, *Esr1* and *Grpr* in Ecad mouse 1014 melanoma cells after siScr versus siCdh1 in the presence of β-catenin (bcat; **b**) and pcDNA3 versus β-catenin transfection (**c**). **d**, Heatmap of sex-hormone receptor expression in Ecad and ΔEcad melanoma cell lines, annotated with Cdh1 and Grpr levels. M, male; F, female **e**, Western blot of E-cadherin and ERα in mouse melanoma cell lines. Actin was used as a loading control. **f,g**, Effect of *Esr1* knockdown (**f**) or overexpression (**g**) on GRPR in mouse and human melanoma cells. **h**, *Grpr* expression after *Cdh1* and/or *Esr1* knockdown in *Cdh1*+ mouse melanoma cells. **i**, Western blot analysis of Ecad

and ERα after siScr, siCdh1 or siESR1 in 1014 cells. Actin was used as a loading control. **j**, Western blot of ECAD in 501mel cells with and without *GRPR* expression and GRP treatment. **k**, Quantification of lung metastases by stereomicroscopy and RT–qPCR for Cre markers in lungs. The log-normalized expression values were compared by two-sided Student's *t*-test (two groups) or by analysis of variance (ANOVA) corrected by Tukeys's test (multiple groups). Metastasis burden was assessed by Fisher's exact test adjusted by the Bonferroni method. Variation of the Cre expression was assessed by two-sided Mann–Whitney tests adjusted for multiple comparisons by a Benjamini–Hochberg test. Data are shown as mean ± s.d. At least three independent biological replicates were performed for each experiment. Ful, Fulvestrant.

(Extended Data Fig. 9). For in vivo analysis, we used two approaches. First, 1057 ΔEcad cells were either pretreated with ICI for three days or left untreated before being injected into the tail vein of female C57BL/6J mice, simulating adjuvant therapy after primary tumour surgery. Second, untreated 1057 cells were injected, followed by Fulvestrant treatment (50 mg per kg) three hours after injection and once a week for three weeks. In both cases, mice were euthanized after 25 days and

lung metastases were analysed (Fig. 5k). In the first approach, 9 of 13 mice injected with ICI-pretreated cells showed no visible metastases, compared with only 1 of 11 in the control group. In the second, 5 of 6 Fulvestrant-treated mice showed no metastases. Metastatic burden, assessed by qPCR for the 1057-specific *Cre* transgene, was reduced in both treatment groups (Fig. 5k). These results demonstrate that oestrogen signalling promotes the invasiveness and metastatic potential

of *Grpr*-positive melanoma cells, and that ICI/Fulvestrant effectively suppresses these processes in vitro and in vivo.

## The CDH1–GRPR axis is active in breast cancer

As shown in Fig. 2j, *GRPR* expression and *CDH1* expression in breast tumours are inversely correlated. Given the oestrogen-dependent growth of many breast tumours[18], we proposed that E-cadherin represses the ESR1–GRPR axis across tissues, including breast cancer. We observed elevated *GRPR* and *ESR1* mRNA levels in breast tumours expressing mutated, non-functional ECAD (Extended Data Fig. 10a,b). Moreover, CDH1-mutant tumours displayed higher ERα activation scores and greater ERα positivity by immunohistochemistry (Extended Data Fig. 10c,d). Several findings supported GRPR's dependence on ERα: the increased GRPR levels in ERα-positive compared with ERα-negative breast tumours (Extended Data Fig. 10e); the strong correlation between *GRPR* mRNA and ERα activation score (Extended Data Fig. 10f); and ERα binding to active regions associated with the GRPR promoter (Extended Data Fig. 10g,h). To validate the role of ERα in regulating GRPR in breast cancer, we did some experiments in MCF7 cells. CDH1 inhibition increased *GRPR* and *ESR1* mRNA, whereas ESR1 inhibition reduced *GRPR* mRNA (Extended Data Fig. 10i). Notably, combined CDH1 and ESR1 silencing failed to induce *GRPR* expression. Activation of ERα by oestradiol increased *GRPR*, whereas ICI 182,780 treatment significantly decreased its expression (Extended Data Fig. 10j).

Together, these findings highlight a conserved role for this regulatory loop, indicating its broader relevance across multiple cancer types and physiological contexts.

## Discussion

Our study uncovers a female-specific metastatic pathway involving E-cadherin. Reduced E-cadherin increases β-catenin transcriptional activity, elevating *ESR1* and inducing *GRPR* transcription, which activates YAP1 and initiates a metastatic cascade. This newly identified CDH1–CTNNB1–ESR1–GRPR–YAP1 axis defines a female-specific tumour metastasis route, providing some potential therapeutic targets (Extended Data Fig. 8). E-cadherin is known to be tumour suppressor in breast and stomach cancers[19,20], in which GRP is abundantly produced and activates GRPR (Extended Data Fig. 2k,l). Thus, GRPR activation probably occurs in primary tumours in these tissues. By contrast, in other carcinomas or melanoma, GRPR is probably activated at metastatic sites by locally produced GRP (Extended Data Fig. 8). Our results indicate that E-cadherin suppresses tumour initiation in GRP-rich primary tissues and acts as to suppress metastasis in contexts in which GRP is restricted to metastatic niches, such as the lung in cutaneous melanoma[21].

The E-cadherin–CTNNB1–ESR1–GRPR loop may be triggered by: E-cadherin alterations through mutation, methylation or epithelial–mesenchymal transition (EMT)[19,22,23]; *WNT*/*BCAT* activation repressing *ECAD* transcription and triggering pathways including ESR1–GRPR[24]; ERα activation through mutation, menstrual cycles, pregnancy or xenobiotics[25,26]; or GRPR activation by GRP, other low-affinity ligands or receptor transactivation[27]. These findings highlight the complex interplay of pathways in cancer progression and emphasize the need to consider them for better cancer management.

We show the sex-dependent role of E-cadherin in repressing ESR1 in human and mouse melanoma and breast cancer cells. Cadherin–catenin interactions, including those involving β-catenin, plakoglobin and p120-catenin, regulate gene expression. Without E-cadherin, β-catenin becomes transcriptionally active and upregulates ESR1. Moreover, E-cadherin loss also alters gene expression through β-catenin-independent mechanisms[28,29]. Other mechanisms might be at play, given the ability of E-cadherin to independently influence gene expression[30]. E-cadherin affects receptor tyrosine kinase signalling

and reshapes chromatin through EMT, influencing *CTCF* expression and chromatin structure[31–34]. Our ChIP–seq data support this, showing that SMARCD1-targeted genes are activated following E-cadherin loss[35]. Given that SMARCD1 interacts with ERα, it probably helps to remodel chromatin at ERα targets such as GRPR, especially in young women[36]. However, understanding chromatin-level effects of E-cadherin and their age and sex dependence needs further study.

A few cancers—mainly of the sex organs—are defined as hormone dependent, but others lack defined hormonal status owing to a lack of research[37]. Our study innovatively links hormone sensitivity to tumour phenotype. In such tumours, sex hormones do not drive proliferation but influence phenotype, metastasis and therapy resistance. This hormone sensitivity probably applies to carcinomas and potentially other cancers, supported by epidemiological data (Supplementary Table 1).

We reveal that loss of E-cadherin activates a metastatic axis involving ERα, GRPR and YAP1, which are therefore potential therapeutic targets. ER modulators and aromatase inhibitors are already used as adjuvants in ER-positive breast cancer[38]. Because ERα regulates *GRPR*, anti-oestrogens might suppress both and improve outcomes. However, aromatase inhibitors can cause joint and muscle pain, possibly affecting survival if treatment is stopped.

The involvement of GRPR in itch and nociception offers potential cancer and pain therapy[39,40]. Despite their relevance, G-protein-coupled receptors remain underused in oncology[12]. We highlight the role of GRPR in cancer progression. The GRPR antagonist RC-3095 reduced melanoma lung metastases and shows promise in breast, lung, prostate and pancreatic cancers[41]. Optimizing pharmacokinetics and reducing toxicity remain key challenges in the development of potent and selective antagonists[42]. It is important to develop selective GRPR antagonists with minimal side effects. Limited adult GRPR expression and mild knockout phenotypes suggest that this approach is safe[43,44]. Structural data from X-ray and cryo-electron microscopy may aid the discovery of better antagonists[45,46]. YAP1 is another target, although it is harder to inhibit and is potentially less safe than GRPR antagonists[47–49].

In summary, our findings underscore the importance of recognizing sex diversity in disease and therapy. This study introduces a sex-specific strategy that could improve cancer treatment in women in whom loss of E-cadherin leads to ERα and GRPR expression.

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

## Methods

### Cancer epidemiology

All the cancer incidence data available from the Global Cancer Observatory's 2020 release were accessed on 24–26 October 2022. Data were collected by sex and age category. The women/men ratios were calculated from the age-standardized rates and logged for representation purposes. A fourth-order polynomial model was fitted for each cancer using prism to represent the women/men ratio dynamic. Each cancer was classified according to a calculated premenopausal variation index (PV) calculated in this way:

$$PV = \mu_{\text{ratio premenopause}} - \frac{(\mu_{\text{ratio prepuberty}} + \mu_{\text{ratio post menopause}})}{2},$$

with $\mu$ being the average of the time period. Three categories were defined according to PV: if PV < −0.2, cancers were defined as male-biased during premenopause; if PV > 0.2, cancers were defined as female-biased during premenopause; otherwise, cancers were defined as unbiased during premenopause. The concordance between PV and the fitted curve was manually checked for each cancer.

### Lists of genes associated with cancers

Lists of genes associated with melanoma (73, hsa05218), gastric cancer (150, hsa05226) and thyroid cancer (37, hsa05216) were retrieved from KEGG. The list of genes associated with breast cancer (172) was from WikiPathway (WP1984). ESR1-associated genes (317) were extracted from ESR1-associated genes from StringDB using the following parameters: search, CDH1 in human; setting, high confidence; and maximum number of interactors, 317. Lists were intersected using a Venn diagram.

### Sex-hormone concentration

Plasma testosterone and oestradiol levels in males and females, categorized by age, were extracted from ref. 50.

### Cancer genomic and transcriptomic data mining

The transcriptome, copy-number alteration, mutations and corresponding clinical data of the TCGA–SKCM ($n$ = 473), TCGA–BRCA ($n$ = 1,215), TCGA–STAD ($n$ = 450), TCGA–LUSC ($n$ = 553) and TCGA–KIRC ($n$ = 606) datasets were retrieved from the National Cancer Institute Genomic Data Commons repository using the TCGAbiolinks R package in August 2022 (ref. 51). The mRNA levels were calculated from RNA-sequencing read counts using RNA-Seq V2 RSEM and normalized to TPM. Expression of *CDH1*, *CCND1*, *BRAF* and *KRAS* was modelled over time using LOWESS smoothing with 1,860 points. Expression levels from acral melanoma were retrieved from GEO under accession number GSE162682. Survival analyses were carried out by separating the cohort into two groups according to gene expression. The threshold was set to 1 TPM, commonly considered to be the limit for sufficient protein expression of the transcript. The negative group was set to an expression 0.1 TPM or less to have clear separation in terms of expression (factor 10) from the positive group. The YAP and melanoma phenotypic state scores were obtained by averaging the expression of detailed marker genes, or for the mice, from their murine orthologues as described in ref. 52. The pigmentation state was defined by the expression of MITF, MLANA, TRPM1, DCT and TYR; the starved-like melanoma cell phenotype by the expression of CD36, DLX5, IP6K3, PAX3 and TRIM67; the invasive state by the expression of AXL, CYR61, TCF4, LOXL2, TNC and WNT5A; and the neural crest cell-like phenotypic state by the expression of AQP1, GFRA2, L1CAM, NGFR, SLC22A17 and TMEM176B[52]. The scoring of YAP1 activation was determined from the expression of CYR61, CTGF, TEAD4, LATS2 and CRIM1. Yap scoring was obtained by averaging the fold change of each Yap1 target. The anoikis-resistance score was calculated on the basis of the expression of S100A7A, MTPN, ATP10B, S100A8, RSAD2, RENBP, CDHR1 and CD36.

The ER-activation score was calculated according to the expressions of GATA4, SDK2, EGR3, IL19, GSG1L, RSPO1, PGR, IL24 and PDZK1. Expression data from human and mouse normal tissue were downloaded from the Human Protein Atlas (https://www.proteinatlas.org/) and from the EBI expression atlas (https://www.ebi.ac.uk/gxa/home), respectively, both accessed on 7 October 2022. The anatogram was generated using the gganatogram R package[53].

### Mice

Animal care, use and all experimental procedures were conducted in accordance with recommendations of the European Community (86/609/EEC) and European Union (2010/63/UE) and the French National Committee (87/848). Mice were housed in a specific-pathogen-free (SPF)-certified animal facility with a 12 h:12 h light:dark cycle in a temperature- and humidity-controlled room (22 ± 1 °C and 60%, respectively) with free access to water and food. Animal care and use were approved by the ethics committee of the Curie Institute in compliance with institutional guidelines. Experimental procedures were carried out under the approval of the ethics committee of the Institut Curie CEEA-IC #118 (CEEA-IC 2016-001) in compliance with international guidelines. The transgenic *Tyr::CreA* (B6.Cg-Tg(Tyr-cre)1Lru/J), named Tyr::Cre, Tg(Tyr-NRAS*Q61K)1Bee, Cdkn2a^tm1Rdp, B6.129-Cdh1tm2Kem/J, named Cdh1F/F, mice, also including *Tyr::CreB*, *bcat** (Tg(Tyr-Ctnnb1/EGFP)#Lru), *Pten* (Pten^tm1Hwu) have been described and characterized previously in the Larue laboratory and elsewhere[11,54–58]. The mouse lines were backcrossed onto a C57BL/6J background for more than ten generations. Genotyping was done according to ref. 59 using specific primers and conditions (Supplementary Tables 9 and 10). Mice were crossed to obtain the desired genotypes. Mice were born with the expected ratio of Mendelian inheritance and no changes in gender ratios were observed. Mice were checked weekly for the appearance of new tumours. Tumour volume ($V$) was calculated using the formula $V = (L \times W^2)/2$, where $L$ is the longest dimension of the tumour and $W$ is the width perpendicular to $L$. Tumours were allowed to grow until reaching a volume of approximately 1 cm³. To comply with ethical guidelines, the total tumour burden per mouse was limited to 1 cm³. Mice were euthanized on reaching any predefined ethical end point. At the end of the experiments, all mice underwent autopsy to assess the presence of metastases in distant organs.

### Injections, in vivo imaging and metastasis detection

Mouse and human melanoma cells ($5 \times 10^5$) were suspended in 200 µl PBS and injected into the tail veins of eight-week-old C57BL/6J and NSG mice, respectively. Mice were monitored daily and weighed twice weekly. Euthanasia and autopsy were performed following 20% weight loss or reaching ethical end points.

For imaging, C57BL/6J mice were shaved three days before injection and biweekly. Mice received 300 µg Xenolight D-luciferin (Revvity) intraperitoneally, followed by isoflurane anaesthesia. After 10 min, luminescence was recorded using an IVIS Spectrum system (Perkin-Elmer) with adjustable shutters focused on the thoracic region. Whole-body and thoracic luminescence were acquired for 2 min on day 0, day 1 and twice weekly. Mice were randomized after injection based on body weight and day 0 lung luminescence. Treatments (10 µg RC/DMSO or PBS/DMSO) were administered twice daily for one month, with luminescence monitoring. After euthanasia, lungs were processed as follows: right lung, incubated in 300 µg ml⁻¹ Xenolight D-luciferin (2 min) and imaged (1 min); left lung, fixed in 4% PFA (4 °C, 24 h), cryoprotected in 30% sucrose and 30% sucrose/50% OCT (48 h each), embedded in OCT, sectioned (7 µm) and stained for *Dct* and *Grpr* mRNA (RNAscope, Bio-Techne; probes, Mm-Dct-C1 460461, Mm-Grpr-C3 317871-C3). Imaging was done using a Leica SP8 confocal microscope.

**Metastasis quantification.** Right lungs were imaged using a Leica MZFLIII binocular microscope with a Scion camera. Metastases were

quantified in ImageJ, with macro-metastases (bigger than 0.1 mm) and micro-metastases (0.1 mm or smaller).

**ICI/Fulvestrant experiments.** In vitro pre-treatment: 1057 melanoma cells ($5 \times 10^5$) were pretreated for 3 days with 1 µM ICI before tail-vein injection. In vivo treatment: 1057 melanoma cells ($5 \times 10^5$) were injected into C57BL/6J mice, followed by 50 mg Fulvestrant (Zentiva) 3 h after injection, then weekly for 3 weeks. Mice were euthanized on day 25 after injection, and lung metastases were assessed by PCR-based *Cre* transgene detection. DNA was extracted from lung tissue and metastasis burden was analysed statistically.

### RNA extraction and transcriptomic analysis
RNA was extracted from cells and mouse tumours using the miRN-easy kit (Qiagen, 217004), according to the manufacturer's protocol. RNA integrity (RIN) was measured using an Agilent Bioanalyser 2100 (Agilent Technologies) and an RNA nano 6000 kit (5067-1511, Agilent Technologies). Only RNA with an RNA integrity number (RIN) of more than 7 was used for analysis. This threshold led to the sequencing of 72 mouse melanoma cell lines, 36 human melanoma cell lines and 32 mouse tumours. RNA concentrations were measured using a NanoDrop (NanoDrop Technologies). RNA-sequencing libraries were prepared from 1 µg total RNA using an Illumina TruSeq Stranded mRNA library preparation kit, which allows strand-specific sequencing. PolyA selection using magnetic beads was done to focus the sequencing on polyadenylated transcripts. After fragmentation, cDNA synthesis was done and the resulting fragments used for dA-tailing, followed by ligation with TruSeq indexed adapters. The fragments were amplified by PCR to generate the final barcoded cDNA libraries (12 amplification cycles). The libraries were equimolarly pooled and subjected to qPCR quantification using the KAPA library quantification kit (Roche). Sequencing was carried out on a NovaSeq 6000 instrument (Illumina) based on a $2 \times 100$ cycle mode (paired-end reads, 100 bases) using an S1 flow cell to obtain approximately 35 million clusters (70 million raw paired-end reads) per sample. Reads were mapped to the mm10 mouse reference genome (gencode m13 version-GRCm38.p5) or hg38 human reference genome (gencode 42 version-GRCh38.p13) using STAR[60]. STAR was also used to create the expression matrices. When applicable, expression was batch-corrected with Combat using the sva package from R[61].

Differential gene-expression analysis was done using R following the DEseq2 pipeline with the DEseq2 package[62]. DEseq2 and edgeR (ref. 63) (to retain only the expressed genes) algorithms were used. The packages are both available from Bioconductor (http://www.bioconductor.org) (accessed in October 2022). The threshold for significantly differentially expressed genes was set as an absolute $\log_2$-fold change greater than 1. The volcano and correlation plots depicting the results were generated using the R package ggplot2 v.3.5.1 (ref. 64). GSEA was done using previously published signatures, described in supplementary Table 8, and expression was obtained after DEseq2 normalization. GSEA parameters were set to 1,000 permutations per gene set. Only gene sets with a normalized enrichment score of more than 1.7 and a false discovery rate of less than 0.05 were considered.

### RNA quantification by RT–qPCR
RNA (3 µg) was reverse transcribed using M-MLV reverse transcriptase (Invitrogen), according to the manufacturer's protocol. The newly synthesized cDNA was used as a template for qPCR with the iTaq Universal SYBR Green Supermix. Technical triplicates were used for each sample and the quantified RNA normalized against *TBP* (human) or *Hprt* (mouse) as housekeeping transcripts (Supplementary Tables 9 and 10).

### Human samples
The retrospective study on lung human melanoma metastases was approved by the ethics committee. The non-opposition or consent (before or after 2004, respectively) of patients for the use of their biological material and data was obtained according to the bioethics law of 2004. We retrieved 43 tissue samples of non-treated lung melanoma metastases registered from 1999 to 2014 from the pathology files of the Bordeaux and Rennes hospital. We selected all available formalin-fixed paraffin-embedded surgical specimens of lung melanoma metastases for further immunostaining analysis.

### Immunohistochemistry
Paraffin was melted at 56 °C overnight. Deparaffinization, using a Bond Dewax Solution (CAR9222, Leica), and rehydration were done with a Leica BONDTM-MAX device. Heat-induced epitope antigen retrieval was performed at 100 °C for 20 min in Bond Epitope Retrieval Solution 1 (AR9961, Leica) for GRPR or Bond Epitope Retrieval Solution 2 (1/100, AR9640, Leica) for E-cadherin. Slides were incubated in anti-GRPR antibody (SP4337P, Acris Antibodies) in Bond Primary Antibody Diluent (1/100, AR9352, Leica) overnight at 4 °C and anti-E-cadherin (1/100, NCL-L-E-Cad, Novocastra) antibody in the same diluent for 30 min at room temperature. Bond Polymer Refine Red Detection (DS9390, Leica) was used, according to the manufacturer's specifications. Slides were counterstained with haematoxylin and cover-slipped. Images were acquired using an Axio Imager Z2 microscope. Each immunostaining was evaluated in a double-blind manner by two pathologists.

### Cell lines
Mouse melanoma cell lines were established from melanomas arising in transgenic mice in the laboratory as previously described[65]. The cell lines mutational landscape was determined by whole-exome sequencing. MDA-MB-435S, 624mel (often referred to as 501mel in the literature; 624mel cells are male), 888-Mel and Dauv-1 human melanoma cell lines were previously established in other laboratories[66–68]. The human breast cancer cell line MCF-7 was previously established[69]. The pGK-Luc2 vector was a gift from Catherine Tomasetto (IGBMC). In brief, the coding sequence of the luciferase reporter gene *luc2* (from *Photinus pyralis*) was amplified by PCR from the pGL4.50[luc2/CMV/Hygro] vector (Promega, E1310) and flanking XhoI restriction sites were added. The digested PCR fragment was subcloned into the SalI site of the pLENTI PGK Blast DEST vector (plasmid 19065, Addgen). 1057-luciferase melanoma cell lines were generated by infecting parental 1057 cells with pGK-Luc2 (LL#1231). Cells were selected using 4.5 µg ml⁻¹ blasticidin for one week. Cell lines 1014 and 1181 Grpr and the corresponding controls were obtained after transfection of the murine Grpr/tGFP plasmid (1045, MG224721, Origene) and tGFP plasmid (1064, pCMV6-AC-GFP, Origene), respectively. The pSpCas9(BB)−2A-GFP (PX458) was a gift from Medhi Khaled (Institut Gustave Roussy). The annealed oligos corresponding to the gRNA sequences were ligated within the BsmbI (R0580, New England Biolabs) digested plasmid using Quick Ligase (M2200S, New England Biolabs). The oligos are listed in Supplementary Table 7. Mouse melanoma cells were transfected with either the plasmid targeting the 5′ side of the second exon or the plasmids targeting the 5′ and the 3′ side of exon 2 using lipofectamine 2000 (11668019, Invitrogen). Two days later, GFP-positive cells were sorted and one cell per well was seeded in 96-well plates. Cells were cultured until first passage when half of the cells were collected for DNA extraction. The status of *Grpr* was assessed by PCR. Transfection of β-catenin was performed using CMV::bcat* (777) or mock control (empty pcDNA3, 297). Cells were transfected with 2−4 µg of plasmid or 100 pmol of siRNA and lipofectamine 2000, following the manufacturer's protocol. Transfected 1014 and 1181 cells were selected using 25 or 150 µg ml⁻¹ geneticin, respectively. Then, 501mel-GRPR and 501mel-Ct were generated by infecting cells with the pLV-Hygro-CMV-Grpr-EGFP (LL1272, 1VB191126-1286xxe, Vector Builder) or pLV-Hygro-CMV-EGFP (LL1271b, VB191126-1289cfv, Vector Builder) plasmids, respectively. For siRNA knockdown, cells were transfected with 100 pmol siRNA (Supplementary Table 11) using Lipofectamine 2000 (Invitrogen) following the manufacturer's instructions. Inducible piSMART shRNA plasmids (V3SM11253-231787949 for the

shGrpr and VSC11655 for the non-targeting shRNA, Dharmacon) were infected in 1057 melanoma cells, and positive cells were selected using 1.2 µg ml⁻¹ puromycin. Before experiments, cells were treated for three days with doxycyclin or mock. For CRISPR, the control was generated using 1.2 µg scramble gRNA + 6 µg Cas9-RFP and 1.2 µg 3′ gRNA + 6 µg Cas9-GFP, and CDH1 was generated using 1.2 µg 5′ gRNA + 6 µg Cas9-RFP and 1.2 µg 3′ gRNA + 6 µg Cas9-GFP. The gRNA and Cas9 constructs (ALT-R optimization) were obtained from Integrated DNA Technologies. Both RNP complexes were transfected into 888-Mel melanoma cells using lipofectamine CRISPRMAX (cmax00008, Invitrogen). GFP⁺/RFP⁺ cells were sorted the next day and cultured in phenol red-free media with 100 nM E2 for four days. The gRNA sequences are shown in Supplementary Table 9.

Murine and human melanoma cell lines were grown in Ham's F12 medium and RPMI 1640, respectively, supplemented with 10% FCS (10270106, Gibco) and 1% PS. The breast cancer cell line MCF7 was grown in DMEM-F12 supplemented with 10% FCS and 1% PS. All cell lines were maintained at 37 °C in a humidified atmosphere with 5% $CO_2$. Cells cultured without phenol red were supplemented with 2 nM glutamine. The genetic status and level of expression of key genes of these cell lines are presented in Supplementary Table 7.

## Whole-exome sequencing
We used 2 million mouse melanoma cells. The DNA was extracted using the DNeasy Blood and Tissues kit (69504, Qiagen). Library preparation was done using the SureSelect XT Mouse All Exon Kit (Agilent) followed by high-throughput sequencing on an Illumina NovaSeq 6000 instrument (Illumina). Analyses were performed with the European galaxy instance. We aligned the fastq to the mouse mm10 genome using BWA. Duplicates were removed using the MarkDuplicates (v.3.1.1.0) function from Picard. Tracks were visualized using IGV.

## Cell growth and clonogenic assay
Six-well tissue-culture plates were seeded with $3 \times 10^5$ melanoma cells in complete medium. After 24 h, the medium was replaced by low-serum medium (0.5% for murine melanoma cell lines and 1% for human melanoma cells) and the cells were incubated for a further 18 h before stimulation with 10 nM GRP (4011670 bachem) and/or 1 µM RC (R9653, Sigma-Aldrich) for 48 h. The plates were trypsinized just after stimulation or 48 h later and the cells were counted. For the MTT assay, 10,000 cells were seeded per well in 96-well plates. After 24 h, cells were starved overnight and then treated with 10 nM GRP and/or 1 µM RC for 48 h in low-serum medium (0.5% FBS). Next, MTT (M5655, Sigma-Aldrich) was added to the wells to a final concentration of 0.5 mg ml⁻¹ and the plates were incubated for 3 h. The medium was removed and formazan crystals dissolved in 200 µl DMSO. The absorbance was read at 570 nM using a LUMIstar Omega luminometer (BMG Labtech). All growth experiments were done using three technical replicates and three biological replicates. For the clonogenic assay, six-well tissue-culture plates were seeded with 500 cells in complete medium. After 10 days (20 days for 1181), colonies were fixed with 4% PFA for 15 min and stained with 10% crystal violet in ethanol for 20 min and counted in images using ImageJ software. Experiments were done in triplicate.

## Anoikis assay
Six-well plates were coated with poly-HEMA to prevent cell attachment to the well surface. Cells were seeded in low-serum medium containing 10 nM GRP and/or 1 µM RC. Then, 48 hours after cell seeding, cells were collected and washed with ice-cold PBS twice and resuspended in annexin V binding buffer (556454, BD Biosciences) and incubated at room temperature in the dark with 7-amino-actinomycin D (7-AAD, 559925, BD Biosciences) and annexin-V for 15 min. Annexin V was coupled to FITC (556420, BD Biosciences) for the 1057 and MDA-MB-435S cell lines and to PE (556421, BD Biosciences) for the other cell lines. Cells were sorted using a FACS LSRFortessa (BD Biosciences) to determine the percentage of annexin V- and/or 7-AAD-positive cells using a 488 nm laser for annexin V-FITC and annexin V-PE and a 675 nm laser for 7-AAD. All quantification was performed using Flow-Jo.

## Matrigel invasion assay
Matrigel invasion assays were done in transwell plates with 8.0-µm pores (353097, Falcon) coated with 100 µl of 200 µg ml⁻¹ Matrigel. Depending on the experiments, cells were seeded directly or 24 h after transfection in low-serum medium (0.5% FCS for murine cells and 1% for human cell lines) with 10 nM GRP, 1 µM RC-3095, and/or 1 µM IC 182,780, or in 10% FCS. Then, 24 hours after stimulation, inserts were washed with PBS and non-invading cells were removed. Cells in the inserts were fixed in methanol at −20 °C overnight. The inserts were rinsed and the membrane carefully removed using a sharp scalpel. The membrane was mounted in prolong glass DAPI (1.5 µg ml⁻¹). Assays were performed in triplicate and automatic counting of invading cells was done using an Image/Fiji macro (https://doi.org/10.5281/zenodo.14509394). Sequences of siRNAs are given in Supplementary Table 11.

## Western blot analysis
Whole-cell lysates were prepared and analysed as described previously[65]. The primary antibodies used were: E-cadherin (610182, BD Transduction Laboratories, dilution 1/1,000); ERα (MA1-27107, Invitrogen, dilution 1/500); YAP (14074, Cell Signaling, dilution 1/1,000); β-actin (A5441, Sigma, dilution 1/5,000); and vinculin (4650, Cell Signaling, dilution 1/5,000). All raw data are provided in Supplementary Fig. 1.

## Kinase assay
Serine/threonine and tyrosine kinase activity were determined using STK PamChips (87102 PamGene International). All assays were done according to the manufacturer's protocol[70]. In brief, cells were seeded to 70% confluence and starved overnight the day after. Then, the medium was replaced by medium containing vehicle (0.1% DMSO) or 10 nM GRP and/or 1 µM RC and the cells incubated for 15 min. Cells were rinsed twice in PBS and then lysed in M-Per buffer (78503, ThermoFisher Scientific) containing Halt Phosphatase Inhibitor (78428, ThermoFisher Scientific) and Halt Protease Inhibitor (78437, ThermoFisher Scientific), both diluted 1:100. The lysates were immediately snap-frozen. Kinase activity was determined using Pamstation PS12 and 1 µg protein for the STK chips, according to the manufacturer's protocol. The data were analysed using BioNavigator software (PamGene International), batch corrected using ComBat[71] and normalized using VSN[72]. Kinase activity was assessed using the 2018 version of the UKA tool using basic parameters (Scan rank from 4 to 12, 500 permutations, 90% homology, equivalent weight for each database, minimal prediction score of 300). The UKA tool infers the active kinase from the differentially phosphorylated peptides using databases and predicted interactions (PhosphoNet database).

## IP1 detection
IP1 levels were quantified using the HTRF IP-one Gq detection kit (62IPA-PEB, Revvity) following the manufacturer's protocol. We treated 4,000 cells for 30 min at 37 °C in StimB buffer with 10 nM GRP, 1 µM RC-3095, 1 µM ICI-182,780 or DMSO. FRET signals (620 nm and 665 nm, excitation at 385 nm) were measured using a Clariostar plate reader (BMG Labtech).

## Immunofluorescence
Cells were seeded on coverslips and cultivated until 100% confluence was achieved. After 24 h of starvation in 0.5% FCS for murine melanoma cell lines and 1% FCS for human melanoma, the medium was complemented with 10 nM GRP and/or 1 µM RC for 1 h. Cells were fixed with 4% PFA for 15 min and blocked with 5% normal goat serum and 0.3% Triton X-100 in PBS. Cells were incubated overnight at 4 °C with anti-YAP D8H1X antibody (14074, cell Signaling, 1/100 dilution) in 1%

BSA and 0.3% Triton X-100 followed by incubation for 1.5 h at room temperature with goat anti-rabbit Alexa fluor 594 (A-11012, Invitrogen, 1/500 dilution). Coverslips were mounted with Prolong Gold containing 1.5 µg ml$^{-1}$ DAPI (P36934, Invitrogen). Images were acquired using an inverted SP8 Leica confocal microscope (Leica Microsystem). YAP1 localization was evaluated using ImageJ Software.

## Plasma stability assay

Each compound was diluted in mouse plasma to a final concentration of 1 µM and incubated at 37 °C for 2 h. The reaction was stopped by the addition of 2.5 volumes of ice-cold acetonitrile. Liquid chromatography–mass spectrometry was performed on the supernatant in multiple reaction-monitoring mode (LC–MS/MS). The percentage of the remaining test compound relative to that present at the start was determined by monitoring the peak area.

## Metabolic stability assay

Each compound was diluted to a final concentration of 1 µM in 100 mM phosphate buffer (pH 7.4) containing 0.5 mg ml$^{-1}$ mouse liver microsomes, 1 mM NADPH regenerating system and 1 mM MgCl$_2$ and incubated at 37 °C for 1 h. At various time points (0, 2, 10, 20, 40 and 60 min), one volume of ice-cold acetonitrile was added and the supernatants were analysed by LC–MS/MS. To obtain a stability curve, the percentage of remaining test compound at each time point was determined by monitoring the peak area. The half-life was estimated from the slope of the initial linear range of the logarithmic curve of the remaining compound (percentage) against time, assuming first-order kinetics.

## LC–MS/MS

Analyses were done on a Shimadzu 8030 LC–MS instrument. Chromatographic separations were done at 40 °C using a 2.6-µm C18 Kinetex column (50 × 2.1 mm; Phenomenex). The mobile phase flow rate was set to 0.5 ml min$^{-1}$ and the following program applied for the elution: 0 min, 5% B; 0–1.2 min, 5–95% B; 1.2–1.4 min, 95% B; 1.4–1.42 min, 95-5% B; and 1.42–2.8 min, 5% B (solvent A, 0.05% formic acid in water; solvent B, acetonitrile). The injection volume was 1 µl. The mass spectrometer was interfaced with the liquid-chromatography system using an electrospray ion source. The nitrogen nebulizing gas flow was set to 1.5 l min$^{-1}$ and the drying gas flow to 15 ml min$^{-1}$. The interface voltage was set to 4,500 V. The temperature of the block heater was maintained at 400 °C and the desolvation line at 250 °C. Argon was used as the collision gas at 230 kPa. The transitions in positive mode were $m/z$ 585.1 → 204.0, 221.0 for PD176252, and $m/z$ 369.5 → 144.1, 110.1 for RC.

## ChIP–seq

Mouse melanoma cell lines were grown in normal medium until reaching 60 × 10$^6$ cells at 70% confluency. Cells were fixed in 10 ml of 0.4% PFA for 10 min at room temperature. Crosslinking reactions were stopped by adding 1 ml of 2 M glycine (pH 8.0). Cells were resuspended in 1 ml PBS and sonicated until DNA was fragmented to an average of 400 bp. For H3K27ac ChIP–seq in mouse cell lines, 100 µg chromatin was then incubated for 4 hours at 4 °C with 40 µl protein G sepharose beads (17-0618-02, VWR Chemicals) blocked with tRNA from yeast (AM7119, Invitrogen) and BSA in ChIP dilution buffer. Supernatant was pipetted to a new tube and incubated overnight on a rotating wheel at 4 °C with 10 µg anti-H3K27ac antibody. The chromatin was incubated for 1 hour at 4 °C on a rotating wheel with 20 µl blocked protein G sepharose beads. The supernatant was removed and beads were washed with the ChIP low-salt buffer, twice with the ChIP high-salt buffer, twice with LiCl buffer and twice with TE buffer. Chromatin was eluted by incubating the beads twice with 250 µl elution buffer for 15 min on a rocking plate at room temperature. Then, 20 µl of 5 M NaCl was added to the chromatin and the mix was incubated at 65 °C overnight to reverse crosslinks. Proteins were degraded by incubating the chromatin with 10 µl TrisHCl (pH 6.8), 10 µl of 0.5 M EDTA and 20 µg proteinase K for 1 hour at 42 °C. DNA was extracted adding 500 µl phenol:chloroform (1:1, pH 7–8) followed by extraction of the supernatant with 500 µl chloroform. The supernatant was removed and DNA was precipitated by adding 50 µl of 3 M sodium acetate, 15 µg Glycoblue and 1 ml ethanol, and incubated at 4 °C for 4 hours. Pellets were washed with ethanol (70%), dried and finally resuspended in 25 µl water. ChIP-buffer compositions are detailed in Supplementary Table 12. Libraries were prepared from input and immunoprecipitated DNA using the Illumina TruSeq ChIP library preparation kit, according to the manufacturer's protocol. In brief, 2–3 ng of DNA was subjected to subsequent steps of end-repair, dA-tailing and ligation of TruSeq RNA UD index Illumina adapters. After a final PCR amplification step (15 cycles), the 13 resulting bar-coded libraries were equimolarly pooled and quantified using a qPCR method (KAPA library quantification kit, Roche). Sequencing was done on a NovaSeq 6000 instrument from Illumina using paired-end 2 × 100 bp, to obtain around 60 million clusters (120 million raw paired-end reads) per sample. Raw files were uploaded on the European instance of galaxy (usegalaxy.eu) for further processing[73]. Reads were aligned to the GRCm38 mm10 reference genome using bowtie2 (ref. 74). The same analysis process was used for the following raw ChIP–seq files from the literature, downloaded from GEO: ERα in mouse uterus (GSM894054; ref. 75); β-catenin in nephron progenitor cells (GSM980186; ref. 76); H3K27ac in MCF7 (GSM2175784; ref. 77); ATAC-seq (GSM2645717; ref. 78); ERα in MCF7 (GSM798434; ref. 79); and β-catenin in hESC (GSM1579346; ref. 80). For human ChIP–seq, reads were aligned against the human GRCh38 hg38 genome. The 3D chromatin interactions were downloaded from GSE207828 (ref. 81) for mouse and from ENCODE[82] (www.encodeproject.org, accession number ENCFF804SET) and ref. 83 for human. All ChIP–seq alignments and genome interactions were visualized using IGV. ChIP–seq data were annotated using chipseeker[84].

## Pharmacological targeting of ERα

For hormone depletion, FCS was stripped using the dextran-coated charcoal method. In brief, Norit activated charcoal (C6241-5G, Sigma-Aldrich, final concentration of 0.25%) and Dextran T-70 (31390, Sigma-Aldrich, final concentration of 0.0025%) in 0.25 M sucrose/1.5 mM MgCl$_2$/10 mM HEPES (H4034, Sigma-Aldrich), pH 7.4, were incubated overnight at 4 °C. The volume equivalent to the volume of the FCS to strip was pipetted into a new 50 ml tube and centrifugated at 500g for 10 min to remove the supernatant. The FCS was incubated with the activated charcoal for 12 h at 4 °C. The treated FCS was then filtered through a 0.22-µM pore filter to ensure sterility and mixed with the appropriate phenol red-free medium and PS. Cells were collected using phenol red-free trypsin and starved for oestrogen using phenol red-free/10% stripped FCS for four days. Cells were then stimulated with β-oestradiol (2824, Tocris Bioscience) or ICI 182,780 (1047, Tocris Bioscience) for 72 h in phenol red-free/5% stripped FBS.

## Statistical analysis

Cell culture-based experiments were performed in at least biological triplicates and validated three times as technical triplicates. The significance of the effects was calculated using the Mann–Whitney test or Student's $t$-test for the comparison of two groups for non-parametric or parametric situations, respectively. When more than two groups were compared, Kruskal–Wallis or ANOVA was used, according to the parametricity of the data. After each Kruskal–Wallis or ANOVA test, results were adjusted for multiple comparisons using the Dunn and Tukey adjustments, respectively. Categorical data were compared using Fisher's exact test when two groups were compared or, otherwise, a Chi-squared test. The significance of the difference between Kaplan–Meier curves was calculated using a log-rank test. Data are represented as mean ± s.d. unless otherwise indicated in the figure legend. All $P$-values are reported as computed by Prism 10. $P < 0.05$ was considered significant; NS, not significant; *$P < 0.05$, **$P < 0.01$, ***$P < 0.001$, ****$P < 0.0001$.

## Reporting summary

Further information on research design is available in the Nature Portfolio Reporting Summary linked to this article.

## Data availability

All sequencing data generated with this manuscript were deposited on the relevant platform. RNA-seq and ChIP–seq data were deposited on GEO at the National Center for Biotechnology Information under the SuperSeries GSE218588. That includes: mouse tumour RNA-seq with accession number GSE218532; mouse cell line RNA-seq with accession number GSE218586; and human cell line data with accession number GSE218530. The ChIP–seq data are available with accession number GSE237500. Whole-exome sequencing of the mouse melanoma cell lines are available from SRA at bioproject PRJNA904253 (https://dataview.ncbi.nlm.nih.gov/object/40767225). Kinase-assay raw data were deposited on Mendeley at https://doi.org/10.17632/nwkpyr2nmh.1. The following datasets from the literature were used: ChIP–seq of ERα in mouse (GSM894054) and human (GSM798434), of β-catenin in mouse (GSM980186) and human (GSM1579346) and H3K27Ac in human MCF7 (GSM2175784). ATAC-seq data from the MCF7 cell line originate from GSM2645717. The 3D chromatin interactions were downloaded from GSE207828 for mouse and from ENCODE ENCFF804SET (https://www.encodeproject.org/experiments/ENCSR549MGQ/) and GSE52457 for human. Expression data from acral melanoma are from GSE190113. TCGA datasets were accessed through TCGABiolinks or CBioPortal (https://www.cbioportal.org/). All data have been aligned to the human reference genome GRCh38 (hg38 gencode 42 version GRCh38.p13) and the mouse reference genome mm10 gencode 13 version GRCm38.p5. All other data are available from the corresponding author upon reasonable request. Source data are provided with this paper.

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

**Acknowledgements** We thank D. Metzger for providing ESR1 expression vector and comments; A. Bellacosa, D. Duteil, M. Dutertre, C. Goding, S. Marullo and E. Steingrimsson for conversations; F. Beermann, R. Kemler and M. Seano for providing the Tg(Tyr-NRAS*Q61K)1Bee, B6.129-Cdh1tm2Kem/J and Cdkn2a^{tm1Rdp} mouse lines, respectively; J. Groten, M. Bel, E. Van Breemen, S. Rangarajan and J. Lebens for PamGene Technology; the staff at the Institut Curie responsible for the animal colony (especially P. Dubreuil and C. Lantoine), histology (S. Leboucher), FACS (C. Lasgi) and PICT-IBiSA imaging (C. Messaoudi and L. Besse); and D. Delmas, J.-C. Delmas, M. Larue and R. Larue for their constant support. This work was supported by FRM EQU202103012599 and the Institut National Du Cancer (INCa; 2023-1-PL BIO-03-ICR-1), Institut Curie, INSERM and CNRS. J.H.R. had a fellowship from PSL and LNCC. M.W. had a fellowship from DIM SEnT. F.L. had a fellowship from MENRT, SFD and LNCC. High-throughput sequencing was done by the ICGex NGS platform of the Institut Curie, supported by grants ANR-10-EQPX-03 (Equipex) and ANR-10-INBS-09-08 (France Génomique Consortium) from the Agence Nationale de la Recherche (Investissements d'Avenir programme), by the ITMO-Cancer Aviesan (Plan Cancer III) and by the SiRIC-Curie programme (SiRIC grant INCa-DGOS-465 and INCa-DGOS Inserm_12554). Data management, quality control and primary analysis were done by the Bioinformatics platform of the Institut Curie.

**Author contributions** L.L. and V.D. conceived the work, designed the experimental set-up, did data analysis and supervised the study. J.H.R., Z.A., V.P., I.M., I.D. and V.D. did biochemical and cell-biological experiments and data analysis. J.H.R., M.P., F.L., M.W., P.G., C.B., D.D., F.N., B.V. and V.D. did in vivo experiments and data analysis. J.H.R. and V.P. did in silico analysis. C.-L.T., R.M.W. and F.M.-B. provided critical resources. J.H.R. and L.L. produced the graphical representation of the data. L.L. and V.D. acquired funding to support the project. The manuscript was written by J.H.R., L.L. and V.D. The manuscript was reviewed by Z.A., V.P., I.D. and R.M.W. All authors approved and contributed to the final manuscript.

**Competing interests** The authors declare no competing interests.

**Additional information**
**Correspondence and requests for materials** should be addressed to Lionel Larue.

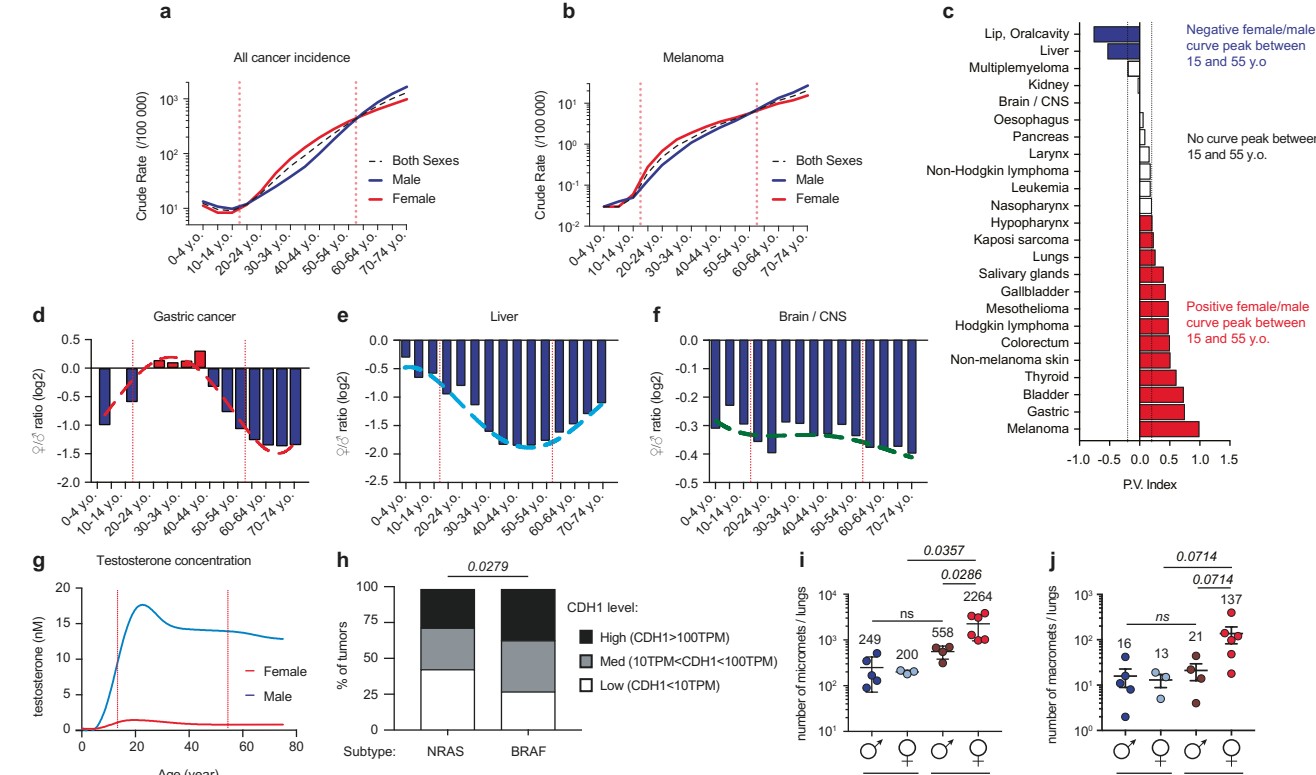

**Extended Data Fig. 1 | Between puberty and the menopause, the incidence of cancer is higher in women than in men. a,b** Crude incidence rates (cases/100,000) for all cancers (**a**) and cutaneous malignant melanoma (**b**) in the world population in 2020. Data are presented for each sex and as the average of both sexes. **c**, Cancers with a positive female/male curve peak and PV > 0.20 (PV = premenopausal variation) are highlighted in red, cancers with a negative curve peak and PV < −0.20 are highlighted in blue, and cancers without a curve peak are colored in white (see Supplementary Table 1). **d-f**, Examples of the three categories: (**d**) gastric cancer (positive peak), (**e**) liver cancer (negative peak), and (**f**) brain/CNS cancer (no peak). Dashed lines represent 4th order polynomial fits. **g**, Median testosterone concentration by age in women and men. **h**, CDH1 expression in melanoma by NRAS or BRAF drivers (TCGA-SKCM), Chi-square test. *p ≤ 0.05. **i,j**, Micro- (**i**) and macro- (**j**) metastasis quantification by Ecad and sex. Each dot is an indenpendant mouse. Proportion were assessed by a Chi-square and metastasis counts by two-sided t-test adjusted. Data are represented as mean ± SD.

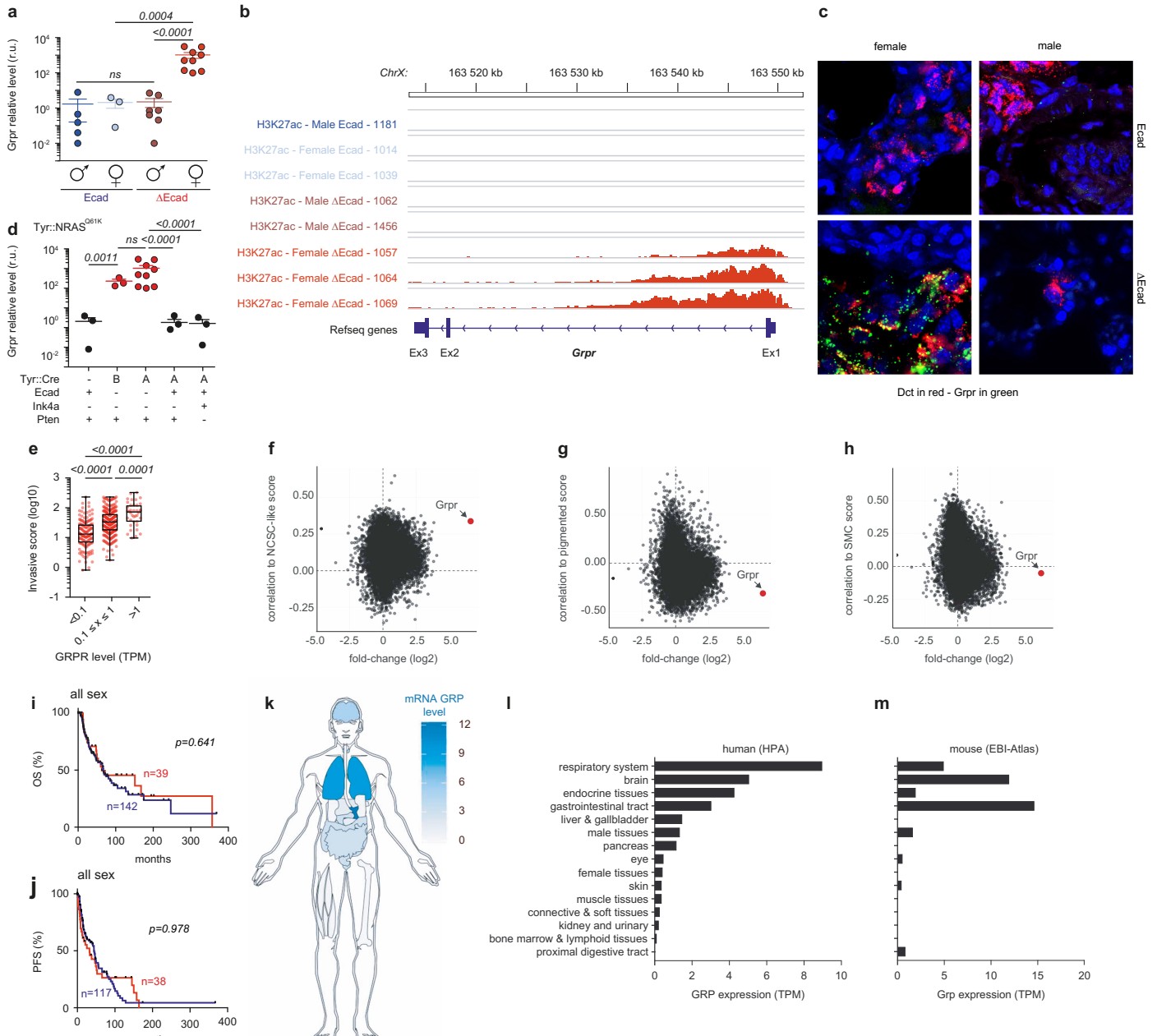

**Extended Data Fig. 2 | Grpr transcription activation in ΔEcad female melanoma and endogenous agonist expression in the lung. a**, RT-qPCR measured *Grpr* mRNA in male and female Ecad and ΔEcad melanoma. **b**, H3K27ac ChIP-seq track for Grpr in mouse melanoma cell lines by sex and Ecad status, aligned to GRCm38 mm10 reference genome. **c**, RNAscope of mouse lungs showing Grpr (green) and Dct (red - as a melanoma marker) in Ecad and ΔEcad male and female transgenic mice; ΔEcad females with lung metastasis show abundant Grpr expression. **d**, Grpr mRNA levels in Tyr::NRAS$^{Q61K}$ female melanomas background with different genetic backgrounds. Tyr::CreA is located on the X chromosome (denoted by 'A'), while Tyr::CreB is located on an autosome (denoted by 'B'). '-' represents the absence of the Cre gene in the genome. Ecad +/+ is indicated by '+', EcadF/F by '-', Ink4a +/+ by '+', and Ink4a +/− by '−'. Similarly, Pten +/+ is represented by '+', and PtenF/+ by '−'. **e**, Invasive score in TCGA-SKCM melanomas with GRPR expression (Spearman coefficient r = 0.35).

**f-h**, Differential gene expression (fold change) correlated with phenotypic scores: (**f**) NCSC-like, (**g**) pigmented, and (**h**) SMC. Grpr is highlighted in red. **i,j**, Overall Survival (OS) (**i**) and Progression-Free Survival (PFS) (**j**) Kaplan-Meier curves for TCGA-SKCM based on GRPR expression (low/absent ≤ 0.1 TPM and expressed >1 TPM). **k-m**, mRNA *GRP* levels in humans (**k,l**) and mice (**m**); abundant in human and mouse lung. Human data sourced from the Human Protein Atlas, mouse data from the EBI Expression Atlas. Anatogram was created with gganatogram. Comparisons significance was assessed by two-sided Mann-Whitney adjusted for multiple comparisons with a Benjamini–Hochberg test. Score significance was assessed by ANOVA adjusted with a Tukey post-test. Survival was assessed by Log-Rank. Data are represented as mean ± SD. Box plot represent the median and the 25–75 percentiles, the whiskers represent the minimum and the maximum. ≥ 3 independent biological replicates were performed per experiment.

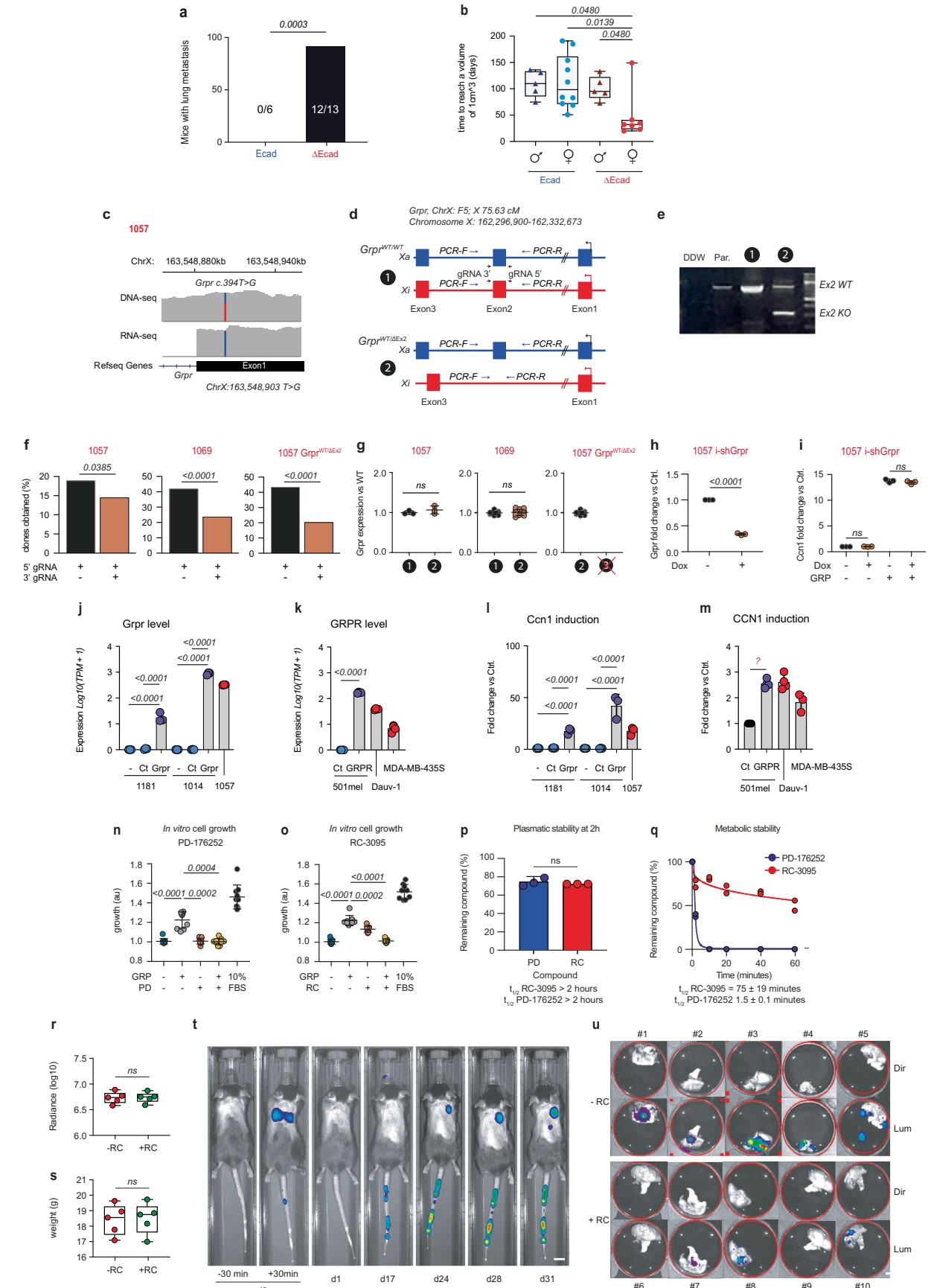

**Extended Data Fig. 3** | See next page for caption.

**Extended Data Fig. 3 | RC-3095 suppresses lung melanoma growth and metastases. a**, Lung metastases observed 30 days post-tail-vein-injection: 12/13 mice injected with ΔEcad cells developed metastasis, whereas no metastasis was observed in 0/6 mice with Ecad cells, **b**, Time to reach 1 cm³ tumor volume in neck-graft reimplantations of male and female Ecad and ΔEcad melanoma in male and female C57BL/6 J mice, respectively. **c**, DNA and RNA sequencing of Grpr in 1057 mouse melanoma cells showing X inactivation. The blue vertical line corresponds to the active X chromosome, while the red line the inactive X chromosome. **d**, Grpr knockout strategy via exon 2 deletion in 1057 mouse melanoma cells. **e**, Genotyping of Grpr deletion clones before and after CRISPR editing. (1) corresponds to Grpr wt/wt, and (2) corresponds to Grpr wt/ΔEx2. DDW = double distilled water. Par = parental. **f**, Clone obtention efficiency in ΔEcad mouse melanoma cells with CRISPR-targeted gRNAs (Chi-square test). **g**, Grpr expression in clonal populations (1) before and (2) after CRISPR editing. The crossed-out number "3" indicates that we were unable to obtain double homozygous knockout mutants Grpr/Cdh1. Statistical analysis was performed using t-test. **h**, Relative mRNA Grpr expression in doxycycline-inducible shRNA-expressing 1057 melanoma cells. **i**, Relative mRNA Ccn1 expression following GRP induction in 1057 melanoma cells. **j,k**, GRPR mRNA levels in mouse (**j**) and human (**k**) melanoma cell lines, normalized in TPM and log-transformed for visualization. Ct = control – cells transfected with an empty expression vector. **l,m**, Fold change in *CCN1* mRNA levels after 4 h 10 nM GRP. Fold changes were calculated for each biological replicate based on TPM-normalized *CCN1* expression. Statistical analysis was performed using one-way ANOVA with a Tukey's post test. **n,o**, Relative growth (au) of 1057 melanoma cells in response to GRP and/or GRPR inhibitors (**n**) PD-176252 (PD) and (**o**) RC-3095 (RC) assessed via MTT assay. **p,q**, Ex vivo stability of RC-3095 (red) and PD-176252 (blue) in murine plasma (**p**) and liver microsomes (**q**). Statistical analysis was performed using t-test. **r,s**, Thorax radiance (**r**) and mean weight (**s**) 30 min post-injection in RC-treated and vehicle-treated groups after randomization. **t**, Representative IVIS images of C57BL/6 J mice intravenously injected with 1057-Luc melanoma cells from day (d) 0 to 31. At d31, a signal was detected in additional organs, and dissection revealed metastases in the liver, adrenal glands, and ovary outside of the lungs at this time. Scale bar = 1 cm. **u**, Luminescence imaging of lungs after euthanasia and dissection of RC-treated (#6 to #10) or untreated (#1 to #5) mice. Scale bar = 1 cm. Lum: luminescence, Dir: direct. Proportions were evaluated by Fisher's exact test. Comparisons significance was assessed by two-sided Mann-Whitney adjusted in case of multiple comparisons with a Benjamini–Hochberg test. Expression data were assessed by a two-sided Student T-test on the log-normalised values. Data are represented as mean ± sd. Box plot represent the median and the 25–75 percentiles, the whiskers represent the minimum and the maximum. ≥ 3 independent biological replicates were performed per experiment.

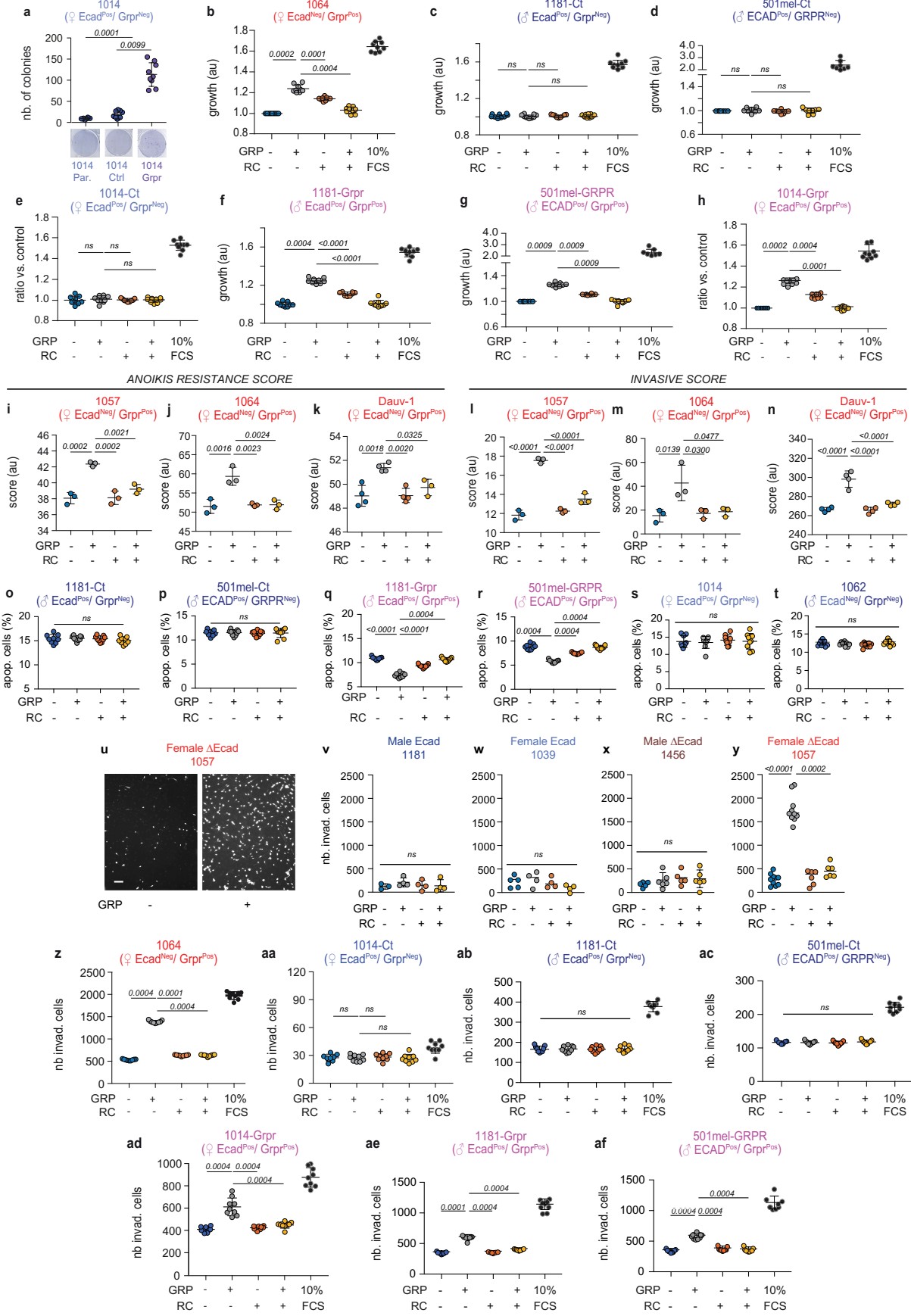

**Extended Data Fig. 4** | See next page for caption.

**Extended Data Fig. 4 | GRPR enhances melanoma cell growth and invasiveness. a**, Clonogenic growth of Grpr[neg] melanoma cell line 1014: parental, control, and after exogenous Grpr expression. **b-h**, Effect of Grpr stimulation on growth of Grpr[pos] and Grpr[neg] mouse and human melanoma cell lines (**b**) 1064, (**c**) 1181-Ct, (**d**) 501mel-Ct, (**e**) 1014-Ct, (**f**) 1181-Grpr, (**g**) 501mel-GRPR, and (**h**) 1014-Grpr. Cell quantification presented as mean ± SD. 1181 and 501mel are GRPR[neg] cells. **i-k**, Anoikis resistance score based on RNA-seq TPM-normalized data for GRPR[pos] murine (**i**) 1057 and (**j**) 1064, and human (**k**) Dauv-1 melanoma cell lines. **l-n**, Invasive score from RNA-seq TPM-normalized data for GRPR[pos] murine 1057 (**l**), 1064 (**m**), and human Dauv-1 (**n**) melanoma cell lines. **o-t**, Resistance to anoikis assays showing apoptotic (apop.) cell percentage after 48 h without matrix attachment. 1181-Ct (**o**), 501mel-Ct (**p**), 1181-Grpr (**q**), 501mel-GRPR (**r**), 1014 (**s**), and 1062 (**t**) cells treated with 10 nM GRP, 1 µM RC-3095 or no treatment. **u-af**, Invasion assays indicating the number (nb) of invading cells (invad.). **u**, Invasion assay of 1057 mouse melanoma cells, with DAPI-stained nuclei after crossing the Matrigel layer. Scale bar = 500 µm. **v-y**, Quantification of invading cells after GRP induction (10 nM) with or without RC-3095 in male Ecad 1181 (**v**), female Ecad 1039 (**w**), male ΔEcad 1456 (**x**), and female ΔEcad 1057 (**y**). **z-af**, Invasion assay in Matrigel® (200 µg/mL) for 1064 (**z**), 1014-Ct (**aa**), 1181-Ct (**ab**), 501mel-Ct (**ac**), 1014-Grpr (**ad**), 1181-Grpr (**ae**), and 501mel-GRPR (**af**) murine and human melanoma cells. Cells were starved overnight and treated with 10 nM GRP, 1 µM RC-3095 and/ or 10% FCS for 24 h (**u-y, z-af**), and for 48 h (**b-h, o-t**). Clonogenic assays were evalued by a Kruskall-wallis corrected by a Dunn's post test. Effect of the GRP induction was assessed by two-sided Mann-Whitney adjusted for multiple comparisons with a Benjamini–Hochberg test. Data shown as mean ± SD. ≥ 3 independent biological replicates were performed per experiment.

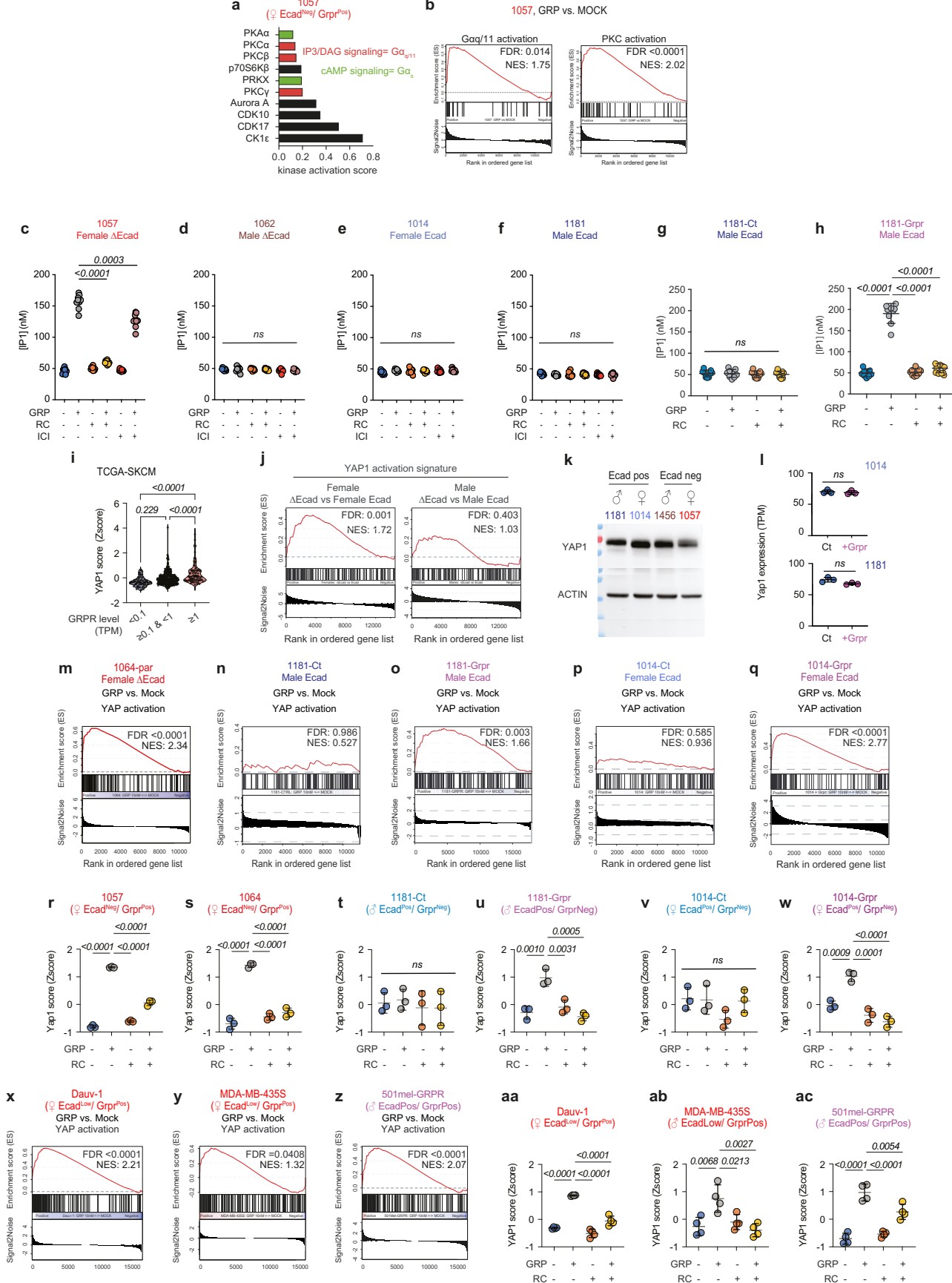

**Extended Data Fig. 5** | See next page for caption.

**Extended Data Fig. 5 | GRPR activation by GRP promotes the YAP1 transcriptional program. a**, Activation of kinases following GRPR activation by 10 nM GRP in GRPR[pos] melanoma cell line 1057, measured 15 min post-induction using the Pamgene® Ser/Thr kinase PamChip (STK). **b**, Gene set enrichment analysis (GSEA) of Gαq (left) and PKC (right) activation signatures 4 h after stimulating GRPR[pos] 1057 cells with 10 nM GRP. Gene expression, determined by RNA-seq, was normalized with DEseq2 before analysis. **c-h**, IP1 levels, indicating Gαq/11 activation, measured in mouse melanoma cell lines after 10 nM GRP stimulation, with or without treatment with the GRPR inhibitor RC-3095 (**c-h**) or pre-treatment with the ERα inhibitor ICI-182,780 (**c-f**). **i**, YAP1 activation score correlated with GRPR expression in TCGA-SKCM melanoma data. TPM: transcripts per million. **j**, GSEA of YAP1 activation signature in mouse primary melanoma tumors: ΔEcad vs Ecad female tumors and male tumors. RNA-seq data normalized with DEseq2 before GSEA. **k**, Western blot showing Yap1 protein levels in mouse melanoma cell lines. Actin as loading control.

Data from one representative experiment of three biological replicates. **l**, Yap1 mRNA expression in melanoma cell line with or without ectopic Grpr and controls not expressing Grpr (1014-Ct, 1014-Grpr, 1181-Ct, 1181-Grpr). **m-q**, GSEA of YAP1 activation in murine melanoma cell lines, 4 h after 10 nM GRP stimulation. RNA-seq data normalized using DEseq2 before GSEA. **r-w**, Yap1 score after GRP induction in murine melanoma cells treated with vehicle, 10 nM GRP, 1 μM RC, or both. **x-z**, GSEA of YAP1 activation in human melanoma cell lines, 4 h after 10 nM GRP stimulation. RNA-seq data normalized using DEseq2 before GSEA. **aa-ac**, Yap1 score in human melanoma cells treated with vehicle, 10 nM GRP, 1 μM RC, or both. GRP effect on the IP1 level was assessed by two-sided Mann-Whitney adjusted for multiple comparisons with a Benjamini–Hochberg test. Significance of the expression was assessed by a tow-sided t-test on the log-normalized data. Significance of the scores was evaluated on the Zscore data by ANOVA adjusted by a Tukey's test. Data are presented as mean ± SD.

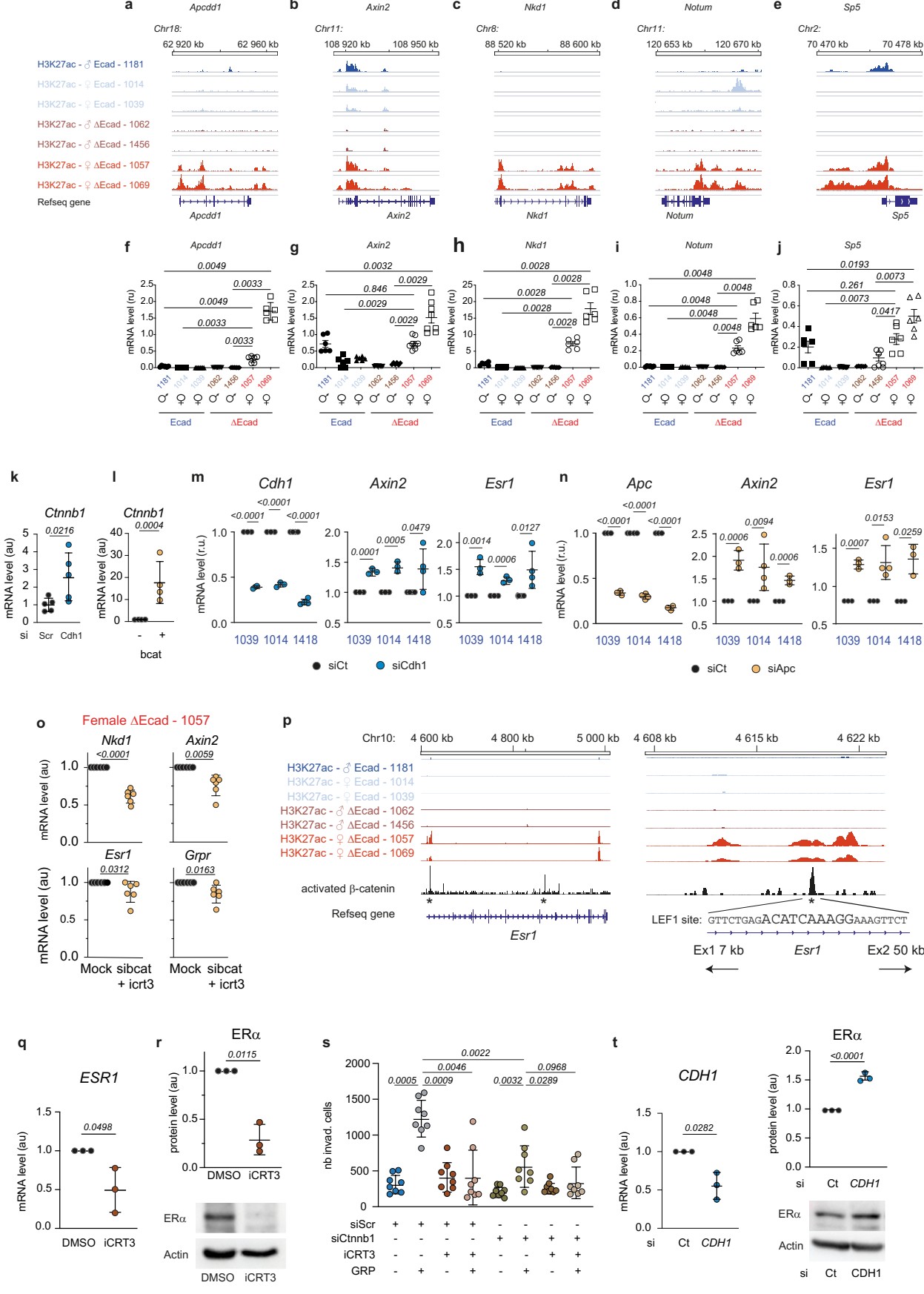

**Extended Data Fig. 6** | See next page for caption.

**Extended Data Fig. 6 | Loss of Ecad in melanoma cells induces β-catenin activity and Esr1 promoter activation. a-e,** ChIP-seq tracks for H3K27ac in male and female NRAS murine melanoma cells with or without Ecad. Known β-catenin target genes: (**a**) *Apcdd1*, (**b**) *Axin2*, (**c**) *Nkd1*, (**d**) *Notum*, and (**e**) *Sp5*. **f-j,** Expression of β-catenin targets (*Apcdd1*, *Axin2*, *Nkd1*, *Notum*, and *Sp5*) in male and female mouse melanoma cells with or without not Ecad. **k,l,** Expression of Ctnnb1 mRNA as determined by RT-qPCR in Ecad mouse 1014 melanoma cells transfected with (**k**) with siScr or siCdh1 in the presence of β-catenin and (**l**) pcDNA3 (bcat -) or with a β-catenin expression vector (bcat +) known as bcat*. **m,** Effect of β-catenin activation via siCdh1 on Cdh1, Axin2, and Esr1 expression in three mouse melanoma cell lines. **n,** Effect of β-catenin activation using siApc on Apc, Axin2, and Esr1 expression in three mouse melanoma cell lines. **o,** RT-qPCR analysis of Nkd1, Axin2, Esr1, and Grpr mRNA levels in ΔEcad mouse 1057 melanoma cells treated with siRNA targeting Ctnnb1 and the pharmacological β-catenin inhibitor iCRT3 (10 µM, 48 h). **p,** β-catenin binding to *Esr1* in nephron precursor cells with significant peaks (called by MACS2) indicated. Genomic locations are based on the mm10. **q,r,** RT-qPCR (**q** and Western blot (**r**) of Esr1 levels in Daju human melanoma cells treated with the β-catenin inhibitor iCRT3 (80 µM, 48 h). **s,** Invasion assays of Grpr-expressing mouse melanoma cells (1057), treated 24 h with siScr, siCtnnb1, iCRT3 and/or GRP. **t,u** RT-qPCR of *CDH1* levels in human melanoma cells treated with siCDH1 (**t**). Western blot analysis of ERα protein levels in human melanoma cells after CDH1 knockdown-three biological replicates (**u**). Relative ERα levels are shown quantified in the histogram. Comparisons significance was assessed by two-sided Mann-Whitney with a Benjamini–Hochberg test. Expression data were assessed by a two-sided Student T-test on the log-normalised values. Zscores were analysed by ANOVA corrected by a Tukey's test. Data are represented as mean ± SD. ≥ 3 independent biological replicates were performed for each experiment.

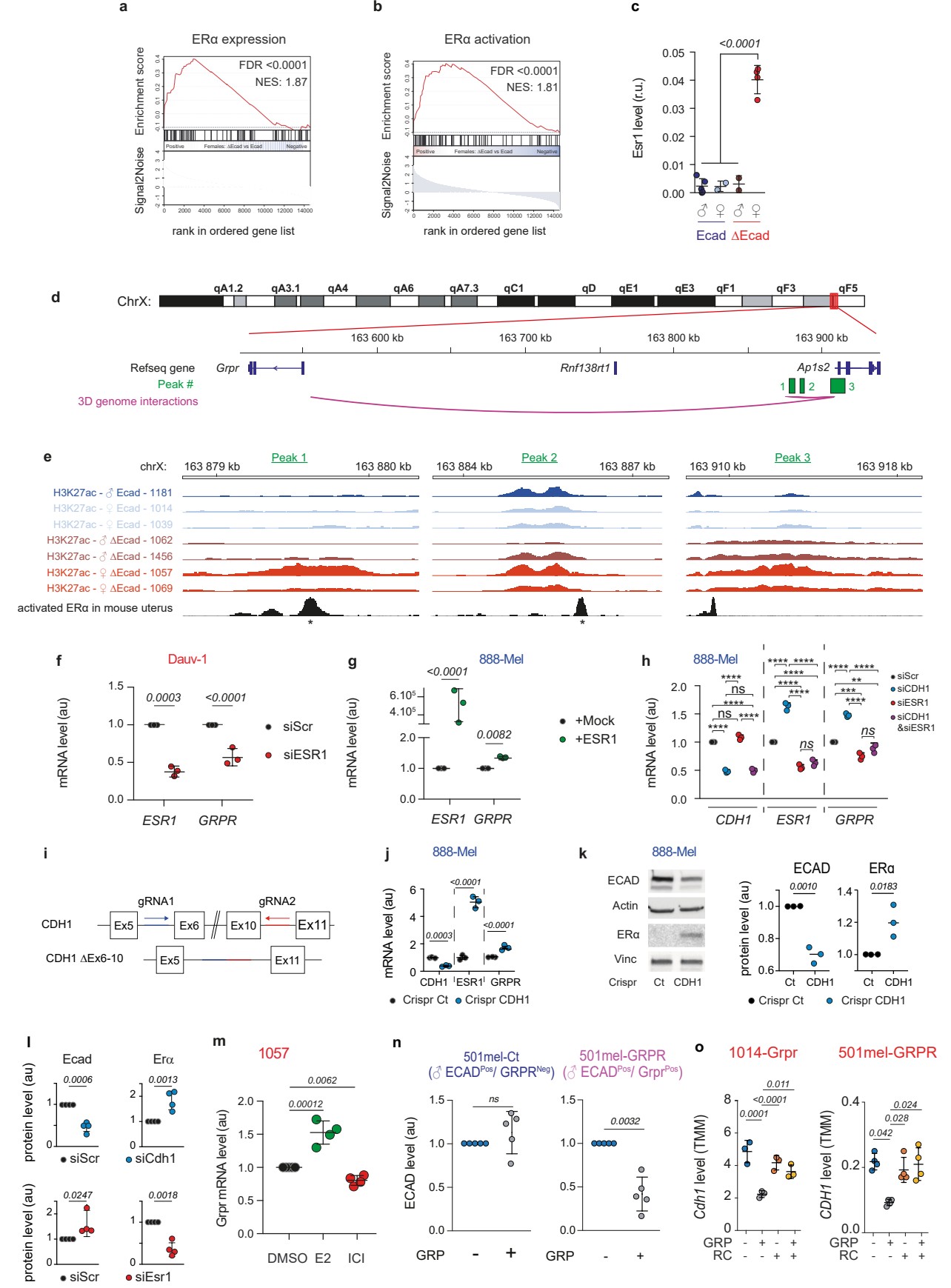

**Extended Data Fig. 7 | Esr1 induces Grpr expression and cell invasion.**
**a,b**, GSEA comparing ERα expression (**a**) and activation (**b**) signatures between murine ΔEcad and Ecad female primary melanoma. **c**, Quantification of Esr1 mRNA level by qRT-PCR and normalized to Hprt expression in various cell lines. **d**, Localization of the Grpr region on the mouse X chromosome, with three upstream regions ERα binding regions (Peak1-3, green boxes) identified through 3D interaction **e**, ChIP-seq tracks for H3K27ac in male and female NRAS murine melanoma cells with or without Ecad in Peak1-3 regions, showing ERα binding peaks from mouse uterus ChIP-seq data (GSM894054). Stars indicate peaks identified by MACS2. **f,g**, Impact of ESR1 silencing (**f**) and overexpression (**g**) on GRPR expression in human melanoma cell lines. Statistical analysis was performed using t-test. **h**, Consequences of CDH1 and/or ESR1 silencing on GRPR expression in CDH1$^{pos}$ human melanoma cells. i, Schematic representation of the CDH1 knockout strategy. Exons 6 to 10 were deleted using two distinct guide RNAs (gRNA1 and gRNA2). j, mRNA expression levels of *CDH1*, *ESR1*, and *GRPR* in the 888-Mel cell population transfected with gRNA targeting *CDH1*. Statistical analysis was performed using t-test. k, Western blot analysis (left) and quantification (right) of ERα and E-cadherin (ECAD) protein levels in the 888-Mel cell population transfected with gRNA targeting *CDH1*. β-Actin and Vinculin (Vinc.) were used as loading controls. Statistical analysis was performed using a *t*-test. **l**, quantification of the western blot analysis of E-cadherin and ERα protein levels in mouse 1014 melanoma cells 48 h after knockdown with siScr (control), siCdh1 (targeting E-cadherin), or siESR1 (targeting ERα) presented in Fig. 5i. **m**, Impact of ERα activation (100 nM β-estradiol, E2) or degradation (1 µM ICI182,780) on GRPR expression in ESR1$^{pos}$ 1057 melanoma cells after estrogen starvation for four days. **n**, Quantification of the western blot analysis of (ECAD) expression in 501mel cells, expressing or not GRPR, in the presence or absence of 10 nM GRP presented in Fig. 5j. **o**, mRNA CDH1 levels in melanoma cells expressing GRPR, measured in the presence or absence of GRP and/or RC-3095. TMM = Trimmed Mean of M-values. For expression data, significance was evaluated by two-sided Student T-test (two groups) or ANOVA adjusted by a Tukey's test (multiple groups) on the log-normalised data. Data are represented as mean ± SD. ≥ 3 independent biological replicates were performed per experiment. *p < 0.05, **p < 0.01, ***p < 0.001, and ****p < 0.0001.

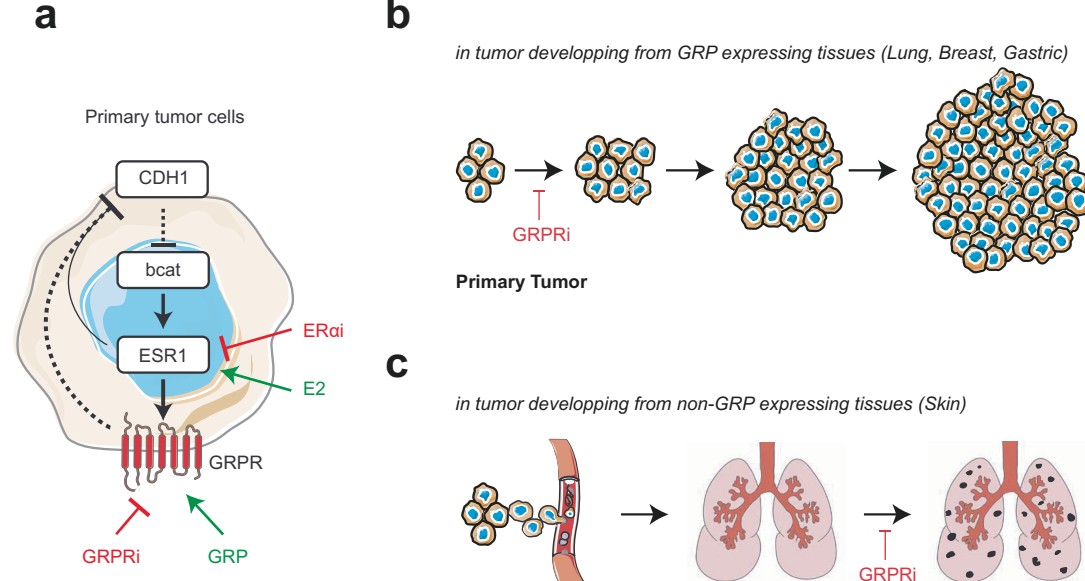

**a**

Primary tumor cells

CDH1

bcat

ESR1 — ERαi

E2

GRPR

GRPRi GRP

**b**

*in tumor developping from GRP expressing tissues (Lung, Breast, Gastric)*

GRPRi

**Primary Tumor**

**c**

*in tumor developping from non-GRP expressing tissues (Skin)*

GRPRi

**Metastasis**

**Extended Data Fig. 8 | Graphical abstract. a**, In primary melanoma, the loss of CDH1 activates β-catenin signaling, subsequently triggering the upregulation of ESR1. Consequently, ERα stimulates the expression of GRPR, particularly potent in the presence of estrogen (E2), especially during the period between puberty and menopause. Activation of GRPR by GRP subsequently reduces CDH1 expression, thereby reinforcing this loop and resulting in elevated levels of GRPR expression. ERa and GRPR inhibitors have the potential to affect this loop of regulation. **b**, ECAD[neg]/GRPR[pos] cancer cells grow in tissues expressing naturally GRP including lung, breast and gastric tissue. **c**, ECAD[neg]/GRPR[pos] melanoma cells gain the ability to disseminate through the bloodstream to distant organs, notably the lungs, where GRP is produced in abundance. Within the lungs, the interaction between GRP and GRPR initiates pro-metastatic signaling in melanoma cells.

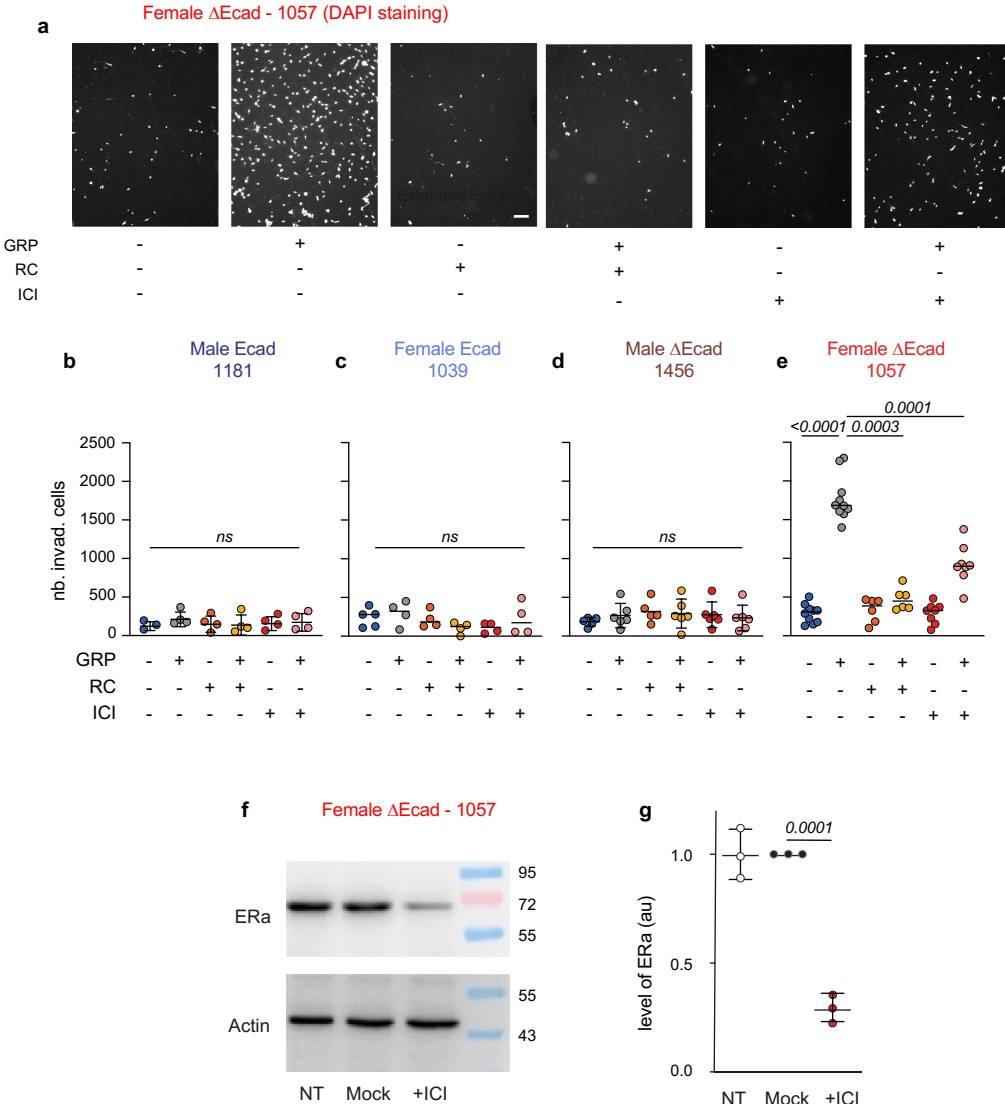

Female ΔEcad - 1057 (DAPI staining)

**a**

| | | | | | |
|---|---|---|---|---|---|
| GRP | - | + | - | + | - | + |
| RC | - | - | + | + | - | - |
| ICI | - | - | - | - | + | + |

**b** Male Ecad 1181   **c** Female Ecad 1039   **d** Male ΔEcad 1456   **e** Female ΔEcad 1057

nb. invad. cells

*ns* (b, c, d)

e: <0.0001  0.0003  0.0001

GRP  - + - + - +
RC   - - + + - -
ICI  - - - - + +

**f** Female ΔEcad - 1057

ERa — 95 / 72 / 55

Actin — 55 / 43

NT   Mock   +ICI

**g** 0.0001

level of ERa (au)

NT   Mock   +ICI

**Extended Data Fig. 9 | GRP and ERα modulate melanoma cell invasion and in a sex- and E-cadherin–dependent manner. a**, Invasion assay of 1057 mouse melanoma cells, quantified by counting DAPI-stained nuclei that migrated through the Matrigel layer. Scale bar = 500 μm. **b-e**, Quantification of invading cells in various melanoma cell lines: male Ecad 1181 (**b**), female Ecad 1039 (**c**), male ΔEcad 1456 (**d**), and female ΔEcad 1057 (**e**) treated with 10 nM GRP, 1 μM RC-3095, and/or 1 μM ICI. **f**, Western blot analysis of ERα protein levels in mouse melanoma cells treated or not with ICI (1 μM, 24 h) with actin as loading control. **g**, Histograms showing ERα quantification from three independent biological experiments. GRP-induction significance was assessed by two-sided Mann-Whitney adjusted with a Benjamini–Hochberg test. Quantification was evaluated by ANOVA on the log transformed data. Data are represented as mean ± SD. ≥ 3 independent biological replicates were performed per experiment.

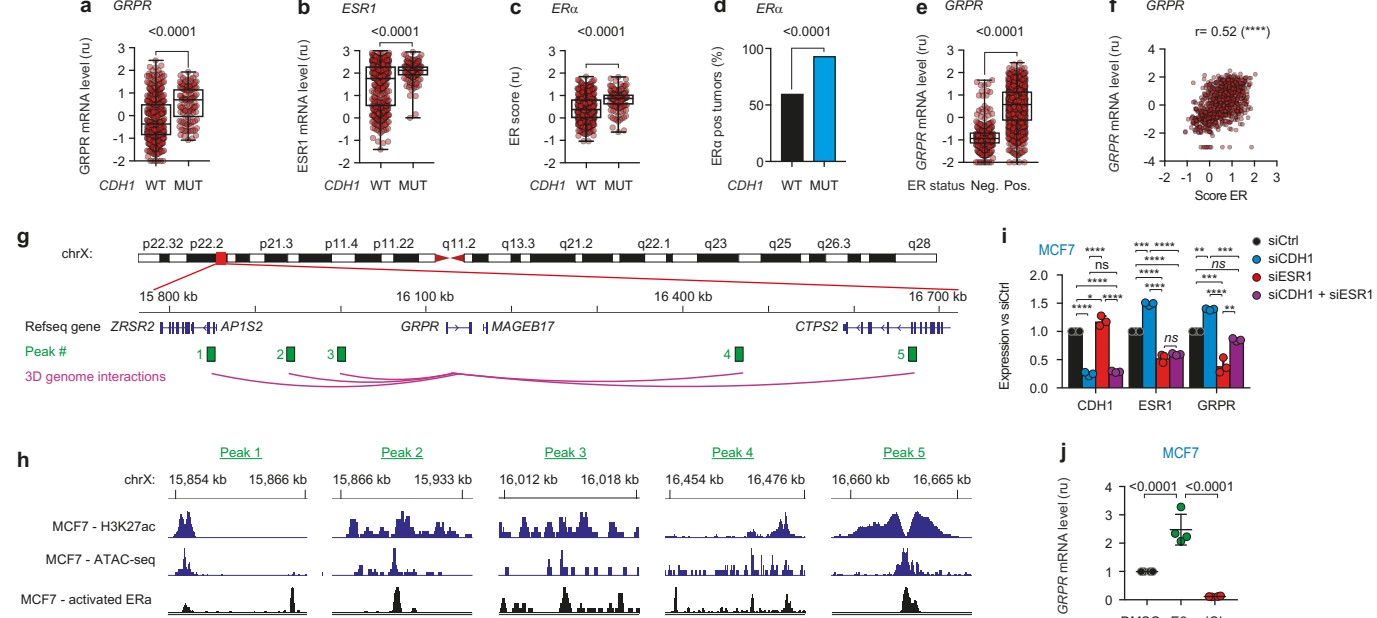

**Extended Data Fig. 10 | The CDH1/CTNNB1/ESR1/GRPR Axis Is Active in Breast Carcinomas. a,b,** Expression of GRPR (**a**) and ESR1 (**b**) mRNA in breast carcinoma from TCGA-BRCA data based on *CDH1* genetic status. **c,** ERα activation score according to the *CDH1* genetic status in breast cancer tumors from TCGA data. **d,** Quantification of ERα positivity in breast carcinomas from the TCGA-BRCA cohort by IHC based on CDH1 mutation. **e,** Expression of GRPR based on ERα status defined by TCGA pathologists using immunohistochemistry in human breast carcinomas from the TCGA-BRCA studies. **f,** Correlation between GRPR expression and the ERα activity score calculated from the same breast tumors from the TCGA-BRCA studies. **g,** Localization of the GRPR region on the human X chromosome (top) and identification of five regions with chromatin openness (H3K27ac) and ERα binding revealed by 3D interaction with the GRPR locus in MCF7 breast cancer cells. **h,** Genome browser view of ChIP-seq tracks for H3K27ac, ATAC-seq, and E2-activated ERα in MCF7 breast

cancer cells on the five regions. Three regions/peaks (1-3) are conserved between humans and mice. **i,** Effects of CDH1 and/or ESR1 silencing on GRPR expression in CDH1-positive MCF7 human breast cancer cells. (see legend of Fig. 5h for more information). **j,** Effect of ESR1 stimulation (17β-estradiol, 100 nM) and inhibition (ICI 182,780, 1 μM) on mRNA GRPR expression in MCF7 cells. Expression were compared with a Student T-test (two groups) or anova with Tukey's posttest (multiple groups) performed on the log-transformed data. Proportions were analyzed by Chi-square and correlation by Pearson. All panels, ns = not significant, *p < 0.05, **p < 0.01, ***p < 0.001, and ****p < 0.0001. The relative mRNA levels in (a, c, d, and g) are presented as Log10 (TPM + 0.01). ≥3 independent biological replicates were performed per experiment. Box plot represent the median and the 25–75 percentiles, the whiskers represent the minimum and the maximum.

# Reporting Summary

## Statistics

For all statistical analyses, confirm that the following items are present in the figure legend, table legend, main text, or Methods section.

| n/a | Confirmed | |
|---|---|---|
| ☐ | ☒ | The exact sample size (*n*) for each experimental group/condition, given as a discrete number and unit of measurement |
| ☐ | ☒ | A statement on whether measurements were taken from distinct samples or whether the same sample was measured repeatedly |
| ☐ | ☒ | The statistical test(s) used AND whether they are one- or two-sided<br>*Only common tests should be described solely by name; describe more complex techniques in the Methods section.* |
| ☐ | ☒ | A description of all covariates tested |
| ☐ | ☒ | A description of any assumptions or corrections, such as tests of normality and adjustment for multiple comparisons |
| ☐ | ☒ | A full description of the statistical parameters including central tendency (e.g. means) or other basic estimates (e.g. regression coefficient) AND variation (e.g. standard deviation) or associated estimates of uncertainty (e.g. confidence intervals) |
| ☐ | ☒ | For null hypothesis testing, the test statistic (e.g. *F*, *t*, *r*) with confidence intervals, effect sizes, degrees of freedom and *P* value noted<br>*Give P values as exact values whenever suitable.* |
| ☒ | ☐ | For Bayesian analysis, information on the choice of priors and Markov chain Monte Carlo settings |
| ☐ | ☒ | For hierarchical and complex designs, identification of the appropriate level for tests and full reporting of outcomes |
| ☐ | ☒ | Estimates of effect sizes (e.g. Cohen's *d*, Pearson's *r*), indicating how they were calculated |

*Our web collection on statistics for biologists contains articles on many of the points above.*

## Software and code

Policy information about availability of computer code

| Data collection | Epidemiological data were dowloaded from the Global Cancer Observatory repository on the International Agency for Research on Cancer (https://gco.iarc.fr). Transcriptomic and genomic data were collected using the open source R package TCGAbiolinks (v 2.28.3) or by directly downloading them from cBioportal (http://www.cbioportal.org/). Kinase activity were acquired using the PamStation PS12 (PamGene International B.V.). |
|---|---|
| Data analysis | No custom codes were developed to analyse the data associated with this manuscript. All codes used originate from well-established R packages: DEseq2 (v1.40.2), edgeR (v3.42.4), sva (v3.48.0), vsn (v3.68.0), ChIPseeker (v1.36.0). Functions from these packages were used using default parameters except if stated otherwise in the Material and Methods section of this manuscript. Data were aligned using RNA-STAR (2.7.11) Bowtie (2.5.3) or BWA (0.7.19). MarkDuplicates (v3.1.1.0) was used to remove PCR duplicates. Peaks were called with MACS2 (v2.2.7.1). TCGA data were downloaded using TCGAbiolinks (v2.28.3).<br>GSEA analyses were performed using the GSEA software from the Broad Institute (v4.0.3). PamGene Kinase assays were analysed using the Bionavigator software, PamGene International B.V.). Automatic counting of invading cells was done using an macro (https://zenodo.org/records/14509394) on ImageJ/Fiji (v2.14.0/1.54f). Data were represented using GraphPad Prism 10 (v10.2.3) or R (4.4.0) and the package ggplot2 (v3.5.1). |

For manuscripts utilizing custom algorithms or software that are central to the research but not yet described in published literature, software must be made available to editors and reviewers. We strongly encourage code deposition in a community repository (e.g. GitHub). See the Nature Portfolio guidelines for submitting code & software for further information.

## Data

Policy information about availability of data

All manuscripts must include a data availability statement. This statement should provide the following information, where applicable:

- Accession codes, unique identifiers, or web links for publicly available datasets
- A description of any restrictions on data availability
- For clinical datasets or third party data, please ensure that the statement adheres to our policy

All sequencing data generated with this manuscript were deposited on the relevant platform. RNA-seq and ChIP-seq data were deposited on GEO at the National Center for Biotechnology Information under the SuperSeries GSE218588 (https://www.ncbi.nlm.nih.gov/geo/query/acc.cgi?acc=GSE218588). That includes: the mouse tumour RNA-seq under accession number GSE218532 (https://www.ncbi.nlm.nih.gov/geo/query/acc.cgi?acc=GSE218532), the mouse cell line RNA-seq under accession number GSE218586 (https://www.ncbi.nlm.nih.gov/geo/query/acc.cgi?acc=GSE218586) and the human cell line data are deposited under accession number GSE218530 (https://www.ncbi.nlm.nih.gov/geo/query/acc.cgi?acc=GSE218530). The ChIP-seq data are available under accession number GSE237500 (https://www.ncbi.nlm.nih.gov/geo/query/acc.cgi?acc=GSE237500). Whole exome sequencing of the mouse melanoma cell lines are available from SRA under the bioproject PRJNA904253 (https://dataview.ncbi.nlm.nih.gov/object/40767225) Kinase assay raw data were deposited on Mendeley at the following DOI 10.17632/nwkpyr2nmh.1 (Currently under publication, will provide the final DOI shortly). All other data supporting the findings of this study are available on reasonable request. Source data are provided with this paper.

The following datasets from the literature were used: ChIP-seq of ERa in mouse (GSM894054, https://www.ncbi.nlm.nih.gov/geo/query/acc.cgi?acc=GSM894054) and human (GSM798434, https://www.ncbi.nlm.nih.gov/geo/query/acc.cgi?acc=GSM798434), of b-catenin in mouse (GSM980186, https://www.ncbi.nlm.nih.gov/geo/query/acc.cgi?acc=GSM980186) and human (GSM1579346, https://www.ncbi.nlm.nih.gov/geo/query/acc.cgi?acc=GSM1579346) and of H3K27Ac in human MCF7 (GSM2175784, https://www.ncbi.nlm.nih.gov/geo/query/acc.cgi?acc=GSM2175784). ATAC-seq data from the MCF7 cell line originate from GSM2645717 (https://www.ncbi.nlm.nih.gov/geo/query/acc.cgi?acc=GSM2645717). The 3D chromatin interactions were downloaded from GSE207828 (https://www.ncbi.nlm.nih.gov/geo/query/acc.cgi?acc=GSE207828) for mouse and from ENCODE accession number ENCFF804SET (https://www.encodeproject.org/experiments/ENCSR549MGQ/) and from GSE52457 (https://www.ncbi.nlm.nih.gov/geo/query/acc.cgi?acc=GSE52457) for human. Expression data from acral melanoma is from GSE190113 (https://www.ncbi.nlm.nih.gov/geo/query/acc.cgi?acc=GSE190113). TCGA datasets were accessed through TCGABiolinks or CBioPortal (https://www.cbioportal.org/).

All data have been aligned to the human reference genome GRCh38 (hg38 gencode 42 version GRCh38.p13) and the mouse reference genome mm10 gencode 13 version GRCm38.p5.

## Research involving human participants, their data, or biological material

Policy information about studies with human participants or human data. See also policy information about sex, gender (identity/presentation), and sexual orientation and race, ethnicity and racism.

| | |
|---|---|
| Reporting on sex and gender | Patient gender was defined by their self reporting. |
| Reporting on race, ethnicity, or other socially relevant groupings | No information has been collected on race, ethnic criteria or any socially relevant grouping because the collection of this type of data is legally prohibited in France (Law no. 78-17 of 6 January 1978 on data processing, data files and individual liberties, Article 226-19 of the Criminal Code) |
| Population characteristics | The population had an average median of 58 years old (25% interquartile range: 48-66). Their genotypes (52% BRAF mutated, 17% NRAS mutated) represent the known frequency of mutation this two main driver genes. |
| Recruitment | The recruitment was made retrospectively on the available sample from the tissue collection of the Rennes and Bordeaux Hospitals. |
| Ethics oversight | The retrospective study on lung human melanoma metastases was approved by the ethics committee. The non-opposition or consent (before or after 2004, respectively) of patients for the use of their biological material and data was obtained according to the bioethics law of 2004. This study was approved by the C.H.U. de Bordeaux. |

Note that full information on the approval of the study protocol must also be provided in the manuscript.

## Field-specific reporting

Please select the one below that is the best fit for your research. If you are not sure, read the appropriate sections before making your selection.

☒ Life sciences   ☐ Behavioural & social sciences   ☐ Ecological, evolutionary & environmental sciences

For a reference copy of the document with all sections, see nature.com/documents/nr-reporting-summary-flat.pdf

## Life sciences study design

All studies must disclose on these points even when the disclosure is negative.

| | |
|---|---|
| Sample size | The required number of mouse per group was determined by calculating the minimum number of subject to have a sufficient study power. The statistical parameter applied were: |

- We expect a minimum of 30% variation in between our conditions
- The type 1 error set at 5%
- The power was set at 80%.
Based on this number it was calculated that we needed 5 mice per group.

| Data exclusions | No data were excluded during this work |
| --- | --- |
| Replication | All in vitro experiments were successfully performed at least with three independent biological replicates. For the experiments involving cell lines, each replicate corresponds to different passages. |
| Randomization | For every in vivo experiment, mouse were blind randomized using an online randomization software (https://www.randomizer.org/). |
| Blinding | Experiments were not performed in a blind manner. However, biological replicates were performed by different individual without prior knowledge of the results of the others. |

# Reporting for specific materials, systems and methods

We require information from authors about some types of materials, experimental systems and methods used in many studies. Here, indicate whether each material, system or method listed is relevant to your study. If you are not sure if a list item applies to your research, read the appropriate section before selecting a response.

## Materials & experimental systems

| n/a | Involved in the study |
| --- | --- |
| ☐ | ☒ Antibodies |
| ☐ | ☒ Eukaryotic cell lines |
| ☒ | ☐ Palaeontology and archaeology |
| ☐ | ☒ Animals and other organisms |
| ☐ | ☒ Clinical data |
| ☒ | ☐ Dual use research of concern |
| ☒ | ☐ Plants |

## Methods

| n/a | Involved in the study |
| --- | --- |
| ☐ | ☒ ChIP-seq |
| ☒ | ☐ Flow cytometry |
| ☒ | ☐ MRI-based neuroimaging |

## Antibodies

| Antibodies used | o YAP: clone D8H1X, Cell Signaling Technology, Cat#14074, RRID: AB_2650491<br>o Gastrin-Releasing Peptide Receptor (GRPR), Acris Antibodies, Cat#SP4337P, RRID: AB_1001744<br>o E-cadherin, Leica Biosystems Cat#E-CAD-L-CE<br>o E-cadherin, BD transduction laboratories, Cat#610182, RRID: AB_397581<br>o Histone H3K27ac, Active Motif, Cat#39133, RRID: AB_2722569<br>o Vinculin, Cell Signaling Technology, Cat#4650, RRID: AB_10559207<br>o ERa,Invitrogen, Cat#MA1-27107, RRID: AB_780508<br>o B-actin, Sigma-Aldrich, Cat#A5441, RRID: AB_476744<br>o goat anti-rabbit Alexa fluor 594, Invitrogen, Cat#A-11012, RRID: AB_2534079 |
| --- | --- |
| Validation | Validation of the antibodies used for this study:<br>o YAP: this antibody has been validated by the knock-out of YAP1 in Hela cells which leads to the loss of detection of the protein in cell whereas YAP remained detectable in the control cells.<br>o GRPR: we validate this antibody human tissues by incubating tissues known to express or not GRPR at the RNA level with the antibody. Only the tissues expressing GRPR were stained in immunohistochemistry.<br>o E-cadherin: we validate this antibody human tissues by incubating tissues known to express or not Ecad at the RNA level with the antibody. Only the tissues expressing Ecad were stained in immunohistochemistry.<br>o Histone H3K27ac: This antibody was validated by the manufacturer notably to for its use in ChIP-seq.<br>o anti-B-actin was validated by the manufacturer for their usage in western blot ("The antibody specifically labels β-actin in a wide variety of tissues and species using immunoblotting (42 kDa)", and "species reactivity: sheep, carp, feline, chicken, rat, mouse, Hirudo medicinalis, rabbit, canine, pig, human, bovine, guinea pig"). This antibody is also highly used in the litterature (>890 citations)<br>o Vinculin antibodies were validated by the manufacturer for their usage in western blot. Both of those antibodies are widely used in the literature in this application (>390 citations).<br>o ERa the antibody was validated by the manufacturer "Estrogen Receptor alpha Antibody (MA1-27107) Antibody specificity was demonstrated by detection of differential basal expression of the target across cell lines and tissues owing to their inherent genetic constitution". We further validated the antibody by immunobloting ERa in our mouse melanoma cell line and showed that the protein expression correlates with the mRNA expression (Figure 5d-e). The specificity was proven by knocking-down Esr1 expression and showing that it decreases ERa signal detected by immunoblotting (Figure 5i). |

## Eukaryotic cell lines

Policy information about cell lines and Sex and Gender in Research

| Cell line source(s) | Mouse cell lines were established by the laboratory during this study. |
| --- | --- |

| | |
|---|---|
| Cell line source(s) | Human melanoma cell lines were described previously by the laboratory (Rambow et all. Cell Report, 2015). MCF7 were gifted by the Dutreix laboratory (Institut Curie, Orsay, France). |
| Authentication | Cells were authenticated by the STR profiling method. |
| Mycoplasma contamination | All cells were frequently tested for mycoplasma contamination. None of them were contaminated. |
| Commonly misidentified lines (See ICLAC register) | No commonly misidentified cell lines were used in this study. |

## Animals and other research organisms

Policy information about studies involving animals; ARRIVE guidelines recommended for reporting animal research, and Sex and Gender in Research

| | |
|---|---|
| Laboratory animals | This study was conducted using mouse of the following strains: <br> - C57BL/6J, RRID:IMSR_JAX:000664, Charles River, aged of 8 weeks when included in the protocol <br> - NSG (NOD.Cg-Prkdcscid Il2rgtm1Wjl/SzJ), RRID:IMSR_JAX:005557, Charles River, aged of 8 weeks when included in the protocol <br><br> As well as te following transgenic models: <br> -Tyr::CreA (B6.Cg-Tg(Tyr-cre)1Lru/J), Larue Lab, RRID: IMSR_JAX:029788Info <br> - Tyr::CreB (Delmas et al., Genesis. (2003), 36, 73-80. PMID: 12820167) <br> -Tyr::NRAS (Tg(Tyr-NRAS*Q61K)1Bee), Beerman Lab, RRID:MGI:5515808 <br> -Cdkn2a+/- (Cdkn2atm1Rdp) De Pinho Lab, RRID:IMSR_NCIMR:01XB1 <br> -Cdh1F/+ B6.129-CDH1tm2kem/J), Kemler Lab, RRID:IMSR_JAX:005319 <br> All transgenic models were included in the study from their birth date up to 120 weeks if no melanoma appeared (see Figure 1F and associated source data for per mouse details). All these transgenic mouse lines have been backcrossed at least 10 times with C57BL/6J mouse (including male and female mice). <br> Mice were housed in a spf certified animal facility with a 12-hour light/dark cycle in a temperature-and humidity-controlled room (22 ±1°C and 60% respectively) with free access to water and food. |
| Wild animals | No wild animals were involved in this study. |
| Reporting on sex | For the transgenic animals, all sexes were considered. For the WT animals used for cell injections, the sex of the animal correspond to the sex of the injected cell line. |
| Field-collected samples | No samples were collected from the field. |
| Ethics oversight | Animal care, use, and all experimental procedures were conducted in accordance with recommendations of the European Community (86/609/EEC) and Union (2010/63/UE) and the French National Committee (87/848). Animal care and use were approved by the ethics committee of the Curie Institute in compliance with institutional guidelines. Experimental procedures were carried out under the approval of the ethics committee of the Institut Curie CEEA-IC #118 (CEEA-IC 2016-836 001) in compliance with international guidelines |

Note that full information on the approval of the study protocol must also be provided in the manuscript.

## Clinical data

Policy information about clinical studies
All manuscripts should comply with the ICMJE guidelines for publication of clinical research and a completed CONSORT checklist must be included with all submissions.

| | |
|---|---|
| Clinical trial registration | N/A |
| Study protocol | N/A |
| Data collection | N/A |
| Outcomes | N/A |

## Plants

| | |
|---|---|
| Seed stocks | N/A |
| Novel plant genotypes | N/A |
| Authentication | N/A |

# ChIP-seq

## Data deposition

☒ Confirm that both raw and final processed data have been deposited in a public database such as GEO.

☒ Confirm that you have deposited or provided access to graph files (e.g. BED files) for the called peaks.

| | |
|---|---|
| Data access links<br>*May remain private before publication.* | https://www.ncbi.nlm.nih.gov/geo/query/acc.cgi?acc=GSE237500<br>token: mxuzyaekdvifnkz |
| Files in database submission | GSM7622286_1007_AC.bigwig<br>GSM7622287_1181_AC.bigwig<br>GSM7622288_1014_AC.bigwig<br>GSM7622289_1039_AC.bigwig<br>GSM7622290_1062_AC.bigwig<br>GSM7622291_1456_AC.bigwig<br>GSM7622292_1057_AC.bigwig<br>GSM7622293_1064_AC.bigwig<br>GSM7622294_1069_AC.bigwig<br>GSM7622295_Ecad_male_Input.bigwig<br>GSM7622296_Ecad_female_Input.bigwig<br>GSM7622297_dEcad_male_Input.bigwig<br>GSM7622298_dEcad_female_Input.bigwig<br>peaks_1007_H3K27ac.bed<br>peaks_1181_H3K27ac.bed<br>peaks_1014_H3K27ac.bed<br>peaks_1039_H3K27ac.bed<br>peaks_1062_H3K27ac.bed<br>peaks_1456_H3K27ac.bed<br>peaks_1057_H3K27ac.bed<br>peaks_1064_H3K27ac.bed<br>peaks_1069_H3K27ac.bed |
| Genome browser session<br>(e.g. UCSC) | http://genome.ucsc.edu/cgi-bin/hgTracks?<br>db=mm10&lastVirtModeType=default&lastVirtModeExtraState=&virtModeType=default&virtMode=0&nonVirtPosition=&po<br>sition=chrX%3A163191359%2D164097582&hgsid=1725092268_Ac6A1PXaGTMqNJPmaWuMKZOlYsEy |

## Methodology

| | |
|---|---|
| Replicates | We performed the ChIP-seq on two cell lines of each Cdh1 genetic status and of each sex. |
| Sequencing depth | AC-1062: 80186942 reads, 78% uniquely mapped<br>AC-1057: 85720740 reads, 80% uniquely mapped<br>AC-1064: 60195608 reads, 76% uniquely mapped<br>AC-1069: 97118449 reads, 72% uniquely mapped<br>AC-1007: 91033436 reads, 81% uniquely mapped<br>AC-1181: 55183731 reads, 72% uniquely mapped<br>AC-1456: 71277500 reads, 73% uniquely mapped<br>AC-1014: 83138809 reads, 74% uniquely mapped<br>AC-1039: 99758567 reads, 81% uniquely mapped<br>Input-1062: 21131673 reads, 58% uniquely mapped<br>Input-1181: 83424648 reads, 64% uniquely mapped<br>Input-1069: 24846923 reads, 39% uniquely mapped<br>Input-1014: 29629266 reads, 60% uniquely mapped<br>All reads were pair-ends, read length is 100bp |
| Antibodies | Histone H3K27ac antibody, Active Motif, Cat#39133, RRID: AB_2722569 |
| Peak calling parameters | Peak calling was performed using MACS2 (v2.2.7.1). Default parameters were used except for the  following:<br> - Input files as paired-end BAM<br> - Effective genome size: 1.87e9 (M.musculus genome size)<br> - q-value threshold= 1e-05<br> - Band width for picking regions to compute fragment size: 300bp |
| Data quality | ChIP seq quality was assessed by the simple exponential smoothing method. Each ChIP fingerprint was plotted and it was verified that most of the reads correspond to a small part of the genome except for the input samples where reads were homogeneously distributed across the genome. For all samples, 50% of the reads were located in only 5% of the genome, which is sign of a strong and localized enrichments. For the input, these 50% of the reads are aligned to 20 to 35% of the genome. |
| Software | ChIP-seq were analysed using Galaxy. Fastq were trimmed using FASTQ trimmer and then align on the mm10 mouse genome using bowtie2. Peaks were called using MACS2 and the parameter defined above. For representation purpose, bigwig were generated. |

These bigwig were generated using the BamCoverage function from DeepTools. Bin size was set at 20 bases. Reads were normalized to bins per million and the scaling factor was set to 1.

