## [Peer Review File · Nature]

Targeting GRPR for sex hormone-dependent cancer after loss of E-cadherin

Corresponding Author: Dr Lionel Larue

Version 1:

Reviewer comments:

Referee #1

(Remarks to the Author)

The manuscript „Targeting GRPR for sex hormone-dependent cancer after loss of E-cadherin“ by Larue and colleagues describes an unexpected role of estrogen receptor signaling in cancer prevalence and metastasis. At the beginning of their paper, the authors separately consider overall tumor incidence in men and women (where overall incidence is reportedly higher in men) and age-related tumor incidence. The latter analysis reveals a striking increase in tumor incidence in females during their reproductive age, which generally correlates with high estradiol levels. This was observed for several tumor types including melanoma. In the following, the authors show – in most parts using melanoma as model tumor - that the surface molecule E-cadherin (CDH1) is functionally linked to estrogen receptor 1 and expression of pro-metastatic GRPR. Specifically, this link is supported by the following observations:

- 1) melanomas from females <50 years have increased genomic alterations in CDH1/Ecad as well as lower CDH1 expression,
- 2) Ecad knockout in an NRAS-driven mouse melanoma model led to the development of lung metastases specifically in female mice
- 3) Ecad knockout in females causes a strong expression of Gastrin-Releasing Peptide Receptor (Grpr), which shows pro-tumorigenic and metastatic abilities itself, e.g. after tailvein injection of murine or human melanoma cells overexpressing exogenous Grpr. Importantly, pharmacological inhibition of Grpr strongly inhibited lung colonization of these cells.
- 4) Grpr induces a YAP-associated pro-metastatic gene program in mouse melanoma cells
- 5) Ecad downregulation leads to the upregulation of Esr1 (probably mediated by beta catenin), which mediates Grpr upregulation. Grpr activation (e.g. by GRP) blocks CDH1 expression, thus forming a loop.

This study is generally of high interest across disciplines. However, most data were generated in mouse models and the deregulations of Grpr and Esr1 are based on RNA expression data. Some points still need to be addressed, as detailed below:

Major points:

- 1) The Ecad  Esr1  Grpr loop, as shown in Figure 6m, should be demonstrated on protein level in human and murine cells. Furthermore, it is important to compare the protein levels of endogenous upregulation of Esr1 and Grpr (as caused by Ecad knockdown) with that of exogenous Grpr overexpression (that was used in Figure 3, Extended Figure 3 and Figure 5).
- 2) According to public databases such as the proteinatlas, Grpr expression is hardly expressed outside the gastrointestinal tract. Also, expression in tumor cell lines is extremely weak in most cell lines except in breast cancer. How can the authors reconcile this with their own observations?
- 3) The age group of interest (approx. 20-50 yrs) is also a group in which the occurrence of heritable cancers is proportionately more likely than in older patients. Thus, the data might point at the possibility that in humans with a cancer predisposition, young women are more likely to develop cancer due to estrogen alterations. This should be discussed. Could the authors provide information about the contribution of familial cancer in males versus females in this age group?
- 4) In my opinion, Figure 1d plays an important role, as it provides the connection to the following studies in the manuscript.

However, it is not quite clear, where the cancer-associated genes" characterizing the chosen cancer entities in Figure 1d (top three female-biased cancers (melanoma, gastric, and thyroid), breast cancer, and those associated with the main estrogen receptor type 1) come from. The information shared in the Materials and Methods are not explicit enough. The authors should provide this information using tables or links. Furthermore, to be able to interpret Extended Figure 1b better, the authors should indicate the ranking of the strongest positive female/male curve peaks in the Figure.

5) Figure 1: The graphs on the right in Figure 1A and B should be supplemented with the information of the overall contribution of this age group to cancer incidence (in order to avoid an overestimation of the patients affected by this process).

6) In Figure 5, the authors report that Grpr overexpression increases lung colonization in mice after tailvein injection. Was this also observed for male mice? It would be important to show if Grpr expression takes away the differences between tumors in males and females. Also, the experiment from Figure 5 is not mentioned in the Material and Methods section.

Minor points:

- 1) Extended Figure 1a, c are redundant to Figure 1a, b, they should be combined.
- 2) What is the data source of Figure 1c (estradiol concentration over time)?
- 3) In Figure 1e and Extended Figure 1g, it would be useful to specify the genomic alterations in CDH1 and CCND1 (e.g. deleterious mutations).
- 4) Page 4: "Grpr overexpression in female melanomas was independent of the Tyr::Cre line used and other genetically engineered melanomas tumors (Extended Data Fig. 2c)."
The Figure legend requires a better explanation of the different mouse lines. Also, are the data only derived from female mice?
- 7) Figure 2c: "c, Heatmap clusters TCGA-SKCM samples based on predominant cell-state signatures: "pigmented," "SMC" (starved-like melanoma cells), "invasive," and "NCSC-like" (neural crest cell-like)."  Here, the reference of the publication first describing the signatures is needed.
- 8) What is the data source for the Kaplan Meier curves in Figure 2d/e and Extended Figure 2h/i?
- 9) Extended Figure 5B: Should be RC in the labeling under the x axis (not PD)

Typos and grammar errors:

- 1) Page 4: "The choice of using an NRAS over BRAF mutation was influenced by the slightly higher frequency of downregulation of CDH1 in NRAS melanoma compared to BRAF melanoma, as well as the lack of targeted therapy for an NRAS-mutated melanoma (Extended Data Fig. 1h).
- 2) Page 5: "Thus, efficient lung colonization and rapid growth after grafting support the cell-autonomous aggressiveness of Δ Ecad female Melanomas were established as cell lines in culture and consistent with the mouse primary melanomas, only Δ Ecad female melanoma cell lines produced GRPR"  should be two sentences.
- 3) Page 9: "As previously showed in Fig. 2j" (should be "shown")

Data presentation, availability and statistics:

- There are no objections to data presentation, availability and statistics. All points are sufficiently addressed by the authors.

Referee #2

(Remarks to the Author)

This is a very interesting and important manuscript and a praise to the authors for considering sex specific factors in their work, which could improve cancer management.

In their manuscript Raymond and colleagues explored the role of E-cadherin loss in the progression of sex-hormone dependent cancers. Providing in vivo and in vitro data, they uncovered a novel axis which involves GRPR/YAP1 in the molecular mechanism of estrogen response in different cancers.

Data are novel and it is of interest that the aim of the manuscript is to change the perspective of estrogen-dependent cancers.

Data and methodology are clearly presented with appropriate statistical analyses.

The flow of the reading should be improved, I would suggest to change the order of some figures, to first describe the effect of E-cadherin loss in lung metastasis promoted by GRPR (figures 1,2 and 5), followed by the role of estrogens in the activation of GRPR in both melanoma (figure 6) and breast cancer (ext fig.7) and finally concluding by depicting the molecular mechanism of GRPR in cell growth and invasion and the role of YAP1 (figures 3 and 4). In this manner, the explanation of the molecular mechanism with a complete signal cascade (from E-cadherin to YAP1) would be at the end of the manuscript and completely depicted in the scheme in figure 6m. This scheme could be improved.

To better describe the link between E-cadherin and GRPR in lung metastasis it would be interesting to determine whether Δ Ecad mouse melanoma cell lines would promote cell growth/invasion and inhibit apoptosis as described for GRPR cells (figure 3d-u), and whether the effect would be reverted upon knocking down GRPR in Δ Ecad cell lines. Moreover, authors used both male and female cell lines in the in vitro experiments described in figure 3: it would be interesting to engraft also Δ Ecad male tumours together with female tumours, as described in figure 3 and ext. figure 3.

Data in figure 4 describing the role of YAP1 should be completed at least with in vitro experiments performed in Δ Ecad cell lines and assess whether these cell lines present upregulation of YAP1 and whether this upregulation is GRPR dependent.

It would be also interesting to assess whether YAP1 knockdown in GRPR cell lines revert GRPR effect on invasion and cell growth.

Authors conclusion about Δ Ecad/ERS1/GRPR pathway and its feedback loop should be completed or at least discussed, including the possibility (as described in the literature for estrogen-dependent cancers) that also estrogens might downregulate the expression of Ecad.

To improve the clinical impact of the use of estrogen or GRPR inhibitors in Ecad-deficient tumours, it would be interesting to perform in vivo or in vitro experiments in Δ Ecad mice or cell lines treated with RC or ICI to assess whether there is any effect on their invasive capacity.

Minor points:

In keywords add sex;

Referee #3

(Remarks to the Author)

This study tackles the important question of sex-bias in tumor progression arises, especially for tumors that are not conventionally thought to be sex-hormone dependent. The work is motivated by the authors' observation that there is a disproportionate incidence of cancers on pre-menopausal women. The authors test this experimentally for melanoma, which is disproportionately affected in their human data and where they show that lung metastasis in mouse models is increased in E-cadherin KO tumors from female animals. They propose a model where (as I understand it) E-cadherin dysfunction upregulates b-catenin signaling, which increases expression of GRP-receptor via upregulation of the estrogen receptor. The latter would presumably be key to conferring the sex-preference for premenopausal females. Of note, activation of GRPR signaling downregulated E-cadherin expression, suggesting that a positive feedback loop would allow this pathway to be amplified by dysfunction at various steps. These are interesting findings, but they also raise a number of questions in my mind.

Major questions.

1. Preclinical significance. Although a wide range of approaches are used, many key experiments are performed using tumor cell lines, either in vitro or as xenografts, including cell lines where GRPR is overexpressed. While this is valuable to probe mechanism, I think that to reinforce the relevance of these findings, key results and predictions would be stronger if they could be tested in the authors' mouse model. In particular, the model predicts that a) GRPR would be differentially upregulated in lung metastases in female mice; and b) metastasis would be reduced by i) inhibiting GRP-GRPR signaling and ii) inhibiting estrogen signaling (either at the receptor or by ovariectomy). I appreciate that the authors have used xenograft models as a more facile way to test this question, but the application of their model for cancer would be stronger if it were done in a more complex genetic model.

2. The causal role of b-catenin signaling. The authors propose that cadherin dysfunction leads to an upregulation of b-catenin/Wnt signaling as a key step in their pathway. The experimental evidence for this is not direct - based on E-cadherin inactivation and APC inactivation being associated with increased ESR expression. Although E-cadherin dysfunction has often been postulated to increase b-catenin/Wnt signaling, this is not always the case (e.g. Christofori lab: Herzig Oncogene, 2007 26: 2290) and both E-cadherin and APC dysfunction have other cellular effects. To make a clear case for b-catenin, the authors need to directly inhibit b-catenin signaling in their experiments where ESR, GRP are upregulated, with the predictions that i) this upregulation will be abrogated and ii) the predicted functional consequences will also be corrected. An ancillary question here is if b-catenin signaling is upregulated by E-cadherin dysfunction in their models, what mechanism would allow this to happen. As noted above, E-cadherin loss does not always lead to an increase in the cytosolic b-catenin pool, unless there is another defect e.g. APC or destruction complex dysfunction. This is probably a subsidiary question, but it is relevant for understanding the precise mechanism.

3. The role of E-cadherin dysfunction. The authors' model (as outlined in the final cartoon) identifies E-cadherin dysfunction as a key upstream step. Further, they present data to suggest that genomic changes in E-cadherin are more frequent in young women (compared with either older women or men) and E-cadherin transcript expression is also reduced in this group. Is the genomic change contributory? If so, a key issue is how genomic changes in E-cadherin become more frequent in young women. Would they not be expected to be found also in older women, or could the frequency be biased by cancer deaths in younger women? The authors show evidence that ESR-GRP signaling can downregulate E-cadherin (considered further below). But if this were operational, how could it cause a genomic alteration?

4. The role of positive feedback. The authors postulate that there is a positive feedback pathway, such as enhanced ER-GRPR signaling would downregulate E-cadherin. This makes the model more complex, but not be surprising (since feedback is so prevalent). However, the evidence for feedback is based on short term studies using cell lines expressing exogenous GRPR. This is a useful test of possibility, however it needs to be tested in a more realistic model. I appreciate that this is difficult to do in their genetic model, but it could potentially be done in xenograft models. For example, they could test if overactivation of b-catenin in E-cad-positive cells upregulates ESR-GRPR signaling; and whether this is upregulation of ESR-GRPR causes E-cadherin to be downregulated.

Version 2:

Reviewer comments:

Referee #1

(Remarks to the Author)

In their revised version of the manuscript "Targeting GRPR for sex hormone-dependent cancer after loss of E-cadherin", the authors have conscientiously addressed all issues raised by the reviewers. They have added a substantial number of experiments, confirming their hypothesis and further strengthening their findings.

All questions raised by me were sufficiently addressed, but I would like to specifically comment on selected points in the paper to provide some context:

R1Q1: The authors could satisfyingly show the regulation of the E-cadherin-ESR1-GRPR axis on protein level. For GRPR, protein detection by western blot was not possible (obviously a known problem in the field), but the authors showed convincingly using several tools that GRPR function was regulated as suggested.

Also, the authors showed that the GRPR expression in the overexpressers is in a similar range as endogenous GRPR.

R1Q4: The revised manuscript now contains a detailed description of the search approach that resulted in the identification of CDH1 as possible key factor in gender specific cancer risk. Their new, updated analyses resulted in a slight increase of candidate genes (now also including KRAS and BRAF), but without altering the general statement and the specific relevance of CDH1 in this patient age group.

R1Q9: In the previous version of the paper, the authors have included a figure that shows a significantly increased number of CDH1 mutations in "hormone dependent" compared to "non hormone dependent" cancers. As the authors derive the genetic information from public databases, reliable information about the functional consequence of the mutations is missing. As this hampers the interpretation of the figure, the authors have now removed it from the revised manuscript. I support this decision, as I believe that it is not relevant for the proposed mechanism.

In my view, it is sufficient to mention "alterations of ECAD by mutations, methylation, and/or EMT" (like written in the Discussion section, page 13).

R2Q5: The authors have provided additional information to show that GRPR-mediated murine melanoma invasiveness is mediated by YAP1 and TAZ1, using siRNA. The effect of GRPR signaling on YAP1 target genes was further confirmed in different mouse and human cell lines, using GRP and an GRPR inhibitors.

R3Q2: The involvement of beta catenin in the proposed signaling process was addressed using a pharmacological beta catenin inhibitor (iCRT3), demonstrating that ESR1 is downregulated in presence of the inhibitor. Also, GRP-induced invasiveness was decreased after beta catenin inhibition or knockdown. This shows that beta catenin is integral part of the E-Cad-beta-cat-ESR1-GRPR signaling axis.

In conclusion, the authors were able further substantiate their findings.

There is only one minor point, where I would recommend different wording:

Lines 320-322: „Furthermore, downregulation of ESR1 upregulated ECAD (Fig. 6k). Finally, we showed that regulation of ER α by E-cadherin is reciprocal (Fig. 6k).”

I would suggest a different wording to avoid misinterpretation:

Furthermore, downregulation of ESR1 upregulated ECAD, showing that regulation of ER α by E-cadherin is reciprocal (Fig. 6k).

In summary, this is a highly interesting manuscript that makes a large contribution to the understanding of cancer biology in women and offers therapeutic avenues for improving female health. I congratulate the authors to their excellent work.

Referee #2

(Remarks to the Author)

Authors have done an excellent job in revising the manuscript, taking into account all criticisms. I believe the manuscript can be accepted.

Referee #3

(Remarks to the Author)

The authors have performed an extensive revision of this study, incorporating substantial new data that very reasonably address the points raised in my earlier review. I think that this is an important study.

Referees' comments:

Referee #1 (Remarks to the Author):

The manuscript „Targeting GRPR for sex hormone-dependent cancer after loss of Ecadherin“ by Larue and colleagues describes an unexpected role of estrogen receptor signaling in cancer prevalence and metastasis. At the beginning of their paper, the authors separately consider overall tumor incidence in men and women (where overall incidence is reportedly higher in men) and age-related tumor incidence. The latter analysis reveals a striking increase in tumor incidence in females during their reproductive age, which generally correlates with high estradiol levels. This was observed for several tumor types including melanoma. In the following, the authors show – in most parts using melanoma as model tumor - that the surface molecule E-cadherin (CDH1) is functionally linked to estrogen receptor 1 and expression of pro-metastatic GRPR.

Specifically, this link is supported by the following observations:

- 1) melanomas from females <50 years have increased genomic alterations in CDH1/Ecad as well as lower CDH1 expression,
- 2) Ecad knockout in an NRAS-driven mouse melanoma model led to the development of lung metastases specifically in female mice
- 3) Ecad knockout in females causes a strong expression of Gastrin-Releasing Peptide Receptor (Grpr), which shows pro-tumorigenic and metastatic abilities itself, e.g. after teilvein injection of murine or human melanoma cells overexpressing exogenous Grpr. Importantly, pharmacological inhibition of Grpr strongly inhibited lung colonization of these cells.
- 4) Grpr induces a YAP-associated pro-metastatic gene program in mouse melanoma cells
- 5) Ecad downregulation leads to the upregulation of Esr1 (probably mediated by beta catenin), which mediates Grpr upregulation. Grpr activation (e.g. by GRP) blocks CDH1 expression, thus forming a loop.

This study is generally of high interest across disciplines. However, most data were generated in mouse models and the deregulations of Grpr and Esr1 are based on RNA expression data.

Some points still need to be addressed, as detailed below:

Major points:

Please note that the figure numbering has been updated. All responses refer exclusively to the new numbering. We included the term “new” before figures in the response to ensure clarity and prevent confusion.

R1.Q1 (Reviewer 1 – Question 1)

The Ecad  Esr1  Grpr loop, as shown in Figure 6m, should be demonstrated on protein level in human and murine cells. Furthermore, it is important to compare the protein levels of endogenous upregulation of Esr1 and Grpr (as caused by Ecad knockdown) with that of exogenous Grpr overexpression (that was used in Figure 3, Extended data Figure 3 and Figure 5).

R1.A1 (Reviewer 1 – Answer 1)

We appreciate the reviewer's question, as it raises an important issue.

In the new version of the ms, we show the existence of a Ecad → Esr1 → Grpr regulatory loop at the protein level in both human and murine melanoma cells. First, we demonstrate that E-cadherin loss leads to increased ERα expression, establishing a strong inverse correlation between these proteins. This effect is consistently observed across multiple experimental models, including siRNA knockdown and CRISPR-Cas9 knockout approaches. Next, we show that ERα regulates GRPR expression. While we were unable to detect GRPR protein using Western blot due to antibody limitations, functional assays confirm that ERα activation leads to increased GRPR expression, whereas ERα inhibition reduces it. Additionally, GRPR signaling activity was validated through IP1 assays, further confirming the functional relevance of this pathway. We also establish that GRPR activation suppresses E-cadherin expression, completing a positive feedback loop in which estrogens enhance ERα activity, further reinforcing CDH1 inhibition. Finally, we directly compare endogenous GRPR upregulation (resulting from E-cadherin loss) with exogenous GRPR overexpression. Our data show that exogenously expressed GRPR levels are comparable to or lower than those observed endogenously. Importantly, GRPR functional activity, as measured by CCN1/CYR61 induction, remains consistent between these models, demonstrating the robustness of our findings. Together, our results provide strong evidence for the Ecad → Esr1 → Grpr loop and its functional consequences in melanoma.

- o - 0 - o -

a) Ecad  Esr1

E-cadherin and ERα (encoded by the *ESR1* gene) antibodies are effective in western blots for both mouse and human samples. There are three parts in this answer noted a1-a3.

a.1) Protein expression of Ecad and ERα proteins

Answer:

We observed a clear anticorrelation between E-cadherin, ERα protein levels, and sex in mouse melanoma cell lines that we established in culture. This finding is now presented in new Fig. 6g.

Figure legend

Western blot analysis of E-cadherin and ERα protein levels in the indicated mouse melanoma cell lines, with Actin as a loading control. The presented data are from a representative experiment out of three independent replicates, with additional replicates available in the supplementary material.

Main text

This information is now included in the text.

After the sentence “None of the other sex hormone receptors (*Esr2*, *Gper1*, *Ar*, and *Pgr*) were expressed in the murine melanoma cell lines and only the Δ Ecad female melanoma cell lines exhibited expression of both Grpr and *Esr1* (new Fig. 6e,f).”, we included “Consistently, ER α levels were detected in female melanoma cell lines that lack E-cadherin (new Fig. 6g).”

a.2) Reciprocal regulation of ECAD and ER α at the protein level

Answer:

We repressed E-cadherin expression in 1014 (female Ecad) mouse melanoma cell line and observed an induction of ER α levels. These results are now shown in new Fig. 6k.

Figure legend

k, Western blot analysis of E-cadherin and ER α protein levels in mouse 1014 melanoma cells 48 hours after knockdown with siScr (control), siCdh1 (targeting E-cadherin), or siESR1 (targeting ER α). Actin served as a loading control. The displayed data represent a selected experiment from four independent replicates, with full data provided in the supplementary material. Histograms show the quantification of protein levels from these four experiments, presented as mean \pm SD. Statistical analysis used Mann-Whitney from normalized values.

We also conducted similar experiments in Daju human melanoma cell lines (female Ecad expressing cells). CDH1 reduction led to an increase of ER α protein levels (new Extended Data Fig. 6q).

Figure legend

q, RT-qPCR of *CDH1* levels in human melanoma cells treated with siCDH1. Western blot analysis of ER α protein levels in human melanoma cells after CDH1 knockdown (three biological

replicates). Relative ER α levels are shown quantified in the histogram. Statistical significance assessed by an unpaired t-test. Data are presented as mean \pm SD.

Main text

See at the end of a.3)

a.3)

Answer:

We knocked out E-cadherin in the 888mel female human melanoma cell line using Crispr-Cas9 (new Extended Data Fig. 7h). Please remember that we analyzed a mixed population of cells, including both knockout and non-knockout cells, and observed an induction of ER α at both the mRNA (new Extended Data Fig. 7i) and protein (new Extended Data Fig. 7j) levels. Additionally, we detected an increase in GRPR expression at the mRNA level (new Extended Data Fig. 7i). Despite extensive testing, we were unable to identify any GRPR antibodies suitable for Western blot analysis. We evaluated antibodies targeting different regions of the protein, including the N-terminus (Santa Cruz, sc-398549), the second extracellular loop (ORIGENE/Acris, SP4337P #331), the third cytoplasmic domain (Acris, SP4708P), and the transmembrane domains (Abcam, ab39833). Unfortunately, none performed well in Western blot; however, SP4337P proved effective in immunohistochemistry on human samples. The difficulty in obtaining high-quality antibodies against GPCRs is well documented, as these transmembrane proteins possess seven helical domains, making their structure highly intricate within the membrane. This complexity limits the exposure of accessible epitopes for antibody recognition. Additionally, GPCRs adopt multiple conformations—active, inactive, or ligand-bound—further complicating the development of antibodies that specifically target a single functional state.

These results are now presented in new Extended Data Fig. 7h-j.

Legend

(h) Schematic representation of the CDH1 knockout strategy. Exons 6 to 10 were deleted using two distinct guide RNAs (gRNA1 and gRNA2).

(i) mRNA expression levels of *CDH1*, *ESR1*, and *GRPR* in the 888-Mel cell population transfected with gRNA targeting *CDH1*.

(j) Western blot analysis (left) and quantification (right) of ER α and E-cadherin (ECAD) protein levels in the 888Mel cell population transfected with gRNA targeting *CDH1*. β -Actin and Vinculin (Vinc.) were used as loading controls. Statistical analysis was performed using a *t*-test.

Main text

We added the following statement at the end of the paragraph: "At the protein level, we confirmed that knockdown/knockout of CDH1 in both mouse and human melanoma cell lines leads to upregulation of ER α (new Fig. 6k and new Extended Data Fig. 6q and 7h-j). Furthermore, downregulation of ESR1 upregulates ECAD (new Fig. 6k). Finally, we showed that regulation of ER α by E-cadherin is reciprocal, as knockdown of ESR1 induces ECAD levels (new Fig. 6k)."

b) ERa -> Grpr

Answer:

Unfortunately, antibodies against GRPR do not function in western blot analysis, making it impossible to directly address the reviewer’s comment on this specific point. However, we have already showed that ectopic expression of ER α in both mouse and human melanoma cells increased GRPR expression, while ER α repression either by siRNA in female mouse and human melanoma cells or through treatment with Fulvestrant/ICI 182,780 reduces GRPR expression (new Fig. 6h,i and new Extended Data Fig. 7e,f,k). While we could not assess the levels of GRPR protein directly by western blot, we did look at GRPR signaling activity (and by extension, GRPR protein levels) by making use of an IP1 assay. This assay measures the activation of G α q signaling downstream of GRPR activation. In 1057 (female Δ Ecad) cells, treatment with GRP increases IP1 levels, whereas ICI 182,780 reduces them (new Extended Data Fig. 5a-d). Specifically, in 1057 (female Δ Ecad) cell lines, ICI reduces the induction of IP1, a reporter of G α q activity, by GRP. As expected, in 1014 (female Ecad), 1062 (male Δ Ecad), and 1181 (male Ecad) cell lines, which do not express Grpr, there is no change in IP1 levels following treatment with GRP, ICI 182,780, or GRP + ICI 182,780.

Legend

a-f, IP1 levels, a readout of G α q/11 activation, were measured in mouse melanoma cell lines following stimulation with 10 nM GRP, with or without treatment with the GRPR inhibitor RC-3095 (a-f) or pre-treatment with the ER α inhibitor Fulvestrant -ICI- (a-d). Statistical differences were analyzed using ANOVA followed by Holm-Šidák’s multiple comparison test.

Main text

In the text, after the sentence : “Treatment with the ER α agonist 17 β -estradiol (E2) increased Grpr expression, while treatment with the ER α degrader ICI 182,780 (ICI) decreased its expression (new Extended Data Fig. 7k).” We included “This effect on Grpr expression was associated with a decrease in its activity, as shown by the reduced IP1 induction by ICI treatment (new Extended Data Fig. 5a-d).”

c) Grpr -> Ecad

Answer

We previously showed that *CDH1* mRNA levels were downregulated under GRP stimulation in cells expressing GRPR (see Fig. 6m). To further investigate, we assessed Ecad protein levels in 501mel cell lines, with or without GRPR expression, in the presence or absence of GRP.

ECAD is lower in cells expressing GRPR upon GRP treatment. These results are shown in new Fig. 6l.

Figure legend

I, Western blot analysis of E-cadherin (ECAD) expression in 501mel cells, with or without GRPR expression, in the presence or absence of 10 nM GRP. Actin served as a loading control. Statistical analysis was performed using data from five independent experiments, with the Mann-Whitney test applied to normalized values.

Main text

“Finally, the CDH1/ESR1/GRPR axis operates through a positive feedback loop, where GRPR activation by GRP repressed CDH1 expression (new Fig. 6l,m) and importantly E2 reinforced this loop by stimulating ER α activity and further inhibiting CDH1 expression (new Fig. 6k and new Extended Data Fig. 7k).”

- o - 0 - o -

d) Question: Furthermore, it is important to compare the protein levels of endogenous upregulation of ESR1 and Grpr (as caused by Ecad knockdown) with that of exogenous Grpr overexpression (that was used in Figure 3, Extended data Figure 3 and Figure 5).

Answer:

We evaluated both the endogenous and exogenous GRPR levels in mouse and human melanoma cell lines to validate our cellular models. Additionally, we assessed GRPR activity by measuring CCN1/CYR61 levels as a readout of GRPR activity (our experiments and (Yu et al. 2012).

First, in mouse cell lines that are all isogenic (C57BL/6J), the expression levels of exogenous Grpr in cells lacking endogenous Grpr (1014 and 1181) are comparable to or even lower than the endogenous Grpr expression levels in the 1057 Grpr-expressing cells. Furthermore, the induction of *Ccn1/Cyr61* by GRP was similar in cells endogenously or exogenously expressing *Grpr*. We conducted similar experiments in human melanoma cell lines. Although GRPR mRNA expression was higher in cells expressing exogenous GRPR, CCN1 induction remained consistent across cell lines expressing GRPR mRNA. These data were obtained via RNA-seq, with similar results confirmed by RT-qPCR. These findings are not surprising, as it is well known in the GPCR field that receptor activity often does not always correlate directly with expression levels. Once expression exceeds a certain threshold, activity remains constant (Raymond et al. 2022).

Figure legend

j,k, GRPR expression in mouse (c) and human (d) melanoma cell lines, normalized in TPM and log-transformed for visualization. Ct = control – cells transfected with an empty expression vector. **l,m**, Fold change in *CCN1* expression after 4 hours of stimulation with 10 nM GRP. Fold changes were calculated for each biological replicate based on TPM-normalized *CCN1* expression.

Main text

We have now added the following in the text “Notably, the 1181 and 501mel cell lines ectopically expressing GRPR did not overexpress GRPR but exhibited expression levels comparable to cell lines that endogenously express this protein (e.g., mouse 1057, and human Dauv-1, MDA-MB-435S) (Extended Data Fig. 3j-m).”

- o - 0 - o -

R1.Q2. According to public databases such as the proteinatlas, *Grpr* expression is hardly expressed outside the gastrointestinal tract. Also, expression in tumor cell lines is extremely weak in most cell lines except in breast cancer. How can the authors reconcile this with their own observations?

R1.A2.

Answer:

The latest update (February 2025) of Protein Atlas (<https://www.proteinatlas.org/ENSG00000126010-GRPR>) indicates that GRPR mRNA is primarily expressed in the pancreas, with lower levels detected in the gastrointestinal tract, breast, and epididymis.

The referee may be questioning the apparent absence of GRPR in the skin, where melanocytes reside. It is crucial to acknowledge that bulk tissue analyses may not detect GRPR expression in specific cell subpopulations. Melanocytes are a minority in the skin. In the basal layer of the epidermis, their ratio is approximately 1 melanocyte for every 5 to 10 keratinocytes, and they are estimated to constitute only 1–3% of all epidermal cells. Given the variability in dermis and hypodermis thickness across different body regions, melanocytes likely represent no more than 0.5% of total skin cells. This low abundance could explain why GRPR expression in melanocytes remains undetected in large-scale transcriptomic datasets.

Regarding protein expression, only one antibody (HPA069604) was used in the Protein Atlas, reporting its localization at the plasma membrane in the PC-3 human cell line. However, it has never been cited/used in any publication and has been discontinued by the company. Given this limitation, we must consider that no reliable information is currently available on GRPR protein expression in human tissue.

In human cancers, data from the TCGA indicate that GRPR is detected at the RNA level in bulk RNA-seq (see table below). These data suggest that GRPR is expressed in a significant proportion of tumors. Specifically, 13.2% of all tumors and 8% of SKCM exhibited GRPR expression levels above 1 TPM -level equivalent to the level of 1 transcript per cell- while 50.2% of all tumors and 69.8% of SKCM tumors exceed 0.1 TPM, respectively. The relationship between 1 TPM and the number of transcripts per cell is not well appreciated. It has been previously showed (Mortazavi et al. 2008). Thus, the reported low expression of GRPR is relative.

Cancer	Nb cases	threshold >1TPM		threshold >0.1TPM	
		Nb	%	Nb	%
ACC	79	11	13.9	39	49.4
BLCA	427	5	1.2	83	19.4
BRCA	1215	714	58.8	1039	85.5
CESC	309	6	1.9	48	15.5
CHOL	45	1	2.2	13	28.9
COAD	328	45	13.7	170	51.8
DBLC	48	0	0.0	8	16.7
WSCA	196	16	8.2	89	45.4
GBM	174	6	3.4	135	77.6
HNSC	566	5	0.9	115	20.3
KICH	566	5	0.9	115	20.3
KIRC	606	30	5.0	353	58.3
LAML	173	1	0.6	114	65.9
LGG	534	8	1.5	326	61.0
LIHC	424	47	11.1	158	37.3
LUAD	576	10	1.7	180	31.3
LUSC	553	25	4.5	213	38.5
MESO	87	4	4.6	18	20.7
OV	309	12	3.9	138	44.7
PAAD	183	14	7.7	112	61.2
PCPG	187	31	16.6	112	59.9
PRAD	550	216	39.3	492	89.5
READ	165	15	9.1	69	41.8
SARC	265	15	5.7	148	55.8
STAD	450	70	15.6	269	59.8
TGCT	166	27	16.3	105	63.3
THCA	572	4	0.7	199	34.8
THYM	122	1	0.8	8	6.6

UCEC	201	21	10.4	128	63.7
UCS	57	7	12.3	34	59.6
UVM	80	0	0.0	7	8.8
SKCM	473	38	8.0	330	69.8
All cancer	10686	1410	13,2%	5367	50,2%
All cancer w/o BRCA	9471	696	7,3%	4328	45,7%

Legend to the table

Number and percentage of cases for each cancer of the TCGA database with an expression of GRPR superior to 1 TPM and to 0.1 TPM. 1 TPM is considered as the equivalent of each cell having one transcript for mRNA of the length of GRPR mRNA (~1kb). By analogy, 0.1 TPM can be considered as 10% of the tumour cells expressing GRPR.

Main text

We added information about GRPR expression in the text. After the sentence “4) the presence of GRP, the natural agonist of GRPR, in the lungs of both human and mice (new Extended Data Fig. 2k-m).” we included “Importantly, GRPR mRNA is found in a large proportion of tumors, including melanoma (Raymond et al. 2022). (Raymond et al., 2022).

- o - 0 - o -

R1.Q3. The age group of interest (approx. 20-50 yrs) is also a group in which the occurrence of heritable cancers is proportionately more likely than in older patients. Thus, the data might point at the possibility that in humans with a cancer predisposition, young women are more likely to develop cancer due to estrogen alterations. This should be discussed. Could the authors provide information about the contribution of familial cancer in males versus females in this age group?

R1.A3.

Answer:

The reviewer raises an important and exciting question. In this study, we simultaneously analyzed concomitantly germinal and somatic alterations.

Unfortunately, research on the influence of sex in familial melanoma remains scarce. Familial melanoma is estimated to represent 5-15% of total melanoma cases (Bertrand et al. 2020). The primary predisposition genes associated with cutaneous melanoma include *CDKN2A*, *CDK4*, *MITF*, *MC1R*, *POT1*, and *BRCA2*.

To our knowledge, no studies specifically examined the age and sex distribution of familial melanoma. However, some partial insights exist. For most melanoma-associated variants (e.g., *CDKN2A*, *CDK4*), the overall incidence does not significantly differ between males and females (D'Ecclesiis et al. 2021; Landi et al. 2020). To date, only certain *MC1R* red hair color variants have been linked to a higher risk of melanoma in females (Wendt et al. 2018).

- o - 0 - o -

R1.Q4. In my opinion, Figure 1d plays an important role, as it provides the connection to the following studies in the manuscript. However, it is not quite clear, where the cancer associated genes” characterizing the chosen cancer entities in Figure 1d (top three female biased cancers

(melanoma, gastric, and thyroid), breast cancer, and those associated with the main estrogen receptor type 1) come from. The information shared in the Materials and Methods are not explicit enough. The authors should provide this information using tables or links. Furthermore, to be able to interpret Extended data Figure 1b better, the authors should indicate the ranking of the strongest positive female/male curve peaks in the Figure.

R1.A4.

Answer:

We thank the reviewer for highlighting the lack of precision in our previous description. In response, we have generated an updated Venn diagram based on the latest information and created a new table (Supplementary Table 2) that includes the list of genes for each component of the Venn diagram.

Main text

In the methods sections, we have revised and expanded the paragraph “Intersection of genes associated with cancers with positive female/male curve peak and estrogen-response” by specifying the pathway numbers for each cancer and providing further details on our search approach. We replaced the original sentences with : “List of genes associated with Melanoma (73, hsa05218), Gastric cancer (150, hsa05226), and Thyroid cancer (37, hsa05216) were retrieved from KEGG. The breast cancer associated genes (172) list were downloaded from WikiPathway (WP1984). ESR1-associated genes (317) were extracted from ESR1 associated genes from StringDB using the following parameter: search “CDH1” in human, setting = “high confidence” and max number of interactor = 317. The lists were then intersected using a Venn diagram (<https://bioinformatics.psb.ugent.be/webtools/Venn/>).”

Ideally, we should have used bladder cancer instead of thyroid cancer for the Venn diagram, as bladder was the third most female-biased cancer, whereas thyroid was the fourth (please see Supplementary Table 1). Unfortunately, we could not find a validated list of bladder cancer-associated genes in the big data/literature. However, multiple studies (Bringuier et al. 1993)(Shimazui et al. 1996) suggest an association between E-cadherin and bladder cancer, leading us to expect similar results.

Using the latest data, four genes (*CDH1*, *CCND1*, *BRAF*, and *KRAS*) emerged in the new gene table, differing from our previous findings. However, upon examining their expression over time, only *CDH1* levels were lower in young female melanoma patients (new Fig. 1e). We have included the updated Venn diagram (new Fig. 1d) and incorporated this information into the text.

d Cancer-associated genes

Figure Legend

new Fig. 1e) Expression of *CDH1*, *CCND1*, *BRAF*, and *KRAS* in human melanoma (TCGA database) stratified by patient sex and age. TPM = transcript per million.

- o - 0 - o -

As requested by the referee for better interpretation of Extended Data Fig. 1b, we ranked cancers based on the peak values of the positive female/male curve. Female-biased and male-biased cancers were ranked separately. These rankings are available in Extended Data Fig. 1c and Supplementary Table 1.

Additionally, we modified new Extended Data Fig. 1c by replacing the Table 1b with a diagram (1c), as shown below.

Figure legend 1c

We replaced the following text in the figure legend of Extended Figure 1c “Cancers with a positive women/men curve peak and $PV > 0.20$ are highlighted in red, cancers with a negative curve peak and $PV < -0.20$ are highlighted in blue and cancers without curve peak and $-0.20 \geq PV \leq 0.20$ are highlighted in green.” with this text: “Cancers with a positive female/male curve peak and $PV > 0.20$ are highlighted in red, cancers with a negative curve peak and $PV < -0.20$ are highlighted in blue and cancers without a curve peak and $-0.20 \geq PV \leq 0.20$ are colored in white.”

- o - 0 - o -

R1.Q5. Figure 1: The graphs on the right in Figure 1A and B should be supplemented with the information of the overall contribution of this age group to cancer incidence (in order to avoid an overestimation of the patients affected by this process).

R1.A5.

Answer

We understand the reviewer's concern and have addressed it by including information of the overall contribution of this age group to cancer incidence in new Extended Data Fig. 1a (all cancers) and new Extended Data Fig. 1b (melanoma).

Legend: Crude incidence rates (cases/100'000) for all cancers (a) and cutaneous malignant melanoma (b) in the world population in 2020. Data are represented for each sex and as the average of both sexes.

As seen in both panels, the number of cases is significantly higher in females between puberty and menopause.

- o - 0 - o -

R1.Q6. In Figure 5, the authors report that Grpr overexpression increases lung colonization in mice after tail vein injection. Was this also observed for male mice? It would be important to show if Grpr expression takes away the differences between tumors in males and females. Also, the experiment from Figure 5 is not mentioned in the Material and Methods section.

R1.A6.

When we injected male 1181 mouse melanoma cells, either expressing or not ectopic GRPR, into female mice in immunocompetent mice, no tumors develop. This is a well-known phenomenon, often attributed to the expression of genes located on the Y chromosome, which can trigger immune rejection.

In contrast, when we injected male 501mel human melanoma cells, with or without ectopic GRPR, into male or female NSG mice, lung metastases developed predominantly in male and female mice injected with 501mel cells expressing ectopic GRPR. This outcome is expected, as GRPR expression is driven by the CMV promoter, ensuring expression consistent expression across both sexes, independent of estrogen levels.

We have incorporated this information in the main text and have also updated the materials and methods section related to the former Figure 5 (new Figure 3).

In the legend of Figure 3, “GRPR Expression and Its Activation Drive Lung Melanoma Metastasis” we included the sex of the cells. **a,b**, Representative lung images captured 30 days post-injection of 5×10^5 male melanoma cells (1181) lacking Grpr or expressing exogenous Grpr (+Grpr) into male C57BL/6J mice tail veins. Scale bar = 2 mm. **c**, Injection of 5×10^5 male 501mel melanoma cells, either expressing GRPR (+GRPR) or not into the tail veins of male C57BL/6J NSG mice.”

- o - 0 - o -

Minor points:

R1.Q7. Extended Figure 1a, c are redundant to Figure 1a, b, they should be combined.

R1.A7.

The reviewer is correct. We have replaced previous Extended data Fig. 1c (melanoma), by gastric cancer (new Extended Data Fig. 1d). Gastric cancer is the second most female-biased tumor after melanoma, which rank first.

- o - 0 - o -

R1.Q8. What is the data source of Figure 1c (estradiol concentration over time)?

R1.A8.

We thank the reviewer for highlighting this missing information. We have added the following to the methods section:

Sexual hormone concentration. Plasma testosterone and estradiol levels in males and females, categorized age, were extracted from <https://doi.org/10.1101/2024.10.07.24315000>.

- o - 0 - o -

R1.Q9. In Figure 1e and Extended Figure 1g, it would be useful to specify the genomic alterations in CDH1 and CCND1 (e.g. deleterious mutations).

R1.A9.

We analyzed the percentage of driver mutations in cancers based on their hormone-dependence status. Driver events -including mutations, fusions, and copy number alterations (CNA)- were identified using the OncoKB and CancerHotspots databases. As shown in the graphs below, CDH1 exhibits a statistically significant difference, whereas CCND1, BRAF, and KRAS do not.

If needed, we can provide an XLS file containing all relevant data. Genetic alterations in *CDH1*, *CCND1*, *BRAF*, and *KRAS* were extracted from all non-mixed cancer studies available in cBioPortal (as of February 18, 2025) using the OQL filter “DRIVER” to select only oncogenic or suspected oncogenic mutations. Data were compiled by cancer type, and alteration percentages were calculated for each gene of interest. Hormone-dependency status was determined through an extensive literature review.

Since the functional roles of these mutations were predicted but not functionally assessed (e.g., LOF, GOF, DN, DA) in our models, we have removed this information in the current version. However, we are ready to reinstate it if the reviewer and/or editor considers it relevant.

- o - 0 - o -

R1.Q10. Page 4: “Grpr overexpression in female melanomas was independent of the Tyr::Cre line used and other genetically engineered melanomas tumors (Extended Data Fig. 2c).” The Figure legend requires a better explanation of the different mouse lines. Also, are the data only derived from female mice?

R1.A10.

We thank the reviewer for pointing this out. Indeed this information was missing. To clarify, unless specified, all mice used in Extended Data Fig. 2c are female. In the Methods section, under the “Mice” paragraph, we have now added references for the *Cdkn2a*^{tm1Rdp} (*Ink4a*) and *Pten*^{tm1Hwu} mouse models, which were previously omitted (Lesche et al. 2002; Serrano et al. 1996).

Additionally, in the Extended Data Fig. 2d legend, we updated the description to include the following: “Grpr mRNA levels in female melanomas from various Tyr::NRAS^{Q61K} female transgenic genetic backgrounds. Tyr::CreA is located on the X chromosome (denoted by 'A'), while Tyr::CreB is located on an autosome (denoted by 'B'). '-' represents the absence of the Cre gene in the genome. *Ecad*^{+/+} is indicated by '+', *Ecad*^{F/F} by '-', *Ink4a* ^{+/+} by '+', and *Ink4a* ^{+/-} by '-'. Similarly, *Pten*^{+/+} is represented by '+', and *Pten*^{F/+} by '-'.”

- o - 0 - o -

R1.Q11. Figure 2c: “c, Heatmap clusters TCGA-SKCM samples based on predominant cell-state signatures: “pigmented,” “SMC” (starved-like melanoma cells), “invasive,” and “NCSC-like” (neural crest cell-like).”  Here, the reference of the publication first describing the signatures is needed.

R1.A11.

We have now included the Rambow reference in the Materials and Methods section, under the “Cancer genomic and transcriptomic data mining” paragraph.

Specifically, we added the reference “(Rambow et al. 2018)” to the following sentence “The pigmentation state was defined by the expression of MITF, MLANA, TRPM1, DCT, and TYR; the SMC phenotype by the expression of CD36, DLX5, IP6K3, PAX3, and TRIM67; the invasive state by the expression of AXL, CYR61, TCF4, LOXL2, TNC, and WNT5A, and the NCSC-like phenotypic state by the expression of AQP1, GFRA2, L1XAM, NGFR, SLC22A17, and TMEM176B”.

- o - 0 - o -

R1.Q12. What is the data source for the Kaplan Meier curves in Figure 2d/e and Extended Figure 2h/i?

R1.A12.

We used data from TCGA-SKCM, which is now included in the legends of new Fig. 2d,e and new Extended Data Fig. 2i,j.

We updated the new Fig. 2d,e legend “Kaplan-Meier survival curves for Overall Survival (d) and Progression-Free Survival (e) of women samples from TCGA-SKCM categorized by GRPR expression (low/absent ≤ 0.1 TPM and expressed > 1 TPM).”

We updated the new Extended Data Fig. 2i,j “Overall Survival (OS) (i) and Progression-Free Survival (PFS) (j) Kaplan-Meier curves for TCGA-SKCM based on GRPR expression (low/absent ≤ 0.1 TPM and expressed > 1 TPM).”

- o - 0 - o -

R1.Q13. Extended Figure 5B: Should be RC in the labeling under the x axis (not PD)

R1.A13.

This error has now been fixed.

Typos and grammar errors:

- o - 0 - o -

R1.Q14. Page 4: “The choice of using an NRAS over BRAF mutation was influenced by the slightly higher frequency of downregulation of CDH1 in NRAS melanoma compared to BRAF melanoma, as well as the lack of targeted therapy for an NRAS-mutated melanoma (Extended Data Fig. 1h).

R1.A14.

We revised the sentence to “The decision to use an NRAS mutation instead of a BRAF mutation was based on the slightly higher occurrence of CDH1 downregulation in NRAS-mutant melanoma compared to BRAF-mutant melanoma, as well as the lack of targeted therapy for NRAS-mutated melanoma (new Extended Data Fig. 1h).”

- o - 0 - o -

R1.Q15. Page 5: “Thus, efficient lung colonization and rapid growth after grafting support the cell autonomous aggressiveness of Δ Ecad female Melanomas were established as cell lines in culture and consistent with the mouse primary melanomas, only Δ Ecad female melanoma cell lines produced GRPR”  should be two sentences.

R1.A15.

We revised the sentence into two for clarification: “Efficient lung colonization and rapid post-grafting growth confirm the cell-autonomous aggressiveness of Δ Ecad female melanomas. Cell lines were established from Δ Ecad female melanomas and consistent with the primary mouse melanomas, only these Δ Ecad female melanoma cell lines exhibited GRPR production compared to Δ Ecad male and Ecad melanoma cell lines (Supplementary Table 7).”

- o - 0 - o -

R1.Q16. Page 9: “As previously showed in Fig. 2j” (should be “shown”)

R1.A16.

Thank you, the issue has now been corrected.

Data presentation, availability and statistics:

- There are no objections to data presentation, availability and statistics. All points are sufficiently addressed by the authors.

Referee #2 (Remarks to the Author):

This is a very interesting and important manuscript and a praise to the authors for considering sex specific factors in their work, which could improve cancer management. In their manuscript Raymond and colleagues explored the role of E-cadherin loss in the progression of sex-hormone dependent cancers. Providing in vivo and in vitro data, they uncovered a novel axis which involves GRPR/YAP1 in the molecular mechanism of estrogen response in different cancers. Data are novel and it is of interest that the aim of the manuscript is to change the perspective of estrogen-dependent cancers. Data and methodology are clearly presented with appropriate statistical analyses.

- o - 0 - o -

R2.Q1. & R2.Q2. (Reviewer 2 – Questions 1 and 2)

The flow of the reading should be improved, I would suggest to change the order of some figures, to first describe the effect of E-cadherin loss in lung metastasis promoted by GRPR (figures 1,2 and 5), followed by the role of estrogens in the activation of GRPR in both melanoma (figure 6) and breast cancer (ext fig.7) and finally concluding by depicting the molecular mechanism of GRPR in cell growth and invasion and the role of YAP1 (figures 3 and 4).

In this manner, the explanation of the molecular mechanism with a complete signal cascade (from E-cadherin to YAP1) would be at the end of the manuscript and completely depicted in the scheme in figure 6m. This scheme could be improved.

Please note that the figure numbering has been updated. All responses refer exclusively to the new numbering. We included the term “new” before figures in the response to ensure clarity and prevent confusion.

R2.A1. & R2.A2. (Reviewer 2 – Answers 1 and 2)

We thank the referee for this helpful suggestion. In fact, we considered several options for structuring the article and propose the following revised flowchart that incorporates the referee's comments. The article will be organized as follows:

- (i) epidemiology: highlighting E-cadherin's role in young female melanoma, confirmed in a mouse model,
- (ii & iii) identification and validation: GRPR as a key player in metastasis formation, both in vitro and in vivo,
- (iv & v) cellular and molecular processes associated with GRPR expression in vitro,
- (vi) signaling pathway: full characterization of the molecular signaling molecular pathway (CDH1-CTNNB1-ESR1-GRPR-YAP).

As suggested by the referee, we have improved the scheme previously presented in Fig. 6m, which is now shown as new Extended Data Fig. 8.

In this revised version, the article concludes with the complete ECAD-BCAT-ESR1-GRPR-YAP pathway, which underscores the key molecular mechanisms driving the rising cancer rates in young women. This pathway also offers a promising combination of therapeutic targets (ER α , GRPR, and YAP) for developing strategies to curb cancer progression, in conjunction with immunotherapy and/or targeted therapies. The insights gained from this pathway are valuable insights for both scientific research and clinical applications.

Regarding the figures:

Figure 1 remains unchanged

Extended Data figure 1 remains unchanged

Figure 2 remains unchanged

Extended Data figure 2 remains unchanged

Figure 3 corresponds to the previous figure 5

Extended Data figure 3 is a combination of the previous Extended Data Figure 3a,b and previous Extended Data figure 5

Figure 4 corresponds to the previous figure 3

Extended Data figure 4 corresponds to the previous Extended Data Figure 3c-o

Figure 5 corresponds to the previous figure 4

Extended Data figure 5 corresponds to the previous Extended Data figure 4

Figure 6 corresponds to the previous figure 6

Extended Data figure 6 became Extended Data figure 6 and Extended Data figure 7

Extended Data figure 8 corresponds to the previous Extended Data figure 8.

Extended Data figure 9 corresponds to the previous Extended Data figure 7.

Please note that new panels were incorporated into the different figures, and some original panels could have moved to other figures.

- o - 0 - o -

R2.Q3.

To better describe the link between E-cadherin and GRPR in lung metastasis it would be interesting to determine whether Δ Ecad mouse melanoma cell lines would promote cell growth/invasion and inhibit apoptosis as described for GRPR cells (figure 3d-u), and whether the effect would be reverted upon knocking down GRPR in Δ Ecad cell lines.

R2.A3.

Before going into details here is a summary of our answers for this comment:

Part 1: Invasion Assays. Female Δ Ecad (1057) melanoma cells showed GRP-induced invasiveness, which was inhibited by RC-3095. In contrast, invasion was unaffected in other cell lines with different genotypes or sex (new Extended Data Fig. 4j-n).

Part 2: Anoikis Resistance. GRP induced anoikis resistance in female Δ Ecad (1057) cells, but had no effect on anoikis resistance in other cell lines with different genotypes or sexes. (new Extended Data Fig. 4h-i).

Part 3: Grpr Knockout Attempt. CRISPR-mediated Grpr knockout in 1057 and 1069 female Δ Ecad cells failed due to its requirement for cell growth (new Extended Data Fig. 3c-i).

- o - 0 - o -

Part 1

We conducted a series of **invasion assays** using the following mouse melanoma cell lines: male Ecad (1181), female Ecad (1039), male Δ Ecad (1456), and female Δ Ecad (1057). Consistent with previous observations, female Δ Ecad (1057) cells displayed high invasiveness in the presence of GRP, which was inhibited by GRP + RC-3095. In contrast, the 1181, 1039, and 1456 cell lines

showed no invasive behavior, and their invasion was neither induced nor suppressed by GRP or RC-3095.

These findings are presented as new Extended Data Figure 4j-n.

Figure Legend:

j, Invading 1057 mouse melanoma cells were identified by counting DAPI-stained nuclei that crossed the Matrigel layer. Scale bar = 500 μ m.

k-n, Quantification of invading cells following GRP induction (10 nM) with or without RC-3095 in each mouse melanoma cell line: male Ecad 1181 (**k**), female Ecad 1039 (**l**), male Δ Ecad 1456 (**m**), and female Δ Ecad 1057 (**n**). Statistical analysis was performed using the unpaired Mann-Whitney test.

Main text

In the text, we updated the paragraph to “Furthermore, GRP induction promoted invasion exclusively in female Grpr-positive 1057 cells, but not in other cell lines derived from the mouse model, including Ecad-positive (male 1181 and female 1039) and male Ecad-negative (1456) cell lines (Extended Data Fig. 4j-n). Furthermore, GRP-induced invasion in GRPR-positive cells, including 1057, 1064, MDA-MB-435S, Dauv-1, 1014-GRPR, 1181-GRPR, and 501mel-GRPR, were inhibited in the presence of RC-3095 (Fig. 4p,q,t,u and Extended Data Fig. 4o,p,q,r,t). As expected, this inhibition was not observed in cells lacking GRPR (1014, 1181, and 501mel) in the presence of GRP (Fig. 4r,s and Extended Data Fig. 4s).”

- o - 0 - o -

Part 2

We previously evaluated anoikis resistance in female Δ Ecad (1057) and male Ecad (1181) cells (see new Fig. 4j,l). In addition, we conducted a new series of **anoikis assays** using the female Ecad (1014) and male Δ Ecad (1062) cells. We showed that GRP-induced anoikis resistance is exclusive to female Δ Ecad melanoma cells. These results are presented in new Extended Data Fig. 4h,i.

Figure Legend:

h,i, Resistance to anoikis assays showing the percentage of apoptotic (apop.) cells after 48 hours without matrix attachment. 1014 (h) and 1062 (i) cells were seeded for 24 hours, then treated for 18 hours under low serum conditions (0.5% FBS) with 10nM GRP, RC-3095 1 μ M or no treatment. Statistical analysis was performed using ANOVA. Bars represent mean \pm SD.

Main text:

After the sentence "Upon inhibition of GRPR with RC, the cells lost their resistance to anoikis (new Fig. 4j-o)." we added "As expected, GRP treatment did not promote anoikis resistance in male Δ Ecad and female Ecad cells (new Extended Data Fig. 4h,i)."

- o - 0 - o -

Part 3

..... and whether the effect would be reverted upon knocking down GRPR in Δ Ecad cell lines.

We knocked down Grpr levels in 1057 melanoma cells stably infected with a doxycycline-inducible shRNA targeting Grpr, and observed a reduction of the level of Grpr (new Extended Data Fig. 3h – see below). However, this reduction did not affect Ccn1 levels following Grpr induction with GRP (new Extended Data Fig. 3i – see below). This result suggests that the remaining Grpr levels are sufficient for proper signaling.

To further investigate, we attempted to completely knock-out Grpr in two female Δ Ecad cell lines but were unsuccessful, indicating that Grpr deficiency in Δ Ecad cells leads to synthetic lethality or, at the very least, a severe growth defect. This finding highlights a compelling therapeutic opportunity. In details, we specifically targeted exon 2 of Grpr, using two guide RNAs (gRNA), each designed to target one end of the exon. This exon was chosen because it encodes the amino acids essential for Grp binding. Due to X inactivation, only one allele of Grpr is transcribed. We generated independent clones from the 1057 and 1069 female melanoma lines, respectively, with one allele inactivated. However, none of the clones were null.

In the experiment with a single guide, 480 cells of the mouse melanoma cell line 1057 were seeded, resulting in 91 clones (18.9%), but none were heterozygous. With a double guide, we seeded 1056 cells of the mouse melanoma cell line 1057, and 156 (14.7%) clones grew, of which three were heterozygous (1057 Grpr +/-). The expression of Grpr was similar in the heterozygous and control cells.

Similarly, with a single guide, 128 cells of the mouse melanoma cell line 1069 were seeded, resulting in 54 clones (42.1%), but again none were heterozygous. Using a double guide, we seeded 640 cells of the mouse melanoma cell line 1069, leading to 153 clones (23.9%) and 12 were heterozygous. Although 1069 cells cloned better than 1057 cells, Grpr expression was similar in the heterozygous and the control cells.

We then used 1057 Grpr +/- heterozygous cells to attempt to knock out the second GRPR allele and performed the same experiment as above. In the case of a single guide, 96 cells of the mouse melanoma cell line 1057 Grpr +/- were seeded, and 42 clones (43.7%) grew, but none were homozygous. With a double guide, we seeded 640 1057 Grpr +/- cells, resulting in 131 clones (20.5%), all of which were heterozygous, with none being homozygous. The Grpr expression was similar across all these cells.

Altogether, these results suggest that the inactivation of Grpr occurred on the inactive X chromosome, but not on the active X chromosome. This implies that the inactivation of Grpr in Δ Ecad cells leads to cellular lethality.

In summary, we attempted to knockout Grpr in 1057 and 1069 female Δ Ecad mouse cell lines using two Crispr guide RNAs surrounding exon 2 (new Extended Data Fig. 3c-g). We successfully generated three and twelve individual heterozygous clones for Grpr from 1057 and 1069 cells, respectively. However, the loss of one Grpr allele did not alter Grpr mRNA levels, and we were unable to generate any homozygous clones. Attempts to generate homozygous knockout cells from 1057 Grpr $^{+/-}$ heterozygotes also failed. Since Grpr is located on the X chromosome and subject to X-inactivation, we were able to knockout the inactive allele but not the active one, which explains the constant expression of Grpr in these cells. These results show that Grpr is essential for the growth of melanoma cells in the absence of E-cadherin.

Moreover, we stably infected 1057 mouse melanoma cells with Doxycycline-inducible shRNA targeting Grpr. Despite achieving a significant reduction of 75% in Grpr mRNA levels, the Grpr signaling remained unchanged (new Extended Data Fig.3h,i). These results suggest that the remaining Grpr in the cells is sufficient to maintain proper signaling. However, we were unable to reduce Grpr levels further, indicating confirming that Grpr is essential for the growth of melanoma cells in the absence of E-cadherin.

We have briefly included this information in new Extended Data Fig.3c-i.

Main text:

Due to space constraints, we kept the text concise. We have now included: “However, we were unable to completely knock-out or knock-down Grpr in female melanoma cells (new Extended Data Fig. 3c-i), suggesting that Grpr is essential for melanoma cell growth in the absence of E-cadherin.”

Figure Legend:

c, DNA and mRNA sequencing tracks of Grpr in 1057 mouse melanoma cells. High-throughput sequencing was performed on an Illumina NovaSeq 6000 and aligned to the mm10 mouse

genome using BWA (version 20080505). The tracks show a heterozygous Grpr c.394 T>G SNP, with the thymine allele in red and the guanine allele in blue. Exclusive detection of one allele in the RNA-seq data suggests that Grpr undergoes X inactivation.

d, Grpr knockout strategy via exon 2 deletion. A schematic diagram illustrates the approach to delete Grpr exon 2, with genotyping forward and reverse primer positions (PCR-F and PCR-R) indicated. 1057 mouse melanoma cells were transfected with plasmids targeting the 5' side of exon 2 or both the 5' and 3' sides, using Lipofectamine 2000. The guide RNA (gRNA) sequences were cloned into the pSpCas9(BB)-2A-GFP plasmid. GFP-positive cells were isolated by FACS and single cells seeded in 96-well plates for genomic DNA extraction.

e, Genotyping results. Representative genotyping outcomes are shown. (1) corresponds to Grpr wt/wt, (2) corresponds to Grpr wt/ Δ Ex2. DDW = double distilled water.

f, Clone efficiency in Δ Ecad mouse melanoma cell lines. The graph shows the percentage of clones obtained from three Δ Ecad cell lines (1057, 1069, 1057-Grpr wt/ Δ Ex2) targeted with one or two gRNAs. The 1057-Grpr wt/ Δ Ex2 clones were generated from the initial CRISPR attempt with the 1057 cell line. Statistical significance was determined using a Chi-square test.

g, Grpr expression in clonal populations. Expression levels of Grpr in clones generated before (1) and after CRISPR (2). Control clones were randomly selected. The crossed-out number three indicates that we were unable to obtain double homozygous knockout mutants Grpr/Cdh1.

h, Relative Grpr expression in shRNA-expressing cells. Relative expression in 1057 melanoma cells stably infected with a doxycycline-inducible shRNA targeting Grpr (i-shGrpr VSC11655, Horizon) is shown. The shRNA targets three regions: 3'UTR (5'-GACTTAATTGACCATACTT-3'), Ex1 (5'-TTTCGCATGGACTTGACCG-3'), and Ex3 (5'-TTTAGTCTAGACATACCCC-3'). Cells were selected with 1.2 μ g/mL puromycin and induced with 1 μ g/mL doxycycline for 3 days prior to RNA extraction.

i, Ccn1 expression following GRP induction. Relative Ccn1 expression in 1057 melanoma cells pre-treated with or without 1 μ g/mL doxycycline for 3 days, followed by a 4-hour treatment with 10 nM GRP. Statistical analyses for panels **k-m** were performed using a t-test.

- o - 0 - o -

R2.Q4.

Moreover, authors used both male and female cell lines in the in vitro experiments described in figure 3: it would be interesting to engraft also Δ Ecad male tumours together with female tumours, as described in figure 3 and ext. figure 3.

R2.A4.

As suggested by the reviewer, we engrafted male and female melanoma cells, with or without E-cadherin expression, into C57BL/6J mice. This data is now included in the new Extended Data Fig. 3b. Notably, only female Δ Ecad melanomas exhibited the most rapid tumor growth compared to other melanoma types.

Figure legend.

Time to reach a 1 cm³ tumor volume in neck-graft reimplantations of Ecad and ΔEcad melanoma in male and female mice. Statistical analysis was performed using an unpaired Mann-Whitney test.

Main Text

We revised the sentence from: “After engrafting melanoma tumors into C57BL6/J mice, we observed that ΔEcad female tumors grew significantly more efficiently than the Ecad female tumors (new Extended Data Fig. 3b).” to “After engrafting melanoma tumors into C57BL6/J mice, we observed that ΔEcad female tumors exhibited significantly faster growth compared to all other tumor types including Ecad male, Ecad female, and ΔEcad male tumors (new Extended Data Fig. 3b).”

- o - 0 - o -

R2.Q5.

Data in figure 4 describing the role of YAP1 should be completed at least with in vitro experiments performed in ΔEcad cell lines and assess whether these cell lines present upregulation of YAP1 and whether this upregulation is GRPR dependent.

R2.A5.

This comment prompted a response structured in three parts. A summary is given first

Part 1: Yap1/Taz Dependency. GRP-induced invasion in female ΔEcad (1057) cells depends on Yap1 and Taz, as shown by siRNA knockdown (new Fig. 5f).

Part 2: YAP1 expression and activation by GRP. YAP1 levels were consistent across melanoma cell lines, with no increase in ΔEcad females or following Grpr expression (new Extended Data Fig. 5i,j).

Part 3: Expanded GSEA and YAP1 Score Activation Analysis. GSEA and YAP score analyses were conducted on additional cell lines (new Extended Data Fig. 5k-Aa).

- o - 0 - o -

* Part 1 of the answer.

Yap1 and Taz were downregulated with specific siRNAs in 1057 female Δ Ecad cells, both in the presence and absence of GRP. Our results confirmed that GRP-induced invasion is dependent on Yap1 and Taz (new Fig. 5f).

Legend

We have updated the title of Figure 5 legend to: “GRPR Activation Induces YAP1-Transcriptional Program and cell invasion”. Additionally, panel f has been added to Figure 5. The legend is “Inhibition of GRP-induced cell invasion after Yap1, Taz, or Yap + Taz silencing.”

Main Text

To incorporate this information into the text, we added, “Finally, we demonstrate that GRP/GRPR-induced invasion relies on Yap1 and Taz (new Fig. 5f).”

- o - 0 - o -

* Part 2 of the answer:

Is there a difference in the level of YAP1 in male/female cells expressing or not Ecad?

Is the regulation of Yap1 dependent of Grpr?

Western blot analysis revealed that YAP1 expression remained consistent across all melanoma cell lines of interest (1181, 1014, 1456, and 1057), indicating that YAP1 expression is not regulated by Ecad. Furthermore, Grpr expression had no effect on YAP1 expression at the RNA level, showing that Grpr does not regulate Yap1 transcription. This data is now presented in new Extended Data Fig. 5i-j.

Figure legend

i, Western blot analysis was conducted to evaluate Yap1 protein levels in the specified mouse melanoma cell lines, with Actin used as a loading control. The data shown are from a representative experiment chosen from three independent biological replicates.

j, Expression of Yap1 mRNA in melanoma cell line expressing or not ectopic Grpr and controls not expressing Grpr (1014-Ctrl and 1014-Grpr [top] and 1181-Ctrl and 1181-Grpr [bottom]). Statistical significance performed by a t-test.

Main Text

The revised text now states “We did not observe an increase in YAP1 levels in Δ Ecad females or following Grpr expression in Ecad cell lines (new Extended Data Fig. 5i,j).”

- o - 0 - o -

* Part 3 of the answer

In the previous version, we showed the nuclear accumulation of YAP1 in cells induced with GRP (previous new Fig. 4d) and showed that the YAP1 signature was enriched after GRP induction in multiple mouse melanoma cell lines: 1057 (Δ Ecad, female, previous Fig. 4e), 1181-Grpr (Ecad, male, previous Extended Data Fig. 4g), and 1014-Grpr (Ecad, female, previous Extended Data Fig. 4i).

We have now expanded our analysis as follows:

- (i) We performed the same GSEA analysis on 1064 cells (Δ Ecad, female, new Extended Data Fig. 5k),
- (ii) We assessed the YAP score -based on Ccn1, Ccn2, Crim1, Tead4, and Lats2 -validated targets of the YAP/TAZ pathway in melanoma- across all six cell lines (new Extended Data Fig. 5p-u).
- (iii) Additionally, we conducted both GSEA and YAP score analyses on two human melanoma cell lines (Dauv-1 and MDA-MB-435S), which endogenously express GRPR, as well as 501mel-GRPR, which ectopically expresses GRPR. These results remained consistent, and these data are now presented in the new Extended Data Fig. 5v-Aa.

Figure legend

k-o, GSEA analysis of YAP1 activation signature in murine melanoma cell lines, 4 hours after stimulation with 10nM GRP. RNA-seq analyzed mRNA expression data were normalized using DEseq2 before GSEA.

p-u, Yap1 score after GRP induction in murine melanoma cell lines. Cells treated with vehicle, 10 nM GRP, 1 μ M RC, or both.

v-x, GSEA analysis of YAP1 activation signature in human melanoma cell lines, 4 hours after stimulation with 10nM GRP. RNA-seq analyzed mRNA expression data were normalized using DEseq2 before GSEA.

y-Aa, Yap1 score after GRP induction in human melanoma cell lines. Cells treated with vehicle, 10 nM GRP, 1 μ M RC, or both.

Main Text

Accordingly, we revised the sentence “Transcriptomic analysis of 1057, 1181-Grpr, and 1014-Grpr cells (GRPR-pos) showed the YAP1 activation signature and YAP score to be significantly increased upon GRPR activation, but not in 1181-Ctrl and 1014-Ctrl (GRPR-neg) cells (Fig. 4e and Extended Data Fig. 4f-i).” to “Transcriptomic analysis of 1057, 1064, 1181-Grpr, 1014-Grpr, Dauv-1, and 501mel-GRPR cells revealed a significant increase in both the YAP1 activation signature and YAP1 score upon GRPR activation, whereas no such increase was observed in 1181-Ctrl and 1014-Ctrl (GRPR-neg) cells (new Fig. 5e and new Extended Data Fig. 5k-Aa).

- o - 0 - o -

R2.Q6.

Authors conclusion about Δ Ecad/ERS1/GRPR pathway and its feedback loop should be completed or at least discussed, including the possibility (as described in the literature for estrogen-dependent cancers) that also estrogens might downregulate the expression of Ecad.

R2.A6.

The referee is fully right and we thank him/her for this comment. The downregulation of ESR1 in female Ecad (1014) leads to the upregulation of Ecad showing that there is indeed a feedback loop that would be independent of Grpr since Grpr is not produced in these cells. Additionally, we confirmed at the protein level that the downregulation of Cdh1 by siRNA leads to the upregulation of ER α (new Fig. 6k).

Figure legend

k, Western blot analysis of E-cadherin and ER α protein levels in mouse 1014 melanoma cells 48 hours after knockdown with siScr (control), siCdh1 (targeting E-cadherin), or siESR1 (targeting ER α). Actin served as a loading control. The displayed data represent a selected experiment from four independent replicates, with full data provided in the supplementary material. Histograms show the quantification of protein levels from these four experiments, presented as mean \pm SD. Statistical analysis used Mann-Whitney from normalized values.

Main text

At the protein level, we confirmed that the knockdown/knockout of CDH1 in both mouse and human melanoma cell lines leads to upregulation of ER α (Fig. 6k, Extended Data Fig. 6q and 7h-j). Furthermore, downregulation of ESR1 upregulates ECAD (Fig. 6k). Finally, we showed that regulation of ER α by E-cadherin is reciprocal, as knockdown of ESR1 induces ECAD levels (Fig. 6k).

Figure legend

l, Western blot analysis of E-cadherin (ECAD) expression in 501mel cells, with or without GRPR expression, in the presence or absence of 10 nM GRP. Actin served as a loading control. Statistical analysis was performed using data from five independent experiments, with the Mann-Whitney test applied to normalized values.

Main Text

We revised the previous and wrote “Finally, the CDH1/ESR1/GRPR axis operates through a positive feedback loop, where GRPR activation by GRP repressed CDH1 expression (Fig. 6l,m) and importantly E2 reinforced this loop by stimulating ER α activity and further inhibiting CDH1 expression (Fig. 6k and Extended Data Fig. 7k).”

- o - 0 - o -

R2.Q7.

To improve the clinical impact of the use of estrogen or GRPR inhibitors in Ecad-deficient tumours, it would be interesting to perform *in vivo* or *in vitro* experiments in Δ Ecad mice or cell lines treated with RC or ICI to assess whether there is any effect on their invasive capacity.

R2.A7.

In this answer, we will have two parts: *in vitro* then *in vivo*.

(i) *In vitro*

We conducted a series of *in vitro* invasion assays using the female Δ Ecad (1057) cell line, along with control cell lines including male Ecad (1181), female Ecad (1039), and male Δ Ecad (1456) cells. Female Δ Ecad (1057) cells show sensitivity to GRP or ICI-182,780 (=ICI). As expected, ICI alone has no effect on the invasion potential of 1057 cells, while GRP promotes cell invasion. However, this effect is attenuated when GRP is combined with ICI. Additionally, we confirmed that ER α levels were reduced at the protein level in 1057 cells treated with ICI. In contrast, the invasion potential of 1181, 1039, and 1456 cells is neither induced nor repressed by GRP or ICI, likely due to the absence of GRPR in these cells. This findings are presented in new Extended Data Fig. 7l-r.

Figure Legend:

l, Invading 1057 mouse melanoma cells were identified by counting DAPI-stained nuclei that crossed the Matrigel layer. Scale bar = 500 μ m. **m-p**, Quantification of invading cells under various conditions for each mouse melanoma cell line: male Ecad 1181 (**m**), female Ecad 1039 (**n**), male

Δ Ecad 1456 (o), and female Δ Ecad 1057 (p) treated with 10 nM GRP , 1 μ M RC-3095, and/or 1 μ M ICI. Statistical analysis was performed using the unpaired Mann-Whitney test. q,r, Effect of ICI treatment on ER α protein levels in murine 1057 melanoma cells. q, Western blot analysis shows ER α expression under three conditions: untreated (NT), mock-treated (mock), or treated with ICI-182,780 (1 μ M, 24 h). Actin served as a loading control. r, Histograms represent the quantification of ER α levels based on three independent experiments.

Main text:
See below

(ii) *in vivo*

We assessed the effects of ICI182,780 (ICI also called Fulvestrant - a ER α degrader) by two strategies. First, we treated the mouse melanoma cells in culture with ICI prior to injecting them into the tail vein, mimicking the use of the drug as an adjuvant following initial surgery on the primary tumor. Second, we administered Fulvestrant (Zentiva®) to the mice at different time points following tail vein injection of untreated melanoma cell lines. Melanoma metastasis in the lung was evaluated 25 days post-tail vein injection, both through visual observation and molecular analysis, as shown in new Fig. 6n.

We added a dedicated paragraph on *in vivo* ICI treatment between the sections “GRPR is expressed in females through a CDH1-CTNNB1-ESR1-GRPR axis” and “The CDH1-GRPR axis is active in human breast carcinomas”. This paragraph integrates both the *in vitro* and *in vivo* findings related to ICI treatment.

Figure legend

n, Quantification of visible lung metastases using a stereomicroscope. The number of injected mice and mice with detectable metastases is indicated. Statistical significance was determined using Fisher's exact test with Bonferroni-adjusted p-values. PCR-based quantification of 1057 melanoma cell burden in the lungs. The number of cells was determined by measuring the Cre transgene (as melanoma specific marker) levels in extracted lung DNA. Statistical analysis was performed using the unpaired Mann-Whitney test.

Main text: A new paragraph was written

Estrogen signaling drives invasiveness and metastasis in Grpr-positive melanomas and is effectively inhibited by Fulvestrant

To assess the clinical relevance of estrogen signaling in Grpr-positive melanomas, we investigated the impact of estrogen inhibition using ICI 182,780 (Fulvestrant) both *in vitro* and *in vivo*.

In vitro, treatment with ICI significantly reduced GRP-induced invasion of 1057 ΔEcad melanoma cells, while no effect was observed in Grpr-negative cell lines derived from transgenic mice (new Extended Data Fig. 7I-r).

For *in vivo* analysis, we used two approaches. First, 1057 ΔEcad melanoma cells were either pre-treated with ICI 182,780 for three days or left untreated before being injected into the tail vein of female C57BL/6J mice. We treated the cells with ICI prior to injecting them into the tail vein, mimicking the use of the drug as an adjuvant following initial surgery on the primary tumor. Second, we injected untreated 1057 cells into the tail vein of female C57BL/6J mice and subsequently treated the mice with Fulvestrant (50 mg/kg) three hours post-injection and once weekly for three weeks.

For both approaches, after 25 days—a timeframe established by previous IVIS experiments—the mice were humanely sacrificed, and lung metastases were analyzed (new Fig. 6n). In the first approach, 9 out of 13 mice injected with ICI 182,780-pretreated 1057 cells showed no visible metastases, compared to only 1 out of 11 mice in the untreated group (new Fig. 6n). In the second approach, 5 out of 6 mice treated with Fulvestrant showed no visible metastases. Additionally, the metastatic burden in the lungs, quantified via qPCR targeting the Cre transgene specific to the 1057 cell genome, was reduced in mice injected with ICI-pretreated 1057 cells and Fulvestrant-treated mice (new Fig. 6n).

Collectively, these findings show that estrogen signaling enhances both the invasiveness and metastatic potential of Grpr-positive melanoma cells, processes that can be effectively suppressed by ICI/Fulvestrant *in vitro* and *in vivo*.

Minor points:

- o - 0 - o -

R2.Q8.

In keywords add sex

R2.A8.

We added “sex” in the keywords.

Referee #3 (Remarks to the Author):

This study tackles the important question of sex-bias in tumor progression arises, especially for tumors that are not conventionally thought to be sex-hormone dependent. The work is motivated by the authors' observation that there is a disproportionate incidence of cancers on premenopausal women. The authors test this experimentally for melanoma, which is disproportionately affected in their human data and where they show that lung metastasis in mouse models is increased in E-cadherin KO tumors from female animals. They propose a model where (as I understand it) E-cadherin dysfunction upregulates b-catenin signaling, which increases expression of GRP-receptor via upregulation of the estrogen receptor. The latter would presumably be key to conferring the sex-preference for premenopausal females.

Of note, activation of GRPR signaling downregulated E-cadherin expression, suggesting that a positive feedback loop would allow this pathway to be amplified by dysfunction at various steps. These are interesting findings, but they also raise a number of questions in my mind.

Major questions.

- o - 0 - o -

Please note that the figure numbering has been updated. All responses refer exclusively to the new numbering. We included the term "new" before figures in the response to ensure clarity and prevent confusion.

R3.Q1. (Reviewer 3 – Question 1)

1. Preclinical significance. Although a wide range of approaches are used, many key experiments are performed using tumor cell lines, either in vitro or as xenografts, including cell lines where GRPR is overexpressed. While this is valuable to probe mechanism, I think that to reinforce the relevance of these findings, key results and predictions would be stronger if they could be tested in the authors' mouse model. In particular, the model predicts that

R3.A1. (Reviewer 3 – Answer 1)

We fully agree with the referee that demonstrating the effects of RC-3095 in transgenic mice would have strengthened our conclusions. However, our primary goal in this study was to establish a proof of concept that pharmacologically targeting GRPR can effectively inhibit melanoma metastasis.

Why not testing RC-3095 on our mouse model?

- 1) It will take a minimum of 4 years to get an answer (crosses from individual mutation, generation of tumors, drug treatment, and analysis)
- 2) RC-3095 is not the appropriate drug since it is not applicable in humans and very unstable.

Once, we will isolate novel, efficient and stable hit(s) against Grpr, it will be the right time to achieve such experiment.

The transgenic model in question (TyrNrasQ61K^o; Ink4a+/-; TyrCre^o; Cdh1 fl/fl) presents significant breeding challenges due to the hemizygous nature of two alleles and the heterozygous nature of one allele, requiring a complex and specific breeding strategy. While using this model

would have been ideal, for ethical and economic reasons, we had to discontinue the breeding necessary to generate this genotype (frozen embryos are available for each individual mutation).

Generating the required genetic combinations, inducing tumors, and conducting treatment experiments would take approximately four years under optimal conditions, assuming no unexpected complications arise. Also RC-3095 is unstable (new Extended Data Fig. 3q, Schwartzmann et al., 2006), which makes it unsuitable for human use and expensive. Moreover, treating these mice would require precise timing, which would vary from individual to individual based on the noticeable apparition of primary tumors, and an expensive drug that, as noted, is not viable for clinical application. We therefore believe this approach would not be feasible.

That said, we fully acknowledge the value of evaluating GRPR antagonists in more complex genetic models. Currently, we are actively seeking new GRPR inhibitors with similar properties to RC-3095 but improved stability *in vivo*. Once a suitable candidate is identified, we intend to conduct these experiments in an appropriate transgenic model.

a) GRPR would be differentially upregulated in lung metastases in female mice; and

a) To address this question, we assessed *Grpr* levels in lung metastases from both male and female transgenic mice, with and without E-cadherin. RNAscope was successfully performed on lung melanoma metastases from both male and female mice, with or without *Cdh1*. Melanoma cells were effectively labeled using probes for *Dct* (red) and *Grpr* (green), and counterstained with DAPI, providing clear and detailed visualization. The RNAscope demonstrate that only Δ Ecad female metastases expressed *Grpr*. This panel is presented in new Extended Data Fig. 2c.

c

Figure legend

c, RNAscope analysis of mouse lungs from *Ecad* and Δ *Ecad* male and female transgenic mice, showing *Grpr* (green) and *Dct* (red) as a melanoma marker. Notably, Δ *Ecad* females with lung metastasis exhibit abundant *Grpr* expression.

Main text

After the sentence “This finding was supported by the H3K27ac ChIP-seq data, which showed that only Δ *Ecad* female melanoma harbors an active *Grpr* promoter (new Extended Data Fig. 2b), we added “*Grpr* expression was preserved in lung metastases in females (new Extended Data Fig. 2c).”

- o - 0 - o -

- b) metastasis would be reduced by
i) inhibiting GRP-GRPR signaling and

We demonstrated that RC-3095 (GRPR antagonist) impacts Grpr signaling using an appropriate **allograft** model (see new Fig. 3f-i, and Extended new Fig. 3r-u). Our results showed a significant reduction in the number of mice with lung metastases (new Fig. 3h), and in cases where metastases were present, the number of metastatic cells was notably lower (new Fig. 3i).

- o - 0 - o -

- b) metastasis would be reduced by
ii) inhibiting estrogen signaling (either at the receptor or by ovariectomy). I appreciate that the authors have used xenograft models as a more facile way to test this question, but the application of their model for cancer would be stronger if it were done in a more complex genetic model.

To answer the referee's point regarding estrogen signaling in GRPR signaling, we first looked in vitro, by treating the four melanoma cell lines types with an ER α inhibitor (ICI 182,780) to evaluate Grpr activity through IP1 production (new Extended Data Fig. 5a-d). Our results show that GRP stimulated IP1 production only in female Δ Ecad cells, and that this production is reduced in the presence of RC or ICI.

Figure legend:

a-d, IP1 levels, a readout of G α q/11 activation, were measured in mouse melanoma cell lines following stimulation with 10 nM GRP, with or without treatment with the GRPR inhibitor RC-3095 or ER α inhibitor Fulvestrant/ICI. Statistical differences were analyzed using ANOVA followed by Holm-Šídák's multiple comparison test.

Main text:

We now find in the text the following sentences: "Treatment with the ER α agonist 17 β -estradiol (E2) increased Grpr expression, while treatment with the ER α degrader ICI 182,780 (ICI) decreased its expression (new Extended Data Fig. 7k). This effect on Grpr expression was associated with a decrease in its activity, as shown by the reduced IP1 induction by the ICI treatment (new Extended Data Fig. 5a-d)."

Moreover, we evaluated the consequences of ICI on invasion *in vitro* and *in vivo*.

We conducted a series of *in vitro* invasion assays using the female Δ Ecad (1057) cell line, along with control cell lines including male Ecad (1181), female Ecad (1039), and male Δ Ecad (1456). Female Δ Ecad (1057) cells show sensitivity to GRP or ICI. As expected, ICI alone has no effect on the invasion potential of 1057 cells, while GRP promotes cell invasion. However, this effect is attenuated when GRP is combined with ICI. Additionally, we confirmed that ER α levels were reduced at the protein level in 1057 cells treated with ICI. In contrast, the invasion potential of 1181, 1039, and 1456 cells is neither induced nor repressed by GRP or ICI, likely due to the absence of GRPR in these cells. These findings are presented in Extended Data Figure 7l-r.

Figure Legend:

l, Invading 1057 mouse melanoma cells were identified by counting DAPI-stained nuclei that crossed the Matrigel layer. Scale bar = 500 μ m.

m-p, Quantification of invading cells under various conditions for each mouse melanoma cell line: male Ecad 1181 (**m**), female Ecad 1039 (**n**), male Δ Ecad 1456 (**o**), and female Δ Ecad 1057 (**p**) treated with 10 nM GRP, 1 μ M RC-3095, and/or 1 μ M ICI. Data are representative of at least three independent experiments. Statistical analysis was performed using the unpaired Mann-Whitney test.

q,r, Effect of ICI treatment on ER α protein levels in murine 1057 melanoma cells. **q**, Western blot analysis shows ER α expression under three conditions: untreated (NT), mock-treated (mock), or treated with ICI-182,780 (1 μ M, 24 h). Actin served as a loading control. **r**, Histograms represent the quantification of ER α levels based on three independent experiments.

Main text:

See below

(ii) *in vivo*

We assessed the effects of ICI by two distinct strategies. First, we treated mouse melanoma cells with ICI prior to injecting them into the tail vein, mimicking the use of the drug as an adjuvant

following initial surgery on the primary tumor. Second, we administrated Fulvestrant (Zentiva®) weekly to the mice following tail vein injection of melanoma cell lines. Melanoma metastasis in the lung was evaluated 25 days post-tail vein injection, both through visual observation and molecular analysis, as shown in new Fig. 6n. We showed that both ICI pretreatment and Fulvestrant treatment significantly reduced lung colonization.

We added a dedicated paragraph on *in vivo* ICI treatment between the sections “GRPR is expressed in females through a CDH1-CTNNB1-ESR1-GRPR axis” and “The CDH1-GRPR axis is active in human breast carcinomas”. This paragraph integrates both the *in vitro* and *in vivo* findings related to ICI treatment.

Figure legend

n, Quantification of visible lung metastases using a stereomicroscope. The number of injected mice and mice with detectable metastases is indicated. Statistical significance was determined using Fisher's exact test with Bonferroni-adjusted p-values. PCR-based quantification of 1057 melanoma cell burden in the lungs. The number of cells was determined by measuring the Cre transgene (as melanoma specific marker) levels in extracted lung DNA. Statistical analysis was performed using the unpaired Mann-Whitney test.

Main text: A new paragraph was written

Estrogen signaling drives invasiveness and metastasis in Grpr-positive melanomas and is effectively inhibited by Fulvestrant

To assess the clinical relevance of estrogen signaling in Grpr-positive melanomas, we investigated the impact of estrogen inhibition using ICI 182,780 (Fulvestrant) both *in vitro* and *in vivo*.

In vitro, treatment with ICI significantly reduced GRP-induced invasion of 1057 ΔEcad melanoma cells, while no effect was observed in Grpr-negative cell lines derived from transgenic mice (new Extended Data Fig. 7l-r).

For *in vivo* analysis, we used two approaches. First, 1057 ΔEcad melanoma cells were either pre-treated with ICI 182,780 for three days or left untreated before being injected into the tail vein of female C57BL/6J mice. We treated the cells with ICI prior to injecting them into the tail vein, mimicking the use of the drug as an adjuvant following initial surgery on the primary tumor. Second, we injected untreated 1057 cells into the tail vein of female C57BL/6J mice and subsequently treated the mice with Fulvestrant (50 mg/kg) three hours post-injection and once weekly for three weeks.

For both approaches, after 25 days—a timeframe established by previous IVIS experiments—the mice were humanely sacrificed, and lung metastases were analyzed (new Fig.

6k). In the first approach, 9 out of 13 mice injected with ICI 182,780-pretreated 1057 cells showed no visible metastases, compared to only 1 out of 11 mice in the untreated group (new Fig. 6k). In the second approach, 5 out of 6 mice treated with Fulvestrant showed no visible metastases. Additionally, the metastatic burden in the lungs, quantified via qPCR targeting the Cre transgene specific to the 1057 cell genome, was reduced in mice injected with ICI-pretreated 1057 cells and Fulvestrant-treated mice (new Fig. 6k).

Collectively, these findings show that estrogen signaling enhances both the invasiveness and metastatic potential of Grpr-positive melanoma cells, processes that can be effectively suppressed by ICI/Fulvestrant *in vitro* and *in vivo*.

- o - 0 - o -

b) metastasis would be reduced by

iii) but the application of their model for cancer would be stronger if it were done in a more complex genetic model.

We already addressed this question at the beginning of the answer.

- o - 0 - o -

R3.Q2.

The causal role of b-catenin signaling. The authors propose that cadherin dysfunction leads to an upregulation of b-catenin/Wnt signaling as a key step in their pathway. The experimental evidence for this is not direct - based on E-cadherin inactivation and APC inactivation being associated with increased ESR expression. Although E-cadherin dysfunction has often been postulated to increase b-catenin/Wnt signaling, this is not always the case (e.g. Christofori lab: Herzig Oncogene, 2007 26: 2290) and both E-cadherin and APC dysfunction have other cellular effects. To make a clear case for b-catenin, the authors need to directly inhibit b-catenin signaling in their experiments where ESR, GRP are upregulated, with the predictions that i) this upregulation will be abrogated and ii) the predicted functional consequences will also be corrected.

R3.A2.

We conducted novel experiments to directly assess the role of β -catenin in our model. Knocked-down of β -catenin in Δ Ecad female melanoma cells resulted in the downregulation of β -catenin target genes, including *Esr1* and *Grpr*, and impaired GRPR/GRP-induced invasion. In contrast, β -catenin overexpression in E-cadherin-positive female melanoma cells led to the upregulation of β -catenin target genes, including *Esr1* and *Grpr*. Additionally, gene expression analysis, combined with chromatin accessibility data (H3K27Ac), revealed that five β -catenin target genes were specifically activated and expressed in Δ Ecad female melanoma cells derived from the transgenic model, but not in other genotypes.

We fully agree with the referee that β -catenin might not be the only pathway involved in activating ESR1 following the E-cadherin knockout or down-regulation. As noted by the referee, previous studies (Herzig et al. 2007; Jeanes et al. 2008) have shown that E-cadherin loss alters gene expression independently of β -catenin and affects interactions between E-cadherin and other proteins. To acknowledge this, we have added a sentence to the discussion: "Moreover, it has been shown that the loss of E-cadherin alters the expression of various genes independently of β -catenin (Herzig et al. 2007; Jeanes et al. 2008)."

Our study was not designed to evaluate all potential pathways but rather to identify one specific pathway. As stated in the discussion, this issue remains unresolved, and our primary

objective is not to fully elucidate all mechanisms involved. However, we have conducted additional experiments to further support our conclusions regarding β -catenin signaling activation.

In our initial submission, we presented multiple lines of evidence showing the involvement of β -catenin signaling in ESR1 induction following CDH1 knockdown/knockout.

- #1) H3K27ac ChIP-seq analysis of mouse melanoma cell lines identified female-specific Δ Ecad-associated chromatin signatures linked to β -catenin (LEF1) (new Fig. 6a).
- #2) Chromatin accessibility (H3K27ac) analysis revealed that five β -catenin target genes were selectively accessible in Δ Ecad female melanoma cells from the transgenic mice, but not in melanoma from other genotypes (new Extended Data Fig. 6a-e).
- #3) A LEF1 binding site was identified in the Esr1 gene, supporting a direct regulatory interaction (new Extended Data Fig. 6m).

In response to the referee's comments, we conducted additional experiments (detailed in R3A2 points 4-7).

- #4) β -catenin knockdown in Δ Ecad female melanoma cells resulted in downregulation of its target genes, including Esr1 and Grpr (new Fig. 6d).
- #5) β -Catenin knockdown also reduced GRPR/GRP-induced invasion (new Extended Data Fig. 6p).
- #6) β -catenin overexpression in E-cad positive female melanoma cells induced upregulation of β -catenin target genes, including Esr1 and Grpr (new Fig. 6c).
- #7) Gene expression analysis linked to chromatin accessibility (H3K27Ac) confirmed that five β -catenin target genes are specifically activated in Δ Ecad female melanoma cells from the transgenic model, but not in other genotypes (new Extended Data Fig. 6f-j).

Collectively, these findings further reinforce the role of β -catenin signaling in the regulatory pathway presented in new Extended Data Fig. 8.

Below are the details of the additional experiments :

- o - 0 - o -

R3A2 point #4. β -catenin knockdown and molecular consequences

We pharmacologically inhibited β -catenin signaling in Δ Ecad mouse 1057 cells by using the inhibitor iCRT3. Inhibition of β -catenin decreased the expression of known β -catenin target genes (e.g. Nkd1 and Axin2), and the expression of both Esr1 and Grpr. This information is included in new Fig. 6d

Figure legend

d, RT-qPCR analysis of *Nkd1*, *Axin2*, *Esr1*, and *Grpr* mRNA levels in Δ Ecad mouse 1057 melanoma cells treated with either siRNA targeting Ctnnb1 or the pharmacological β -catenin inhibitor iCRT3 (10 μ M, 48hrs). Gene expression reduction was statistically assessed using the Mann-Whitney unpaired t-test. Data are presented as mean \pm SD.

Main text:

We added in the text: “Conversely, inhibition of β -catenin signaling using siRNA and iCRT3 reduced the expression of *Nkd1*, as well as *Esr1* and *Grpr* in Δ Ecad mouse melanoma cells (Fig. 6d).”

We also assessed the effects of iCRT3 in human melanoma cells expressing E-cadherin and observed a reduction in ER α levels at both the RNA and protein levels (new Extended Data Fig. 6n,o).

Figure legend

n,o, RT-qPCR and Western blot analysis of *Esr1* levels in ECAD human melanoma cells treated with the pharmacological β -catenin inhibitor iCRT3 (80 μ M, 48 hours). mRNA and protein levels were analyzed using unpaired t-test. Data are presented as mean \pm SD.

Main text:

We added in the text: “The regulation of *ESR1* by β -catenin in human melanoma cells was validated through treatment with iCRT3, a β -catenin inhibitor (new Extended Data Fig. 6n,o).”

- o - 0 - o -

R3A2 point #5. β -catenin knock down and functional consequences

Regulation of invasion by β -catenin in female mouse melanoma cells expressing Grpr. We observed that while treating Δ Ecad mouse 1057 melanoma cells with GRP stimulated invasion, treating these GRP-stimulated cells with iCRT3 abrogated this invasion. This information is included in new Extended Data Fig. 6p.

Figure legend

p, Invasion assays of mouse melanoma cell lines expressing Grpr (1057), assessed after 24 hours, in the presence of siScr, siCtnnb1, iCRT3 and/or GRP. Statistical significance of invasion across conditions was determined using the Wilcoxon test. Data are presented as mean values \pm SD.

Main text:

The following sentence was included in the text: “Furthermore, treatment with iCRT3, siCtnnb1, or their combination significantly reduced GRP-induced cell invasion in mouse melanoma cells (new Extended Data Fig. 6p).”

- o - 0 - o -

R3A2 point #6 β -catenin overexpression and molecular consequences

We overexpressed β -catenin in Ecad positive cells (1014) and showed upregulation of *Nkd1* (a β -catenin target used as a positive control), as well as *Esr1* and *Grpr*. (new Fig. 6c).

C Female Ecad - 1014

Figure legend:

c, Expression of *Ctnnb1*, *Nkd1*, *Esr1* and *Grpr* as determined using RT-qPCR in Ecad mouse 1014 melanoma cells transfected with pcDNA3 (bcat -) or with a β -catenin expression vector (bcat +) known as bcat* (Delmas et al., 2007). Mann-Whitney unpaired t-test was used to evaluate the significance of expression of the various genes. Data shown as mean values \pm SD.

Main text:

Inhibition of CDH1 expression, ectopic transfection of β -catenin, or reduction of Apc levels in mouse melanoma cells expressing Ecad leads to an increased expression of β -catenin targets *Nkd1* and/or *Axin2*, as well as *Esr1* and *Grpr* (new Fig. 6b,c and new Extended Data Fig. 6k,l).

- o - 0 - o -

R3A2 point #7. Expression of β -catenin targets in various mouse melanoma cell lines

As we had shown in new Extended Data Fig. 6, the loci corresponding to *Apcdd1*, *Axin2*, *Nkd1*, *Notum*, and *Sp5* are open in female Δ Ecad melanoma cells but not in others, indicating β -catenin activation. To confirm the transcriptional induction, we provide evidence that the expression levels of these target genes are indeed higher in the Δ Ecad female melanoma cells compared to the other genotypes, as shown below. This information is included in new Extended Data Fig. 6a-j.

Figure legend

f-j, Expression of β -catenin targets (*Apcdd1*, *Axin2*, *Nkd1*, *Notum*, and *Sp5*) in male and female mouse melanoma cells with or without E-cadherin. Statistical analysis was performed using an unpaired *t*-test.

Main text:

We now included in the text: “This activation is primarily observed in Δ Ecad female melanoma and is accompanied by the expression of these target genes (new Extended Data Fig. 6a-j).”

- o - 0 - o -

R3.Q3.

An ancillary question here is if β -catenin signaling is upregulated by E-cadherin dysfunction in their models, what mechanism would allow this to happen. As noted above, E-cadherin loss does not always lead to an increase in the cytosolic β -catenin pool, unless there is another defect e.g. APC or destruction complex dysfunction. This is probably a subsidiary question, but it is relevant for understanding the precise mechanism.

R3.A3.

This question is very interesting but it is not at the heart of the article.

Even though, nobody deciphered all associated mechanisms, we have shown that the classical β -catenin targets are induced in mouse melanoma models. See for instance new Extended Data Fig. 6a-j.

- o - 0 - o -

R3.Q4.

The role of E-cadherin dysfunction. The authors’ model (as outlined in the final cartoon) identifies E-cadherin dysfunction as a key upstream step. Further, they present data to suggest that genomic changes in E-cadherin are more frequent in young women (compared with either older women or men) and E-cadherin transcript expression is also reduced in this group. Is the genomic change contributory? If so, a key issue is how genomic changes in E-cadherin become more frequent in young women. Would they not be expected to be found also in older women, or could the frequency be biased by cancer deaths in younger women?

The authors show evidence that ESR-GRP signaling can downregulate E-cadherin (considered further below). But if this were operational, how could it cause a genomic alteration?

R3.A4.

We appreciate the referee's insightful comment and share the curiosity regarding the high frequency of *CDH1* alterations. At this stage, we do not have sufficient evidence to propose that the *ESR1-GRPR* pathway directly induces *CDH1* alterations. However, it is worth noting that under certain conditions, YAP1 has been linked to DNA damage (e.g., Tan et al. 2008).

In young women, high estrogen levels activate the physiological *ESR1-GRPR* pathway. A well-established mechanism in breast cancer initiation and progression is the induction of DNA damage, as estrogen treatment can lead to double-stranded breaks and genomic instability (Caldon 2014). However, the precise molecular mechanisms remain unclear.

One possible explanation is that cells exposed to estrogen gain a selective advantage in the presence of GRP, potentially contributing to the observed increase in *E-cadherin* alterations (~3% across all cancers, ~2% in melanoma). This pattern does not hold for *CCND1*, which is rarely mutated (<1% across cancers and melanoma) but can be amplified in cancer (~5% across all cancers, ~6% in melanoma). Unlike *CDH1* alterations, *CCND1* amplifications/mutations typically occur later in tumor progression and do not preferentially arise between puberty and menopause.

In older women, *CDH1* genomic alterations are also present (new Fig. 1e presented in the previous version). However, it is possible that individuals with such alterations were underrepresented due to disease progression or the emergence of additional tumor-suppressor mutations. In response to the question of why these alterations are not more prevalent in older women, one could speculate that, unfortunately, affected individuals may not have survived to later ages.

As this question is beyond the primary focus of our study, we have opted to remove this panel (previously Figure 1e) from the manuscript.

- o - 0 - o -

R3.Q5. & R3Q6

The role of positive feedback. The authors postulate that there is a positive feedback pathway, such as enhanced ER-GRPR signaling would downregulate E-cadherin. This makes the model more complex, but not be surprising (since feedback is so prevalent). However, the evidence for feedback is based on short term studies using cell lines expressing exogenous GRPR. This is a useful test of possibility, however it needs to be tested in a more realistic model. I appreciate that this is difficult to do in their genetic model, but it could potentially be done in xenograft models.

For example, they could test if overactivation of b-catenin in Ecad- positive cells upregulates ESR-GRPR signaling;

R3.A5.

First, as requested by the referee, we overexpressed β -catenin in Ecad positive cells (1014) and showed an upregulation of *Nkd1* (known β -catenin target), *Esr1*, and *Grpr*. (new Fig. 6c).

C Female Ecad - 1014

Figure legend:

RT-qPCR measured *Ctnnb1*, *Nkd1*, *Esr1* and *Grpr* mRNA levels in Ecad mouse 1014 melanoma cells transfected with pcDNA3 (bcat -) or with a β -catenin expression vector (bcat +) known as bcat* (Delmas et al., 2007). Mann-Whitney unpaired t-test was used to evaluate the significance of expression of the various genes. Data shown as mean values \pm SD.

Second, we used siCdh1 in Ecad positive cells (1014) and showed similar induction of gene expression (new Fig. 6b).

b Female Ecad - 1014

Figure legend:

Expression of *Ctnnb1*, *Nkd1*, *Esr1* and *Grpr* mRNA as determined by RT-qPCR in Ecad mouse 1014 melanoma cells transfected with siScr or siCdh1. Mann-Whitney unpaired t-test was used to evaluate the significance of expression of the various genes. Data shown as mean values \pm SD.

Main text:

Inhibition of CDH1 expression, ectopic transfection of β -catenin, or reduction of Apc levels in mouse melanoma cells expressing Ecad leads to an increased expression of β -catenin targets *Nkd1* and/or *Axin2*, as well as *Esr1* and *Grpr* (new Fig. 6b,c and new Extended Data Fig. 6k,l).

- o - 0 - o -

and whether this is upregulation of ESR-GRPR causes E-cadherin to be downregulated.

We evaluated the feedback using an in cellulo model that we established after transfecting human 501mel and mouse 1014 melanoma cell lines with an expression vector for GRPR and a Mock.

We then evaluated the consequences of the presence of GRP at the protein and RNA levels (new Fig. 6l,m). Panel “m” was already presented in the previous version.

Please note, established GRPR-expressing cells (501mel-GRPR) produce less E-cadherin than mock-transfected cells (501mel-Ctrl) (see panel l, lines 1 and 3).

Figure legend:

l, Western blot analysis of E-cadherin (ECAD) expression in 501mel cells, expressing or not GRPR, in the presence or absence of 10 nM GRP. Actin served as a loading control. Statistical analysis was performed using data from five independent experiments, with an unpaired t-test applied to normalized values.

m, mRNA CDH1 levels in melanoma cells expressing GRPR, measured in the presence or absence of GRP and/or RC-3095. TMM = Trimmed Mean of M-values.

Main text:

We added in the text : “Finally, the CDH1/ESR1/GRPR axis operates through a positive feedback loop, where GRPR activation by GRP repressed CDH1 expression (new Fig. 6l,m)”

* To continue to evaluate this loop, we reduced the level of either Cdh1 or ESR1 in 1014 mouse melanoma cells to evaluate the consequences on the level of Ecad and ERα (new Fig. 6k).

Figure legend

k, Western blot analysis of E-cadherin and ERα protein levels in mouse 1014 melanoma cells 48 hours after knockdown with siScr (control), siCdh1 (targeting E-cadherin), or siESR1 (targeting

ER α). Actin served as a loading control. The displayed data represent a selected experiment from four independent replicates, with full data provided in the supplementary material. Histograms show the quantification of protein levels from these four experiments, presented as mean \pm SD. Statistical analysis used Mann-Whitney from normalized values.

Main text

We included “At the protein level, we confirmed that the knockdown/knockout of CDH1 in both mouse and human melanoma cell lines leads to upregulation of ER α (Fig. 6k, Extended Data Fig. 6q and 7h-j). Furthermore, downregulation of ESR1 upregulates ECAD (Fig. 6k). Finally, we showed that regulation of ER α by E-cadherin is reciprocal, as knockdown of ESR1 induces ECAD levels (Fig. 6k).”

- o - 0 - o -

References

Bertrand JU, Steingrimsson E, Jouenne F, Bressac-de Paillerets B, Larue L. Melanoma Risk and Melanocyte Biology. *Acta Derm Venereol.* 2020;100(11):adv00139

Bringuier PP, Umbas R, Schaafsma HE, Karthaus HF, Debruyne FM, Schalken JA. Decreased E-cadherin immunoreactivity correlates with poor survival in patients with bladder tumors. *Cancer Res.* 1993;53(14):3241–5

Caldon CE. Estrogen signaling and the DNA damage response in hormone dependent breast cancers. *Front Oncol.* 2014;4:106

D'Ecclesiis O, Caini S, Martinoli C, Raimondi S, Gaiaschi C, Tosti G, et al. Gender-Dependent Specificities in Cutaneous Melanoma Predisposition, Risk Factors, Somatic Mutations, Prognostic and Predictive Factors: A Systematic Review. *Int J Environ Res Public Health.* 2021;18(15):7945

Herzig M, Savarese F, Novatchkova M, Semb H, Christofori G. Tumor progression induced by the loss of E-cadherin independent of beta-catenin/Tcf-mediated Wnt signaling. *Oncogene.* 2007;26(16):2290–8

Jeanes A, Gottardi CJ, Yap AS. Cadherins and cancer: how does cadherin dysfunction promote tumor progression? *Oncogene.* 2008;27(55):6920–9

Landi MT, Bishop DT, MacGregor S, Machiela MJ, Stratigos AJ, Ghorzo P, et al. Genome-wide association meta-analyses combining multiple risk phenotypes provide insights into the genetic architecture of cutaneous melanoma susceptibility. *Nat Genet.* 2020;52(5):494–504

Lesche R, Groszer M, Gao J, Wang Y, Messing A, Sun H, et al. Cre/loxP-mediated inactivation of the murine Pten tumor suppressor gene. *Genesis.* 2002;32(2):148–9

Morali OG, Delmas V, Moore R, Jeanney C, Thiery JP, Larue L. IGF-II induces rapid beta-catenin relocation to the nucleus during epithelium to mesenchyme transition. *Oncogene.* 2001;20(36):4942–50

Mortazavi A, Williams BA, McCue K, Schaeffer L, Wold B. Mapping and quantifying mammalian transcriptomes by RNA-Seq. *Nat Methods*. 2008;5(7):621–8

Rambow F, Rogiers A, Marin-Bejar O, Aibar S, Femel J, Dewaele M, et al. Toward Minimal Residual Disease-Directed Therapy in Melanoma. *Cell*. 2018;174(4):843-855 e19

Raymond JH, Aktary Z, Larue L, Delmas V. Targeting GPCRs and Their Signaling as a Therapeutic Option in Melanoma. *Cancers (Basel)*. 2022;14(3):706

Serrano M, Lee H, Chin L, Cordon-Cardo C, Beach D, DePinho RA. Role of the INK4a locus in tumor suppression and cell mortality. *Cell*. 1996;85(1):27–37

Shimazui T, Schalken JA, Girolodi LA, Jansen CF, Akaza H, Koiso K, et al. Prognostic value of cadherin-associated molecules (alpha-, beta-, and gamma-catenins and p120cas) in bladder tumors. *Cancer Res*. 1996;56(18):4154–8

Tan K, Feizi H, Luo C, Fan SH, Ravasi T, Ideker TG. A systems approach to delineate functions of paralogous transcription factors: role of the Yap family in the DNA damage response. *Proc Natl Acad Sci U S A*. 2008;105(8):2934–9

Wendt J, Mueller C, Rauscher S, Fae I, Fischer G, Okamoto I. Contributions by MC1R Variants to Melanoma Risk in Males and Females. *JAMA Dermatol*. 2018;154(7):789–95

Yu F-X, Zhao B, Panupinthu N, Jewell JL, Lian I, Wang LH, et al. Regulation of the Hippo-YAP pathway by G-protein-coupled receptor signaling. *Cell*. 2012;150(4):780–91

Response to Reviewers

We sincerely thank the reviewers for their thoughtful and constructive feedback, and we are grateful for their positive assessment of our revised manuscript entitled "*Targeting GRPR for sex hormone-dependent cancer after loss of E-cadherin.*"

Referee #1 (Remarks to the Author):

Referee: In their revised version of the manuscript "Targeting GRPR for sex hormone-dependent cancer after loss of E-cadherin", the authors have conscientiously addressed all issues raised by the reviewers. They have added a substantial number of experiments, confirming their hypothesis and further strengthening their findings. All questions raised by me were sufficiently addressed, but I would like to specifically comment on selected points in the paper to provide some context:

Author: We are particularly grateful for your detailed reading of our revised manuscript and the helpful contextual comments you provided.

Referee: R1Q1: The authors could satisfyingly show the regulation of the E-cadherin-ESR1-GRPR axis on protein level. For GRPR, protein detection by western blot was not possible (obviously a known problem in the field), but the authors showed convincingly using several tools that GRPR function was regulated as suggested. Also, the authors showed that the GRPR expression in the overexpressers is in a similar range as endogenous GRPR.

Author: We appreciate your recognition of the challenges associated with detecting GRPR protein by Western blot and your acknowledgment of our multi-faceted approach to demonstrate its regulation and functional activity. We are pleased that you found our use of functional assays and expression validation convincing.

Referee: R1Q4: The revised manuscript now contains a detailed description of the search approach that resulted in the identification of CDH1 as possible key factor in gender specific cancer risk. Their new, updated analyses resulted in a slight increase of candidate genes (now also including KRAS and BRAF), but without altering the general statement and the specific relevance of CDH1 in this patient age group.

Author: Thank you for your positive evaluation of the updated search approach and your support of the inclusion of KRAS and BRAF, which emerged from the extended analysis. We agree that these additions do not change the overarching conclusion regarding the central role of CDH1 in gender-specific cancer risk and are glad this came across clearly.

Referee: R1Q9: In the previous version of the paper, the authors have included a figure that shows a significantly increased number of CDH1 mutations in "hormone dependent" compared to "non hormone dependent" cancers. As the authors derive the genetic information from public databases, reliable information about the functional consequence of the mutations is missing. As this hampers the interpretation of the figure, the authors have now removed it from the revised manuscript. I support this decision, as I believe that it is not relevant for the proposed mechanism. In my view, it is sufficient

to mention “alterations of ECAD by mutations, methylation, and/or EMT” (like written in the Discussion section, page 13).

Author: We appreciate your support for the removal of the mutation figure due to the limited interpretability of publicly available data regarding mutation functionality. We agree that this change helps focus the manuscript on mechanistically relevant findings and are pleased you found the revised discussion appropriate.

Referee: R2Q5: *The authors have provided additional information to show that GRPR-mediated murine melanoma invasiveness is mediated by YAP1 and TAZ1, using siRNA. The effect of GRPR signaling on YAP1 target genes was further confirmed in different mouse and human cell lines, using GRP and an GRPR inhibitors.*

Author: Thank you for acknowledging the additional experiments using siRNA and GRPR modulation across multiple models. We are encouraged that the data sufficiently demonstrate GRPR's role in driving invasiveness through YAP1/TAZ1.

Referee: R3Q2: *The involvement of beta catenin in the proposed signaling process was addressed using a pharmacological beta catenin inhibitor (iCRT3), demonstrating that ESR1 is downregulated in presence of the inhibitor. Also, GRP-induced invasiveness was decreased after beta catenin inhibition or knockdown. This shows that beta catenin is integral part of the E-Cad-beta-cat-ESR1-GRPR signaling axis.*

Author: We are grateful for your positive remarks regarding our use of beta-catenin inhibition and knockdown to demonstrate its key role in the signaling axis. We are pleased that these experiments strengthened the mechanistic insight of our proposed model.

Referee: *In conclusion, the authors were able further substantiate their findings. There is only one minor point, where I would recommend different wording: In conclusion, the authors were able further substantiate their findings. There is only one minor point, where I would recommend different wording:*

Author: Thank you for the helpful wording suggestion. We agree with your proposed revision and have updated the manuscript accordingly to avoid any ambiguity.

Referee: *In summary, this is a highly interesting manuscript that makes a large contribution to the understanding of cancer biology in women and offers therapeutic avenues for improving female health. I congratulate the authors to their excellent work.*

Author: We are deeply appreciative of your thoughtful evaluation and encouragement. Your comments greatly improved the clarity and strength of our manuscript.

- o - O - o -

Referee #2 (Remarks to the Author):

Authors have done an excellent job in revising the manuscript, taking into account all criticisms. I believe the manuscript can be accepted.

Author: We thank you very much for your supportive and encouraging comments. We are pleased that the revisions satisfactorily addressed your concerns, and we are grateful for your recommendation for acceptance.

- o - O - o -

Referee #3 (Remarks to the Author):

The authors have performed an extensive revision of this study, incorporating substantial new data that very reasonably address the points raised in my earlier review. I think that this is an important study.

Author: We sincerely thank you for your positive feedback and for recognizing the significance of our study. We greatly appreciate your acknowledgment of the extensive revisions and additional data included in the revised manuscript. Your support reinforces the importance of our findings, and we are grateful for your role in strengthening the final version of this work.